# Global Diversity and Updated Phylogeny of *Auricularia* (Auriculariales, Basidiomycota)

**DOI:** 10.3390/jof7110933

**Published:** 2021-11-03

**Authors:** Fang Wu, Ablat Tohtirjap, Long-Fei Fan, Li-Wei Zhou, Renato L. M. Alvarenga, Tatiana B. Gibertoni, Yu-Cheng Dai

**Affiliations:** 1Institute of Microbiology, School of Ecology and Nature Conservation, Beijing Forestry University, Beijing 100083, China; fangwubjfu2014@bjfu.edu.cn (F.W.); ablat31@bjfu.edu.cn (A.T.); longfeifan@bjfu.edu.cn (L.-F.F.); 2State Key Laboratory of Mycology, Institute of Microbiology, Chinese Academy of Sciences, Beijing 100101, China; liwei_zhou1982@im.ac.cn; 3Centro de Biociências, Departamento de Micologia, Universidade Federal de Pernambuco, Avenida da Engenharia, S/N—Cidade Universitária, Recife 50740-600, Brazil; renatolma@gmail.com (R.L.M.A.); tbgibertoni@hotmail.com (T.B.G.)

**Keywords:** Auriculariaceae, edible mushroom, multi-gene analysis, taxonomy

## Abstract

*Auricularia* has a worldwide distribution and is very important due to its edibility and medicinal properties. Morphological examinations and multi-gene phylogenetic analyses of 277 samples from 35 countries in Asia, Europe, North and South America, Africa, and Oceania were carried out. Phylogenetic analyses were based on ITS, nLSU, rpb1, and rpb2 sequences using methods of Maximum Likelihood and Bayesian Inference analyses. According to the morphological and/or molecular characters, 37 *Auricularia* species were identified. Ten new species, *A. camposii* and *A. novozealandica* in the *A. cornea* complex, *A. australiana*, *A. conferta*, *A. lateralis*, *A. pilosa* and *A. sinodelicata* in the *A. delicata* complex, *A. africana*, *A. srilankensis,* and *A. submesenterica* in the *A. mesenterica* complex, are described. The two known species *A. pusio* and *A. tremellosa*, respectively belonging to the *A. mesenterica* complex and the *A. delicata* complex, are redefined, while *A. angiospermarum*, belonging to the *A. auricula-judae* complex, is validated. The morphological characters, photos, ecological traits, hosts and geographical distributions of those 37 species are outlined and discussed. Morphological differences and phylogenetic relations of species in five *Auricularia* morphological complexes (the *A. auricula-judae*, the *A. cornea*, the *A. delicata*, the *A. fuscosuccinea* and the *A. mesenterica* complexes) are elaborated. Synopsis data on comparisons of species in the five complexes are provided. An identification key for the accepted 37 species is proposed.

## 1. Introduction

Members of *Auricularia* Bull. (Auriculariaceae, Auriculariales), typified by *A. mesenterica* (Dicks.) Pers. [1], are widely distributed and are recognized for their ecological and economic values and medicinal properties. Most species play an important role in degradation in forest ecosystems, especially in tropical forests, usually inhabiting angiosperm wood, such as dead trees, stumps, fallen trunks and branches, and rotten wood, with a few growing on gymnosperm wood [2,3]. Several species are widely used as important edible and medicinal mushrooms in China and other East Asian countries, e.g., *A. heimuer* F. Wu & al., and *A. cornea* Ehrenb. [4,5]. *Auricularia heimuer* was considered as a delicacy of the emperor in the Eastern Zhou Dynasty 2000 years ago [6], and it has been cultivated for over 1400 years [7]. In China in 2019, 7.1 billion kg (fresh weight) of *Auricularia heimuer*, valued at more than ¥36.5 billion RMB (≈$5.6 billion USD). The species is widely cultivated and is the second most important edible species, after *Lentinula edodes* (Berk.) Pegler, in China. Besides being a food, *A. heimuer* has properties of lowering blood sugar and fat levels, antitumor, antioxidant, and immunity enhancement [8,9].

Morphologically, *Auricularia* is characterized by gelatinous, resupinate to substipitate basidiomata with hairs on the upper surface, cylindrical to clavate and transversely three-septate basidia with oil guttules and hyaline, thin-walled and allantoid basidiospores [10,11,12,13,14]. Species identification, for a long time, has been based on macro-morphological characters, such as colour and size of basidiomata and length of hairs as introduced by Barrett and Kobayasi [15,16]. Lowy introduced the hyphal structure of internal stratification of basidiomata as a species differentiating character [10], and subsequently, this method of identification was accepted by Kobayasi [12], Lowy [17], and Li [18]. However, the macro-morphological characters of *Auricularia* species present plasticity, resulting in inaccurate identification of similar species.

The introduction of molecular methods revealed several misidentifications. Scientific names of some important species were revised, and some new species were described. A Chinese *Auricularia* species named as “Maomuer”, widely cultivated in southern China, was identified as *A. polytricha* (Mont.) Sacc. for over 100 years [4,19]. However, “Maomuer” is actually *A. cornea*, and *A. polytricha*, originally described from Jamaica, is a synonymy of *A. nigricans* (Sw.) Birkebak et al. [20]. Another Chinese species named as “Heimuer”, the most important cultivated species of *Auricularia* in China, has been mistakenly named *A. auricula-judae* (Bull.) Quél., a species originally described from Europe [21], but it is a new species, *A. heimuer* [5]. An additional ten new species, including *A. scissa* Looney et al., *A. subglabra* Looney et al., *A. angiospermarum* Y.C. Dai et al., *A. minutissima* Y.C. Dai et al., *A. tibetica* Y. C. Dai & F. Wu, *A. villosula* Malysheva, *A. brasiliana* Y.C. Dai & F. Wu, *A. orientalis* Y.C. Dai & F. Wu, *A. asiatica* Bandara & K.D. Hyde, and *A. thailandica* Bandara & K.D. Hyde were described after molecular analyses [22,23,24,25,26]. In addition, the phylogenetic analysis based on nLSU sequences showed that *Auricularia* was polyphyletic [27,28,29], but the genus was shown to be monophyletic in recent multi-gene phylogenetic analyses based on ITS and nLSU sequences [30,31].

Although afore-mentioned studies on phylogeny and taxonomy of *Auricularia* were carried out [20,22,23,24,25,26], they are either based on limited samples or focus on one species complex. Thus, the aim of this study is to improve the current knowledge of the phylogeny and species diversity of *Auricularia* worldwide by analyzing 277 samples (including 27 type specimens) from 35 countries in Asia, Europe, North and South America, Africa and Oceania. According to morphological examinations and phylogenetic analyses, 37 species of *Auricularia* are recognized, and morphological differences and phylogenetic relations of species in five *Auricularia* complexes, viz., *A. auricula-judae*, *A. cornea*, *A. delicata*, *A. fuscosuccinea*, and *A. mesenterica*, are elaborated.

## 2. Materials and Methods

### 2.1. Morphological Studies

180 specimens were deposited at the herbarium of the Beijing Forestry University (BJFC) and 97 specimens were borrowed from other herbaria, including Beijing Museum of Natural History (BJM), Institute of Microbiology, Chinese Academy of Sciences (HMAS), Royal Botanic Gardens (K), U.S. Forest Service, NRS, Center for Forest Mycology Research (CFMR), Komarov Botanical Institute of RAS (LE), University of Helsinki (H), Mae Fah Laung University (MFLU), Swedish Museum of Natural History (S), New Zealand Fungal and Plant Disease Collection (PDD), Royal Botanic Gardens Victoria (MEL), Universidade Federal de Pernambuco,, Departamento de Micologia (URM), National Museum of Nature and Science (TNS), Farlow Herbarium of Harvard University (FH) and the personal collection of J. Vlasák (JV). Descriptions of species are according to Wu et al. [23] and Lowy [10,17]. Colour terms follow Petersen [32]. Macro-morphological description of thirteen species based on 31 herbarium specimens are described after their whole basidiomata were rehydrated. Cross-sections of dried basidiomata were mounted in 5% potassium hydroxide (KOH) and 1% Congo Red and examined at 1000× magnification using a Nikon Eclipse 80i microscope. Microscopic structures were photographed using a Nikon Digital Sight DS-Fi1 camera. Basidiospores were measured in Cotton Blue (CB) and to test for cyanophily. Melzer’s reagent (IKI) was used to note any chemical reaction of spores. When presenting spore size variation, 5% of measurements from each end of the range were excluded and indicated in parentheses. The following abbreviations were used: IKI + = amyloid or dextrinoid reaction, IKI– = neither amyloid or dextrinoid, CB– = acyanophilous, L = mean spore length (arithmetic average of all spores), W = mean spore width (arithmetic average of all spores), Q = L/W ratio for each specimen studied, n (a/b) = number of spores (a) measured from given number of specimens (b).

### 2.2. DNA Extraction, PCR and Sequencing

CTAB rapid plant genome extraction kit-DN14 (Aidlab Biotechnologies Co., Ltd., Beijing, China) was used to obtain DNA from 277 dried specimens, according to the manufacturer’s instructions with some modifications. 2× EasyTaq PCR SuperMix (TransGen biotech, Beijing, China) was used for reaction to obtain PCR products. ITS region was amplified with primer pairs ITS5 (GGA AGT AAA AGT CGT AAC AAG G) and ITS4 (TCC TCC GCT TAT TGA TAT GC) [33]. Nuclear LSU region was amplified with primer pair LR0R (ACC CGC TGA ACT TAA GC) and LR7 (TAC TAC CAC CAA GAT CT) (https://sites.duke.edu/vilgalyslab/rdna_primers_for_fungi/, accessed on 3 September 2021), the rpb1 with primers Af (GAR TGY CCD GGD CAY TTY GG) and Cr (CCN GCD ATN TCR TTR TCC ATR TA) [34], and the rpb2 with primers b6F (TGG GGK WTG GTY TGY CCT GC) and b7.1R (CCC ATR GCT TGY TTR CCC AT) [35]. PCR conditions were as follows: for ITS and rpb2, initial denaturation at 95 °C for 3 min, followed by 34 cycles at 94 °C for 40 s, 54 °C for 45 s, 72 °C for 1 min; for nLSU, initial denaturation at 94 °C for 1 min, followed by 34 cycles at 94 °C for 30 s, 50 °C for 1 min, 72 °C for 1.5 min, for rpb1 initial denaturation at 95 °C for 3 min, followed by 9 cycles at 94 °C for 40 s, 60 °C for 40 s and 72°C for 2 min, then followed by 37 cycles at 94 °C for 45 s, 55 °C for 1.5 min and 72 °C for 2 min, and a final extension of 72 °C for 10 min. DNA sequencing was performed at Beijing Genomics Institute, China, with the same primers. Sequences of nLSU, rpb1 and rpb2 were generated in both directions.

### 2.3. Phylogenetic Analyses

All newly generated sequences and additional related sequences downloaded from GenBank are listed in Table 1 and were aligned using MAFFT 7.0 online service with the Q-INS-i strategy using the default parameters [36]. Multiple sequences were concatenated in Mesquite v. 3.10 [37]. Sequence alignments were deposited in TreeBase (submission ID 28649).

To define the high-rank phylogenetic relations within the Auriculariales, we used a dataset composed of concatenated ITS+nLSU sequences resulting in a final alignment comprising 156 different specimens. *Heteroradulum deglubens* (Berk. & Broome) Spirin & Malysheva was used as the outgroup. The concatenated ITS+nLSU dataset has an aligned length of 1953 characters, of which 600 characters are from locus ITS, and 1353 characters are from locus nLSU. We used another dataset composed of concatenated ITS+nLSU+rpb1+rpb2 sequences resulting in a final alignment comprising 106 different specimens to define phylogenetic relations within *Auricularia*, especially within closely related *Auricularia* species. *Elmerina efibulata* (Y.C. Dai & Y.L. Wei) Y.C. Dai & L.W. Zhou was used as the outgroup to root trees because *Elmerina* Bres. is closer to *Auricularia* than other genera of Auriculariales in phylogeny [30]. The concatenated ITS+nLSU+rpb1+rpb2 dataset has an aligned length of 3817 characters, of which 594 characters are from locus ITS, 1353 characters are from locus nLSU, 1336 characters are from locus rpb1, and 534 characters are from locus rpb2. All characters were equally weighted and gaps were treated as missing data.

Maximum Likelihood (ML) and Bayesian Inference (BI) methods were used for each dataset to enhance the reliability of phylogenetic analyses. The optimal substitution models suitable for both datasets were determined using the Akaike information criterion (AIC) implemented in MrModeltest 2.3 [38]. The ML tree was constructed using raxmlGUI 1.2 [39,40], and the BI tree was calculated with MrBayes3.2.5 [41]. The SYM+I+G and GTR+I+G models were selected as the best models for the ITS alignment and the nLSU alignment of the concatenated ITS+nLSU dataset respectively. The GTR+I+G model was selected as the best model for each alignment of the concatenated ITS+nLSU+rpb1+rpb2 dataset. Therefore, the ML analysis based on each dataset was calculated under the model GTRGAMMA. The BI analysis based on the concatenated ITS+nLSU dataset was calculated under the SYM+I+G model and the GTR+I+G model in two alignments, and based on the concatenated ITS+nLSU+rpb1+rpb2 dataset under the GTR+I+G model in four alignments. Four Markov chains were run for two runs from random starting trees for 1 million generations for the concatenated ITS+nLSU dataset, 1 million generations for the concatenated ITS+nLSU+ rpb1+rpb2 dataset, and trees were sampled every 100 generations. BI analysis stopped after effective sample sizes (ESSs) reached more than 200 and the potential scale reduction factors (PSRFs) were close to 1.000 for all parameters. The first one-fourth generations were discarded as burn-in. A majority rule consensus tree of all remaining trees was calculated. Branches that received bootstrap support for Maximum Likelihood (BS) and Bayesian Posterior Probabilities (BPP) greater than or equal to 50% (BS) and 0.90 (BPP) were considered as significantly supported, respectively.

## 3. Results

### 3.1. Phylogenetic Analyses

The ML analysis based on the concatenated ITS+nLSU dataset resulted in a similar topology as Bayesian Inference analysis, so only the ML tree is presented (Figure 1). The phylogeny demonstrated that 149 *Auricularia* specimens formed one large clade with high support, which confirmed the monophyletism of *Auricularia*. Most of the 31 *Auricularia* species formed monophyletic lineages with high support, and several species including *A. heimuer* and *A. submesenterica* didn’t form monophyletic lineages, however, these two species formed two distinct lineages with high support in the phylogeny based on the concatenated ITS+nLSU+rpb1+rpb2 dataset (Figure 2). The 31 *Auricularia* species formed three major clades, Clade A–C. Clade A includes 16 species in the *A. cornea*, *A. delicata* and *A. fuscosuccinea* complexes. Clade B includes seven species belonging to the *A. auricula-judae* complex, and Clade C includes eight species belonging to the *A. mesenterica* complex. Although species in the *A. auricula-judae* complex and the *A. mesenterica* complex clustered into two monophyletic clades, Clade B and Clade C respectively, species in the other three morphological complexes were scattered in small different clades in Clade A. The analyses showed that the morphological complexes do not fully correspond to the phylogenetic clades.

ML analysis based on the concatenated ITS+nLSU+rpb1+rpb2 dataset resulted in a similar topology as Bayesian analysis, so only the ML tree was presented (Figure 2). The phylogeny demonstrated a similar topology in *Auricularia* as the phylogeny based on the concatenated ITS+nLSU dataset (Figure 1), and also showed three major clades, Clade A–C. In Clade A, species of *A. delicata*, *A. cornea* and *A. fuscosuccinea* complexes did not form their own subclades. Species of *A. auricula-judae* and *A. mesenterica* complexes formed their own two clades, Clade B and Clade C.

### 3.2. Taxonomy

(1) *Auricularia africana* Y.C. Dai & F. Wu, sp. nov. Figure 3a and Figure 4.

MycoBank number: MB 825096.

Type—Uganda. Western Region, Kibale National Park, Kabarole District, Makere University, N 0°30′, E 30°24′, on dead wood, 20 April 2002, L. Ryvarden, *Ryvarden 44929* (K 133591, holotype).

Etymology—*Africana* (Lat.): refers to the distribution of the species in Africa.

Basidiomata—Gelatinous when rehydrated, greyish brown to fuscous, caespitose, resupinate to effused-reflexed; pileus sometimes observed, free lobed, margin undulate, projecting up to 1 cm, 1.5–2 mm thick, 0.2–0.4 mm thick when dry; upper surface tomentose, concentrically zoned with canescent zones and dark bands, becoming yellowish brown to orange-brown upon drying; hymenophore surface venose with folds, becoming greyish brown to fuscous upon drying.

Internal features—Medulla absent; crystals present, usually scattered in the hymenium; abhymenial hairs with a slightly swollen base, hyaline, thick-walled, with a narrow lumen, apical tips acute or obtuse, tufted, 300–500 × 1–2 µm; hyphae with clamp connections and simple septa, slightly inflated with a lumen in KOH, up to 4.5 µm in diam; basidia clavate, transversely 3-septate, with oil guttules, 50–80 × 4.5–6 µm, sterigmata sometimes observed; cystidioles absent.

Spores—Basidiospores allantoid, hyaline, thin-walled, smooth, usually with one or two large guttules, IKI–, CB–, (11.5–)12–14(–14.2) × (4.8–)4.9–5.4(–5.5) µm, L = 12.54 µm, W = 5.12 µm, Q = 2.45 (n = 30/1).

Distribution—Kenya and Uganda.

Notes—*Auricularia africana* belongs to the *A. mesenterica* complex and is characterized by resupinate to effused-reflexed basidiomata with concentric zones on the upper surface. It is macro-morphologically similar to *A. pusio* Berk. (Figure 3g), but differs from the later by shorter abhymenial hairs (300–500 µm vs. 400–800 µm), slightly narrower basidia and basidiospores (4.5–6 µm vs. 5–7.5 µm and 4.9–5.4 µm vs. 5–6 µm). In the phylogeny, *A. africana* is distantly related to *A. pusio* and forms a single lineage with high support (Figure 1 and Figure 2).

Additional specimen (paratype) examined—Kenya. Central Province, Nairobi, Karura Forest, on angiosperm tree, May 1992, A. Osano, *T 3* (K 34551).

(2) *Auricularia americana* Parmasto & I. Parmasto ex Audet, Boulet & Sirard

Figure 5a and Figure 6.

≡*Auricularia americana* Parmasto & I. Parmasto, Biblthca Mycol. 115: 137, 1987.

Basidiomata—Gelatinous when fresh, orange-brown to reddish brown, solitary, sometimes caespitose, sessile or substipitate; pileus cupulate or auriculate, sometimes with lobed margin, projecting up to 8.5 cm, 1–3 mm thick, 0.2–0.3 mm thick when dry; upper surface pilose, becoming greyish brown upon drying; hymenophore surface usually smooth, without folds, becoming fuscous to mouse-grey upon drying.

Internal features—Medulla present or absent, usually in the middle of the cross-section if present; crystals absent; abhymenial hairs with a slightly swollen base, hyaline, thick-walled, with a narrow lumen, apical tips acute or obtuse, tufted, 67–125(–150) × 4.5–6 µm; hyphae with clamp connections, 0.5–3.5 µm in diam. in KOH; basidia clavate, transversely 3-septate, with oil guttules, 55–71 × 4–5 µm, sterigmata rarely observed; cystidioles absent.

Spores—Basidiospores allantoid, hyaline, thin-walled, smooth, usually with one or two large guttules, IKI–, CB–, 14–16.5(–16.8) × (4.2–)4.5–5.4 µm, L = 15.38 µm, W = 4.92 µm, Q = 2.9–3.39 (n = 60/2).

Distribution—North America and north Asia.

Notes—*Auricularia americana* is one of two species that grows on gymnosperm wood. It differs from the other gymnosperm-inhabiting species, *A. tibetica* Y.C. Dai & F. Wu, by smaller basidia (55–71 × 4–5 µm vs. 70–103 × 4–7 µm) and smaller basidiospores (14–16.5 × 4.5–5.4 µm vs. 15–18.5 × 5.8–6.2 µm). *Auricularia americana* was originally described from Canada [13], and recently reported from China [5]. Samples from North America and north Asia have a few differences in molecular data and show intraspecific genetic variations in our phylogenies (Figure 1).

Specimens examined—China. Heilongjiang Province, Dailing, Liangshui, on dead tree of *Picea*, 26 August 2014, B.K. Cui, *Cui 11657* (BJFC 016841); Ningan County, Jingbohu Underground Forest Park, on fallen trunk of *Abies*, 5 September 2013, Y.C. Dai, *Dai 13461* (BJFC 014922), *Dai 13463* (BJFC 014924); Tangyuan County, Daliangzihe National Park, on fallen trunk of *Picea*, 25 August 2014, B.K. Cui, *Cui 11509* (BJFC 016751); Yichun, Fenglin Nature Reserve, on fallen trunk of *Picea*, 2 August 2011, B.K. Cui, *Cui 9887* (BJFC 010780). Inner Mongolia Autonomous Region, Hulun Buir, on fallen trunk of *Larix*, 12 August 2013, Y.C. Dai, *Dai 13636* (BJFC 015102). Jilin Province, Antu County, Changbaishan Nature Reserve, on fallen trunk of *Abies*, 7 September 2013, Y.C. Dai, *Dai 13476* (BJFC 014937), *Dai 13480A* (BJFC 014942), *Dai 13488* (BJFC 014949). Russia. Amur Region, Khingansky Reserve, 10 October 1991, *Mikulin* (VLA M-11352); Primorye Territory, Sikhote-Alin Nature Reserve, on fallen branch of *Pinus*, 16 August 2012, *Gromyko* (LE 296428). USA. Minnesota, Bohall Trail, Itasca State Park, on *Abies*, 6 August 1981, *Burdsall*, *HHB-11370* (CFMR), *HHB-11374* (CFMR); Wisconsin, Treehaven, Tomahawk, on fallen trunk of *Abies*, 7 September 1991, *Burdsall*, *HHB-14337* (CFMR).

(3) *Auricularia angiospermarum* Y.C. Dai, F. Wu & D.W. Li, sp. nov. Figure 5b and Figure 7.

MycoBank number: MB 840864.

=*Auricularia angiospermarum* Y.C. Dai, F. Wu & D.W. Li, in Wu et al., Mycol. Prog. 14(95): 8, 2015, *nom. inval.*, Art. F.5.6 (San Juan).

Type—USA. Connecticut, Windsor, N 41°51′, W 72°38′, on fallen trunk of *Quercus*, 4 December 2014, B.K. Cui, *Cui 12360* (BJFC 017274, holotype).

Basidiomata—Gelatinous when fresh, fawn to vinaceous brown, solitary, sometimes caespitose, sessile or substipitate; pileus cupulate or auriculate, sometimes with lobed margin, projecting up to 8 cm, 1–3 mm thick, 0.2–0.36 mm thick when dry; upper surface pilose, sometimes with a few folds, becoming greyish brown upon drying; hymenophore surface usually smooth, without folds, becoming vinaceous grey to fuscous upon drying.

Internal features—Medulla absent; crystals absent; abhymenial hairs with a slightly swollen base, hyaline, thick-walled, with a narrow lumen, sometimes separated, apical tips acute or obtuse, single or tufted, 80–140 × 5–6 µm; hyphae with clamp connections, 0.5–5 µm in diam in KOH; basidia clavate, transversely 3-septate, with oil guttules, 47–63 × 3.5–5 µm, sterigmata rarely observed; cystidioles absent.

Spores—Basidiospores allantoid, hyaline, thin-walled, smooth, usually with one to three large guttules, IKI–, CB–, (12.5–)13–15 × (4.5–)4.8–5.5(–6) µm, L = 13.72 µm, W = 4.97 µm, Q = 2.75–2.86 (n = 40/2).

Distribution—USA.

Notes—*Auricularia angiospermarum* was described as a new species by Wu et al. [23], but the name was invalid because the registration identifier cited in the protologue, ‘MycoBank no.: MB 812851′, was not issued for the name published. Therefore, the name is validated here.

The species was segregated from *A. americana*. Morphologically, there is no distinct differences between the two species and they were considered as a single species by Looney et al. [20]. However, these two species are clustered in two distinct lineages with strong support in the phylogenies (Figure 1 and Figure 2), and *A. angiospermarum* grows on angiosperm rather than on gymnosperm wood. *Auricularia angiospermarum* has so far been reported in North America only, while *A. americana* occurs both in North America and north Asia. The description taken from Wu et al. [23].

Specimens examined—USA. Arizona, Sycamore Canyon, on *Juglans*, 7 July 1980, *Burdsall*, *HHB-11037* (CFMR). Illinois, Pomona Natural Bridge, Shawnee National Forest, on angiosperm wood, 29 May 1986, *Nakasone, FP-102097* (CFMR). Wisconsin, Wyalusing State Park, on living tree of *Carya*, 31 May 1993, *Volk, TJV-93-12-Sp* (CFMR).

(4) *Auricularia asiatica* Bandara & K.D. Hyde Figure 3b,c and Figure 8.

Basidiomata—Gelatinous when fresh, fawn to greyish brown or buff to cream, caespitose, resupinate to effused-reflexed; pileus free lobed, margin undulate, projecting up to 5 cm, 1–3 mm thick, 0.12–0.16 mm thick when dry; upper surface villose, distinctly and concentrically zoned with wide whitish zones and thin black bands in the central pileal surface, becoming clay pink upon drying; hymenophore surface venose with folds, becoming dark greyish blue upon drying.

Internal features—Medulla absent; crystals present, usually scattered in hymenium; abhymenial hairs with a slightly swollen base, hyaline, thick-walled, with a narrow lumen, apical tips acute or obtuse, tufted, 800–1200 × 1.5–3 µm; hyphae with clamp connections, obviously inflated with a lumen in KOH, up to 11 µm in diam; basidia clavate, transversely 3-septate, with some small oil guttules, 40–52 × 3–6 µm, sterigmata usually observed; cystidioles absent.

Spores—Basidiospores allantoid, hyaline, thin-walled, smooth, usually with one or two large guttules, IKI–, CB–, (11–)11.2–12.3(–12.6) × (4.1–)4.5–5.2(–5.5) µm, L = 11.83 µm, W = 4.93 µm, Q = 2.4 (n = 30/1).

Distribution—Asia.

Notes—*Auricularia asiatica* was recently described from Asia by Bandara et al. [26]. It belongs to the *A. mesenterica* complex and is very similar to *A. brasiliana* Y.C. Dai & F. Wu by sharing resupinate to effused-reflexed, basidiomata with a free lobed pileus and similar-sized basidiospores, but the latter is distinctly and concentrically zoned with wide whitish zones and thin black bands throughout the pileal surface [25]. Phylogenetically, the species is distant from *A. brasiliana* and closely related to a new species, *A. srilankensis* (Figure 1 and Figure 2). Differences between *A. asiatica* and *A. srilankensis* are discussed in the notes on *A. srilankensis*.

Specimens examined—China. Hainan Province, Changjiang County, Bawangling Nature Reserve, on dead angiosperm tree, 2 July 2015, Y.C. Dai, *Dai 15285* (BJFC 019396); Wuzhishan, Wuzhishan Nature Reserve, on fallen angiosperm trunk, 14 November 2015, Y.C. Dai, *Dai 16149* (BJFC 020246), 15 November 2015, Y.C. Dai, *Dai 16224* (BJFC 020310). Indonesia. Sumatera Barat, Padang, Limau Manis, 29 July 2009, O. Miettinen, *OM 13932* (H).

(5) *Auricularia auricula-judae* (Bull.) Quél. Figure 5c and Figure 9.

≡*Tremella auricula-judae* Bull., Herb. Fr. (Paris) 9: Table 427, Figure 2, 1789.

=*Auricularia albicans* Berk., J. Linn. Soc., Bot. 13: 170, 1872.

=*Auricularia auricula* (L.) Underw., Mem. Torrey bot. Club12: 15, 1902.

=*Auricularia sambuci* Pers., Mycol. eur. (Erlanga) 1: 97, 1822.

=*Auricularia auricularis* (Gray) G.W. Martin, Am. Midl. Nat. 30: 81, 1943.

=*Auricularia lactea* (Quél.) Bigeard & H. Guill., Fl. Champ. Supér. France (Chalon-sur-Saône) 2: 491, 1913.

Basidiomata—Gelatinous when fresh, reddish brown to fuscous or white, solitary, sometimes caespitose, sessile or substipitate; pileus cupulate or auriculate, sometimes with lobed margin, projecting up to 9 cm, 1–3 mm thick, 0.3–0.48 mm thick when dry; upper surface pilose, becoming yellowish brown or pinkish buff upon drying; hymenophore surface usually smooth, without folds, becoming fuscous to black or cinnamon buff upon drying.

Internal features—Medulla present or absent, usually near hymenium if present; crystals present, usually scattered in hymenium; abhymenial hairs with a slightly swollen base, hyaline, thick-walled, with a narrow lumen, apical tips acute or obtuse, tufted, 100–150(–250) × 5–7.5 µm; hyphae with clamp connections, 1–5 µm in diam in KOH; basidia clavate, transversely 3-septate, with oil guttules, 65–85 × 4–5.5 µm, sterigmata rarely observed; cystidioles absent.

Spores—Basidiospores allantoid, hyaline, thin-walled, smooth, usually with one or two large guttules, IKI–, CB–, (14–)15–22(–23) × (4–)5–7(–8) µm, L = 19.03 µm, W = 6.16 µm, Q = 2.81–3.33 (n = 150/5).

Distribution—Europe.

Notes—*Auricularia auricula-judae* is characterized by thick basidiomata, dense and tufted hairs, and large basidia and basidiospores. The species was reported to have a wide distribution worldwide [12]. However, the so-called “*A. auricula-judae”* previously reported in the USA and Asia is confirmed as *A. americana* and *A. heimuer*, respectively, based on morphological characters and molecular analyses [5,20]. *Auricularia auricula-judae* can be distinguished from *A. americana* and *A. heimuer* by its larger basidia (55–71 × 4–5 µm in *A. americana*, 40–67 × 3–6.5 µm in *A. heimuer*) and basidiospores (14–16.5 × 4.5–5.4 µm in *A. americana*, 11–13 × 4–5 µm in *A. heimuer*). Phylogenetically, the species is distantly related to *A. americana* and *A. heimuer* and forms a single lineage with high support (Figure 1 and Figure 2).

Specimens examined—Czech Republic. Brno, on fallen trunk of *Carpinus*, 26 March 2014, M. Tomšovský, *MT 4/2014* (duplicate, BJFC 018550). Rousinov, on dead branch of *Sambucus*, 11 April 2014, M. Tomšovský, *MT 7/2014* (duplicate, BJFC 018561). Denmark. Nordjylland, Tranum, 16 September 1998, K. Jaederfeldt, *F 6772* (S). France. Lyon, Botanical Garden, on living tree of *Platanus*, 26 November 2012, Y.C. Dai, *Dai 13210* (BJFC 014074), on fallen trunk of *Robinia*, 3 November 2015, Y.C. Dai, *Dai 16353* (BJFC 020440). Orliénas, on fallen trunk of *Cornus*, 5 October 2013, B. Rivoire, Dai 13549 (duplicate, BJFC 015011); on angiosperm wood, 11 May 2014, B. Rivoire, *LYBR 5404* (duplicate, BJFC 018555). Germany. Munich, Botanical Garden, on *Ulmus*, November 1963, E. Albertshofer, *Albertshofer* (PDD 62991). Russia. Western Caucasus, Karachaevo-Cherkesia, Teberda State Biosphere Reserve, on fallen branch of *Fraxinus*, 22 August 2009, V. Malysheva, (LE 254030); on fallen trunk of *Euonymus*, 10 August 2009, V. Malysheva, (LE 254071). UK. Scotland, Fife, Kilconquhar Castle, 16 January 2014, *JT 04* (MFLU); Fife, Balbuthie Wood, on *Sambucus*, 5 Jan 2014, *JT 06* (MFLU); Humbie, Church Wood, on fallen branch of *Acer*, 5 October 2013, *JT 14* (MFLU).

(6) *Auricularia australiana* Y.C. Dai & F. Wu, sp. nov. Figure 10a and Figure 11.

MycoBank number: MB 840810.

Type—Australia. New South Wales, Gulaga, Dromedary Mountain, S 36°18′, E 150°02′, on dead wood, 10 April 2012, H. T. Van der, *HT 190* (MEL 2385783, holotype).

Etymology—*Australiana* (Lat.): refers to the species distributed in Australia.

Basidiomata—Gelatinous when rehydrated, fawn to reddish brown, solitary or caespitose, sessile or substipitate; pileus discoid or auriculate, margin entire, projecting up to 4 cm, 0.5–2 mm thick, 0.12–0.3 mm thick and dark grey when dry; upper surface pilose, sometimes with a few folds; hymenophore surface conspicuously porose-reticulate.

Internal features—Medulla absent; crystals absent; abhymenial hairs with a slightly swollen base or center, hyaline, thick-walled, with a wide or narrow septate lumen, apical tips acute or obtuse, single, 60–100 × 7–11 µm; hyphae with clamp connections and rare simple septa, 1–4 µm in diam in KOH; basidia clavate, transversely 3-septate, with oil guttules, 45–55 × 4.5–6 µm, sterigmata rarely observed; cystidioles absent.

Spores—Basidiospores allantoid, hyaline, thin-walled, smooth, usually with one to two large guttules, IKI–, CB–, (10.5–)11–12.8(–13) × (4–)4.4–5(–5.4) µm, L = 11.57 µm, W = 4.8 µm, Q = 2.37–2.46 (n = 60/2).

Distribution—Australia.

Notes—*Auricularia australiana* forms a distinct lineage and is closely related to *A. delicata* (Mont. ex Fr.) Henn., *A. sinodelicata* Y.C. Dai & F. Wu, *A. lateralis* Y.C. Dai & F. Wu and *A. conferta* Y.C. Dai & F. Wu in our phylogenies (Figure 1 and Figure 2), but it differs from *A. delicata* and *A. sinodelicata* by its wider hairs (5–8 µm in diam in *A. delicata* and 6–9 µm in diam in *A. sinodelicata*), and from *A. lateralis* by shorter hairs (95–250 µm long in *A. lateralis*). The species is easily confused with *A. conferta* also described from Australia, but *A. conferta* has dense and thick folds on the hymenophore surface and wider hairs (45–95 × 8–15 µm in *A. conferta*) usually with a wide and regular septate lumen.

Additional specimen (paratype) examined—Australia. Queensland, Cook, west of Mossman, Mt Lewis State Park, on wood, 10 April 2012, R. Vilgalys, *HN 3213* (MEL 2238923).

(7) *Auricularia brasiliana* Y.C. Dai & F. Wu Figure 3d and Figure 12.

Basidiomata—Gelatinous when fresh, greyish brown to vinaceous brown or pinkish buff, caespitose, resupinate to effused-reflexed; pileus free lobed, margin undulate, projecting up to 9 cm, 1.5–3 mm thick, 0.2–0.4 mm thick when dry; upper surface tomentose, distinctly and concentrically zoned with wide whitish zones and thin black bands throughout pileal surface, becoming clay pink upon drying; hymenophore surface venose with folds, becoming dark greyish blue to blackish blue upon drying.

Internal features—Medulla absent; crystals absent; abhymenial hairs with a slightly swollen base, hyaline, thick-walled, with a narrow lumen, apical tips acute or obtuse, tufted, 1000–1500 × 2–3.5 µm; hyphae rarely with clamp connections, obviously inflated with a lumen in KOH, up to 13 µm in diam; basidia clavate, transversely 3-septate, with oil guttules, 30–47 × 3–5 µm, sterigmata rarely observed; cystidioles absent.

Spores—Basidiospores allantoid, hyaline, thin-walled, smooth, usually with one or two large guttules, IKI–, CB–, 11.5–12 × 4.5 µm, L = 11.8 µm, W = 4.5 µm, Q = 2.62 (n = 2/1).

Distribution—Brazil.

Notes—*Auricularia brasiliana* was described from Brazil [25], and is macro-morphologically similar to *A. asiatica*, which, however, lacks whitish zones and thin black bands at margin of pileal surface [26]. These two species form two distinct lineages with strong support in our phylogenies (Figure 1 and Figure 2). The morphological description of *Auricularia brasiliana* taken from Wu et al. [25].

Specimens examined—Brazil. Alagoas, RPPN São Pedro, 12 November 2016, R.L. M. Alvarenga, *RLMA 344* (URM 93413). Bahia, Santa Teresinha, Serra da Jibóia, 24 January 2011, T. B. Gibertoni, *TGB 24* (URM 83482). Ceará, Crato, Floresta Nacional do Araripe, 16 May 2012, C.R.S. Lira, *CRSL 886* (URM 84563). Maranhão, Cidelândia, Reserva Extrativista do Ciriaco, Povoado do Ciriaco, 30 July 2013, L. Araujo-Neta, *AN-MA 42* (URM 85567, holotype). Mato Grosso, Terra Indígena Sete de Setembro—Aldeia Apoena Meirelles, Rondolândia, 31 May 2009, *Gomes-Silva* et al. *s/n* (URM 80967). Pernambuco, Jaqueira, RPPN Frei Caneca, Mata do Quengo, 2012, R.S. Chikowski, *RSC 359* (URM), *RSC 487* (URM). Piauí, Caracol, Parque Nacional da Serra das Confusões, 1 January 2011, A.C. Gomes-Silva, *ACGS 187* (URM 83468). Rondônia, Terra Indígena Sete de Setembro, Aldeia Lapetanha, Cacoal, 04 June 2009, *Gomes-Silva* et al. *s/n* (URM 80933).

(8) *Auricularia camposii* Y.C. Dai & F. Wu, sp. nov. Figure 13.

MycoBank number: MB 825097.

Type—Brazil. Paré, Maracanã, Ilha de Maiandeua, near the village of Algodoal, S 0°59′, W 47°58′, on *Rhizophora mangle*, July 1998, E.L. de Campos; A. Luz s.n. (URM 76905, holotype; BJFC018567, isotype).

Etymology—*Camposii* (Lat.): refers to the late Ezequias Lopes de Campos, mycologist and collector of the samples of the new species.

Basidiomata—Gelatinous when rehydrated, orange-brown to reddish brown, solitary or caespitose, sessile or substipitate; pileus discoid or auriculate, margin entire, projecting up to 5 cm, 1–3 mm thick, 0.2–0.45 mm thick when dry; upper surface densely tomentose, becoming greyish brown upon drying; hymenophore surface usually smooth, without folds, becoming bluish grey to greyish blue upon drying.

Internal features—Medulla present in the middle of the cross-section; crystals present, usually scattered throughout the cross-section; abhymenial hairs with a slightly swollen base, hyaline, thick-walled, with a wide or narrow lumen, apical tips acute or obtuse, tufted, 120–250 × 6–7 µm; hyphae with clamp connections and simple septa, 0.5–4 µm in diam in KOH; basidia clavate, transversely 3-septate, with oil guttules, 60–75 × 4–5.5 µm, sterigmata rarely observed; cystidioles absent.

Spores—Basidiospores reniform, hyaline, thin-walled, 12.5–15 × 5–6 µm.

Distribution—Brazil.

Notes—*Auricularia camposii* is characterized by a densely tomentose upper surface, with an obvious medulla in the middle of the cross-section. It is similar to *A. cornea* in macro-morphology, but has slightly shorter abhymenial hairs (120–250 µm vs. 180–425 µm). In the phylogeny, *A. camposii* is very distantly related to *A. cornea* and samples of these two species form two lineages (Figure 1 and Figure 2). *Auricularia camposii* is relatively related to *A. nigricans* (Figure 1 and Figure 2) which, however, has distinctly longer abhymenial hairs (300–600 × 7–9 µm vs. 120–250 × 6–7 µm). The type and paratype are good enough and ITS and nLSU sequences are available from them. Therefore, we venture to publish a description for *A. camposii* even though we know it is inadvisable to do so based on a few basidiospores.

Additional specimen (paratype) examined—Brazil. Paraíba, Areia, Parque Estadual Mata do Pau Ferro, 2010, C.R.S. Lira, *CL8* (URM 83464; duplicate, BJFC 018581).

(9) *Auricularia conferta* Y.C. Dai & F. Wu, sp. nov. Figure 10b and Figure 14.

MycoBank number: MB 840811.

Type—Australia. Queensland, Cairns, Crater Lakes National Park, S 17°14′, E 145°37′, 17 May 2018, Y.C. Dai, *Dai 18825* (BJFC 027293, holotype).

Etymology—*Conferta* (Lat.): refers to the species having dense folds on the hymenophore surface.

Basidiomata—Gelatinous when fresh, fawn to reddish brown, solitary or caespitose, sessile; pileus auriculate, margin entire, projecting up to 6 cm, 1–3 mm thick, 0.3–0.5 mm thick and reddish brown to fuscous when dry; upper surface pilose, usually with a few folds; hymenophore surface conspicuously and densely porose-reticulate.

Internal features—Medulla present indistinctly near the hymenium; crystals absent; abhymenial hairs with a slightly swollen base or center, hyaline, thick-walled, usually with a wide and regular septate lumen, apical tips acute or obtuse, single, 45–95 × 8–15 µm; hyphae with clamp connections, rarely simple septa, 1–4.5 µm in diam in KOH; basidia clavate, transversely 3-septate, with oil guttules, 44–58 × 4.5–6.5 µm, sterigmata rarely observed; cystidioles absent.

Spores—Basidiospores allantoid, hyaline, thin-walled, smooth, usually with one to two large guttules, IKI–, CB–, (10.8–)11–13(–13.5) × (4–)4.5–5.2(–5.5) µm, L = 11.91 µm, W = 4.9 µm, Q = 2.43 (n = 30/1).

Distribution—Australia.

Notes—*Auricularia conferta* is macro-morphologically similar to *A. sinodelicata* because *A. sinodelicata* sometimes also has dense and thick folds on the hymenophore surface. However, *A. conferta* has larger basidia (44–58 × 4.5–6.5 µm vs. 30–45 × 4–5.5 µm in *A. sinodelicata)* and mean basidiospores length (L = 11.91 µm vs. L = 10.87 µm in *A. sinodelicata*).

Additional specimen (paratype) examined—Australia. Queensland, Cairns, Crater Lakes National Park, 17 May 2018, Y.C. Dai, *Dai 18826* (BJFC 027294).

(10) *Auricularia cornea* Ehrenb. Figure 15a–h and Figure 16.

≡*Exidia cornea* (Ehrenb.) Fr., Syst. mycol. (Lundae) 2(1): 222, 1822.

=*Auricularia polytricha* var. *argentea* D.Z. Zhao & Chao J. Wang, Acta Mycol. Sin., 10(2): 108, 1991.

=*Auricularia reticulata* L.J. Li, Acta Mycol. Sin. 4(3): 151, 1985.

=*Auricularia leucochroma* Kobayasi, Bull. cent. nat. Mus. Manchoukuo 4: 26, 1942.

Basidiomata—Gelatinous when fresh, fawn to reddish brown or buff to white, solitary or caespitose, sessile or substipitate; pileus discoid or auriculate, sometimes with lobed margin, projecting up to 9 cm, 1–2 mm thick, 0.2–0.48 mm thick when dry; upper surface densely pilose, becoming light vinaceous grey or cream upon drying; hymenophore surface usually smooth, without folds, becoming mouse-grey to black or buff-yellow upon drying.

Internal features—Medulla present in the middle of the cross-section or near the abhymenium; crystals present, usually scattered in the hymenium; abhymenial hairs with a slightly swollen base, hyaline, thick-walled, with a wide or narrow lumen, apical tips acute or obtuse, tufted, 180–425 × 6–9 µm; hyphae with clamp connections and simple septa, 0.5–3 µm in diam in KOH; basidia clavate, transversely 3-septate, with oil guttules, 60–75 × 4–6 µm, sterigmata sometimes observed; cystidioles absent.

Spores—Basidiospores allantoid, hyaline, thin-walled, smooth, usually with one to three large guttules, IKI–, CB–, (13–)13.8–16.5(–17.6) × (4.2–)4.5–6(–6.4) µm, L = 14.89 µm, W = 5.39 µm, Q = 2.42–3.06 (n = 270/9).

Distribution—Africa, North and South America, Asia and Europe.

Notes—*Auricularia cornea* is characterized macroscopically by the variability in color of fresh basidiomata, dense hairs on the upper surface, and microscopically by the presence of an obvious medulla. It has a wide distribution, being recorded almost all over the world, and is very common in both natural and managed forests of subtropical and tropical areas. It is cultivated in China as “Maomuer” and was first reported as *A. polytricha* by Patouillard and Olivier [19], a name used in almost all Chinese reports until 2015 and in other Asian and Pacific areas [12,17,42,43,44,45,46,47,48,49,50,51]. However, *A. polytricha* is a synonym of *A. nigricans*, which is distributed throughout America. Zhao & Wang described *A. polytricha* var. *argentea* D.Z. Zhao & Chao J. Wang based on its white basidiomata [52]. According to our study, *A. polytricha* var. *argentea* [basidia measuring 50–65 × 4–5.5 µm; basidiospores measuring (12.7–)13–15(–15.2) × (4.7–)4.9–5.4(–5.6) µm, L = 13.91 µm, W = 5.02 µm, Q = 2.77 (n = 30/1), from type material] is a synonym of *A. cornea.* In fact, we found that *A. cornea* produced both brown and white basidiomata in nature (Figure 15h). In our phylogenies (Figure 1 and Figure 2), all samples of *A. cornea,* wild *A. polytricha* var. *argentea* (Dai 14876) and cultivated *A. polytricha* var. *argentea* (Wu 07) form a lineage although with some variations. The white basidiomata of *A. cornea* is called “Yumuer” and is now cultivated in China (Figure 15g).

In addition, *A. reticulata* L.J. Li was reported as a new species from China due to its reticulate sterile surface [53]. However, *A. cornea* sometimes has a reticulate sterile surface, too, e.g., Cui 7517 (Figure 15f). We examined the type of *A. reticulata*, and found it has an obvious medulla and the abhymenial hairs are almost the same as those of *A. cornea*. We did not find basidiospores from the type, but the spores were reported as 14.5 × 5–6 µm in the original description [53], which fit the dimensions of those of *A. cornea.* For these reasons, we treat *A. reticulata* as a synonym of *A. cornea*. Another species, *A. leucochroma,* was also described from Asia [16], and treated as a synonym of *A. nigricans* [20]. We rejected this opinion because we examined the type material of *A. leucochroma,* and treated it as a synonym of *A. cornea*. Because of the variation of morphology Figure 15a–h) and of the molecular data (Figure 1 and Figure 2), in the present study all these samples with variable morphology are treated as *A. cornea* for the time being.

Specimens examined—Benin. On fallen angiosperm trunk, 2 July 2015, Y.C. Dai, *Dai 15447* (BJFC 019552), *Dai 15448* (BJFC 019553). Brazil. Acre, Sena Madureira, Floresta Nacional de São Francisco, 28 January 2016, A Meiras-Ottoni, *AMO 573* (URM 93416). Maranhão, Reserva Extrativista do Ciriaco, Cidelândia, 26 July 2013, L. Araujo-Neta, (URM 85561), L. Araujo-Neta, (URM 85563); Ceará, Tianguá, Trilha do Riacho, 2012, C.R.S. Lira, *Lira 663* (URM 83696); Pernambuco, Tamandaré, Reserva Biológica de Saltinho, 11 March 2016, R.L.M. Alvarenga, *RLMA 310* (URM 93419). Paraíba, Areia, Parque Estadual Mata do Pau Ferro, 16 April 2012, C.R.S. Lira, *CRSL* 663 (URM 83696). São Paulo, São Paulo, on dead angiosperm tree, 18 June 2015, S.s.n. Visnadi, *AG 6* (BJFC 020594), 7 June 2015, A.M. Gugliotta, Y. Gafforov, *AG 1546* (BJFC 020595), A.M. Gugliotta, Y. Gafforov, *AG 1547* (BJFC 020596); 7 August 2015, A.M. Gugliotta, Y. Gafforov, *AG 1559* (BJFC 020601), A.M. Gugliotta, Y. Gafforov, *AG 1560* (BJFC 020601); 16 July 2015, A.M. Gugliotta, Y. Gafforov, *AG 1554* (BJFC 020600); Jardim Botânico de São Paulo, April 2017, R.L.M. Alvarenga, *RLMA 467* (URM 93423), *RLMA 464* (URM 93422). China. Fujian Province, Nanjing County, Huboliao Nature Reserve, on fallen angiosperm trunk, 26 October 2013, B.K. Cui, *Cui 11333* (BJFC 015449), Xiamen, Botanical Garden, on rotten angiosperm wood, 27 October 2013, B.K. Cui, *Cui 11346* (BJFC 015462). Guangdong Province, Shixing County, Chebaling Nature Reserve, on rotten wood of *Idesia polycarpa*, 17 September 2009, B.K. Cui, *Cui 7517* (BJFC 006005). Guangxi Autonomous Region, Nanning, Campus of Guangxi University, on fallen angiosperm trunk, 9 December 2013, Y.C. Dai, *Dai 13640* (BJFC 015103), *Dai 13641* (BJFC 015104), Qingxiu Mountain, on fallen angiosperm branch, 7 June 2015, Y.C. Dai, *Dai 15368* (BJFC 019479); Shangsi County, Shiwandashan Nature Reserve, on fallen angiosperm trunk, 6 June 2015, Y.C. Dai, *Dai 15336* (BJFC 019447), on fallen angiosperm branch, 6 June 2015, Y.C. Dai, *Dai 15338* (BJFC 019449), *Dai 15346* (BJFC 019457). Guizhou Province, Suiyang County, Kuankuoshui Nature Reserve, on rotten wood of *Fagus*, 25 November 2014, Y.C. Dai, *Dai 15016* (BJFC 018129), 26 November 2014, Y.C. Dai, *Dai 15070* (BJFC 018182). Hainan Province, Changjiang County, Bawangling National Forest Park, on rotten angiosperm wood, 16 June 2014, Y.C. Dai, *Dai 13679* (BJFC 017419); Danzhou County, Botanical Garden, on fallen angiosperm trunk, 19 October 2013, Y.C. Dai, *Dai 13547* (BJFC 015009), *Dai 13548* (BJFC 015010), 15 June 2014, Y.C. Dai, *Dai 13659* (BJFC 017398); Ledong County, Jianfengling Nature Reserve, on rotten angiosperm wood, 11 May 2009, Y.C. Dai, *Dai 10845* (BJFC 005087). Hebei Province, Laishui County, Yesanpo Grand Canyon, on dead *Acer*, 18 November 2014, Y.C. Dai, *Dai 14876* (BJFC 017989). Henan Province, Neixiang County, Baotianman Nature Reserve, on fallen angiosperm trunk, 21 June 2015, Y.C. Dai, *Dai 15379* (BJFC 019490). Jiangxi Province, Fuzhou, on rotten wood of *Albizia*, 1 July 2013, Y.C. Dai, *Dai 13449* (BJFC 014910). Jiangsu Province, Taizhou, on rotten wood of *Salix*, 30 October 2013, Y.C. Dai, *Dai 13621* (BJFC 015084). Jilin Province, Antu County, Changbaishan Nature Reserve, Cultivated, 9 August 2017, F. Wu, *Wu 07* (BJFC 026013). Shanxi Province, Mei County, Taibai Mounty, Honghegu Forest Park, on fallen angiosperm trunk, 9 September 2013, B.K. Cui, *Cui 11162* (BJFC 015277). Yunnan Province, Jinghong, Xishuangbanna Tropical Botanical Garden, on fallen angiosperm trunk, 21 July 2014, Y.C. Dai, *Dai 13750* (BJFC 017481), *Dai 13753* (BJFC 017484), *Dai 13754* (BJFC 017485); Puer, Taiyanghe National Forest Park, on fallen angiosperm trunk, 8 July 2013, B.K. Cui, *Cui 11026* (BJFC 015142); Yingjiang County, Tongbiguan Reserve, on fallen trunk of *Prunus*, 30 October 2012, Y.C. Dai, *Dai 13124* (BJFC 013341); Yongde County, Daxueshan Nature Reserve, on rotten angiosperm stump, 27 August 2015, Y.C. Dai, *Dai 15668* (BJFC 019772). Germany. On fallen angiosperm wood, 7 August 2015, Y. Gofforov, *YG-Dr1* (duplicate, BJFC 020603). Ghana. Bunso Arboretum Garden, 8 November 2016, Y.C. Dai, *Dai 17352* (BJFC 024110). Indonesia. Papua, on fallen angiosperm trunk, 2 March 2007, O. Miettinen, *OM 11587* (H). Japan. Bonin Islands, 1936, Okabe, (F193009, type of *A. leucochroma*). Singapore. Bukit Timah Nature Reserve, on fallen angiosperm trunk, 20 July 2017, Y.C. Dai, *Dai 17865* (BJFC 025397). South Africa. Durban, on fallen angiosperm trunk, 1 October 2011, Y.C. Dai, *Dai 12579* (BJFC 012167), *Dai 12583* (BJFC 012169), *Dai 12592* (BJFC 012178); on fallen trunk of *Ficus*, 1 October 2011, *Dai 12587* (BJFC 012173). Sri Lanka. Mitirigala Nissarana Vanaga Forest, on fallen angiosperm trunk, 4 March 2019, Y.C. Dai, *Dai 19650* (BJFC 031327). Tanzania. Southern Highland Province, on fallen angiosperm trunk, 16 December 1990, *TS 649* (H), *TS 655* (H). USA. Puerto Rico, on fallen angiosperm trunk, 24 January 2014, O. Miettinen, *OM 18106* (H). Vietnam. Ho Chi Minh City, Saigon Zoo & Botanical Garden, on dead angiosperm tree, 13 October 2017, Y.C. Dai, *Dai 18315* (BJFC 025838).

(11) *Auricularia delicata* (Mont. ex Fr.) Henn. Figure 10c and Figure 17.

≡*Laschia delicata* Mont. ex Fr., Linnaea 5: 533, 1830.

Epitype (designated here):—Cameroon. Nsimalen Airport, near Yaounde, on fallen angiosperm trunk, 27 July 1996, D. Mossebo, *P 14* (K 43873, epitype).

Basidiomata—Gelatinous when rehydrated, dark brownish to vinaceous brown, solitary or caespitose, sessile or substipitate; pileus orbricular, sometimes with lobed margin, projecting up to 8 cm, 1.5–2.5 mm thick, 0.18–0.28 mm thick and dark reddish brown to fuscous when dry; upper surface pilose, sometimes with several folds; hymenophore surface conspicuously porose-reticulate.

Internal features—Medulla absent or indistinctly present near the hymenium; crystals absent; abhymenial hairs with a slightly swollen base, hyaline, thick-walled, with a wide or narrow septate lumen, apical tips acute or obtuse, single, 60–100 × 5–8 µm; hyphae with clamp connections, 0.5–6 µm in diam in KOH; basidia clavate, transversely 3-septate, with oil guttules, 48–65 × 4–6 µm, sterigmata rarely observed; cystidioles absent.

Spores—Basidiospores allantoid, hyaline, thin-walled, smooth, usually with one large guttule, IKI–, CB–, (9.5–)10–11.5(–11.8) × (4.3–)4.5–5(–5.5) µm, L = 10.83 µm, W = 4.93 µm, Q = 2.08 (n = 30/1).

Distribution—Western Africa.

Notes—*Auricularia delicata* was originally described from Equatorial Guinea in West Africa and it was reported throughout tropical America, Africa, Asia, Australia and South Pacific [12,17]. *Auricularia delicata* was found to be a species complex, and two new species belonging to the complex, *A. scissa* and *A. subglabra*, were described because they have schizomedulla, which is distinctly different from the original description of the medulla of *A. delicata* by Looney et al. [20]. In addition, samples from Australia and Mexico form two other lineages in the phylogeny, which were treated, respectively, as *A. delicata* clade I and *A. delicata* clade II by Looney et al. [20], because of the shortage of samples from the type locality of *A. delicata*.

In the present study, one collection from Cameroon (K 43873) was morphologically examined and phylogenetically analyzed, and its morphology fits the description of *A. delicata.* In addition, Cameroon is very close to Equatorial Guinea, so we consider the specimen (K 43873) represents the real *A. delicata*, and K 43873 is treated as epitype of *A. delicata*. The *A. delicata* clade I and *A. delicata* clade II are, respectively, treated as *A. australiana* and *A. tremellosa* (Fr.) Pat. in our study (Figure 1). *Auricularia tremellosa* was originally described from Mexico [53], and more samples were morphological and phylogenetically studied. These samples form a distinct lineage (Figure 1 and Figure 2) and their morphology fits the description of *A. tremellosa*. 

*Auricularia delicata* is characterized by a conspicuously porose-reticulate hymenophore, short hairs and relatively small basidiospores. It is closely related to *A. sinodelicata* Y.C. Dai & F. Wu, but *A. sinodelicata* has smaller basidia (30–45 µm × 4–5.5 µm), slightly shorter hairs (30–80 µm × 6–9 µm) and a distribution in Asia.

(12) *Auricularia eburnea* L.J. Li & B. Liu Figure 15i and Figure 18.

Basidiomata—Gelatinous when rehydrated, cream to honey-yellow, solitary or caespitose, sessile or substipitate; pileus discoid or auriculate, sometimes with lobed margin, projecting up to 4.8 cm, 0.6–0.8 mm thick, 0.2–0.26 mm thick when dry; upper surface densely pilose, becoming cinnamon-buff upon drying; hymenophore surface usually with several folds, becoming honey-yellow upon drying.

Internal features—Medulla present in the middle of the cross-section or near the abhymenium; crystals absent; abhymenial hairs with a slightly swollen base, hyaline, thick-walled, with a wide or narrow septate lumen, apical tips acute or obtuse, tufted, 280–380 × 5–6.5 µm; hyphae with clamp connections, 0.5–2.5 µm in diam in KOH; basidia clavate, transversely 3-septate, with oil guttules, 50–65 × 4–5 µm, sterigmata rarely observed; cystidioles absent.

Spores—Basidiospores not observed in the type, but reported as (14.5–)15.6–18 × 5.5–7.5 µm by Li and Liu [54].

Distribution—China.

Notes—*Auricularia eburnea* is characterized by cream to honey-yellow fresh basidiomata. According to our observations of the type, the species is macro-morphologically similar to *A. cornea*, but *A. eburnea* has obviously larger basidiospores than those in *A. cornea* (15.6–18 × 5.5–7.5 µm vs 13.8–16.5 × 4.5–6 µm). We failed to extract DNA, because the type material is in poor condition. For the time being we accept *A. eburnea* as an independent species based on its original description. 

Specimen examined—China. Hainan Province, Sansha, East of Xisha Islands, on rotten angiosperm wood, July 1983, L.J. Li, B. Liu, HPNHW 1449 (BJM, holotype).

(13) *Auricularia eminii* Henn. Figure 15j and Figure 19.

=*Auricularia squamosa* Pat. & Har., Bull. Soc. mycol. Fr. 9: 210, 1893.

Basidiomata—Gelatinous when rehydrated, reddish brown to vinaceous brown, solitary or caespitose, sessile; pileus discoid or auriculate, margin entire, projecting up to 12 cm, 0.6–0.9 mm thick, 0.2–0.4 mm thick when dry; upper surface densely pilose, becoming vinaceous grey upon drying; hymenophore surface usually smooth, without folds, becoming fuscous upon drying.

Internal features—Medulla present in the middle of the cross-section or near the abhymenium; crystals absent; abhymenial hairs with a slightly swollen base, hyaline, thick-walled, with narrow lumen, apical tips acute or obtuse, tufted, up to 1 cm long and 4–6 µm broad; hyphae with clamp connections and simple septa, 0.5–4.5 µm in diam in KOH; basidia clavate, transversely 3-septate, with oil guttules, 40–60 × 4–6 µm, sterigmata rarely observed; cystidioles absent.

Spores—Basidiospores not observed in the studied specimens, but reported as 12–14 × 4–5 µm by Lowy [17].

Distribution—Africa.

Notes—*Auricularia eminii* is characterized by the longest hairs on the upper surface in the genus and is distributed in Africa only. Lowy indicated that *A. squamosa* was a synonym of *A. eminii* [17]. In the present study, we confirm this synonym because the morphological differences between the two type specimens are too small to justify the segregation of these two species.

Specimens examined—Congo. 1893, F 19526 (type of *Auricularia eminii*, S), 1893, FH (type of *Auricularia squamosa*). Ethiopia. Bonga, 1984, D. Abate, (K 27093). Tanzania. Flora, 1966, R.F. Cain, (K 89904).

(14) *Auricularia fibrillifera* Kobayasi Figure 20a and Figure 21.

Basidiomata—Soft gelatinous when fresh, fawn to reddish brown, solitary or caespitose, sessile or substipitate; pileus discoid or auriculate, margin entire, projecting up to 6 cm, 0.35–0.5 mm thick, 0.04–0.2 mm thick and fawn to fuscous when dry; upper surface scantly pilose; hymenophore surface usually with folds.

Internal features—Medulla indistinctly present in the middle of the cross-section; crystals present, very small, scattered throughout the cross-section; abhymenial hairs with an obviously swollen base, hyaline, thick-walled, with a wide septate lumen, apical tips acute or obtuse, single, 60–100 × 10–20 µm; hyphae with clamp connections, 1–3 µm in diam in KOH; basidia clavate, transversely 3-septate, with oil guttules, 41–57 × 4–6 µm, sterigmata rarely observed; cystidioles absent.

Spores—Basidiospores allantoid, hyaline, thin-walled, smooth, usually with one or two large guttules, IKI–, CB–, (10.7–)11–14(–14.5) × 4–5(–5.6) µm, L = 12.39 µm, W = 4.47 µm, Q = 2.45–3.03 (n = 150/5).

Distribution—Africa, Asia and Oceania.

Notes—*Auricularia fibrillifera* was originally described from Papua New Guinea [55], and the type specimen (F 234519) was studied. *Auricularia thailandica* Bandara & K.D. Hyde was recently described from Thailand by Bandara et al. [24], and it is very close to *A. fibrillifera* both in morphology and phylogeny. Both species occur in Southeast Asia and it is very difficult to distinguish them by morphology, especially the dry basidiomata. However, these two species clustered into two distinct lineages in our phylogeny based on the concatenated ITS+nLSU dataset (Figure 1); furthermore, the fresh basidiomata of *A. fibrillifera* are softer than those of *A. thailandica*. Therefore, we treat *A. fibrillifera* and *A. thailandica* as two independent species in the present study.

Specimens examined—China. Guizhou Province, Anshun County, Huangguoshu Nature Reserve, on fallen angiosperm trunk, 17 July 2014, Y.C. Dai, *Dai 13744* (BJFC 017475). Hainan Province, Changjiang County, Bawangling National Forest Park, on fallen angiosperm trunk, 9 May 2009, B.K. Cui, *Cui 6318* (BJFC 004174), *Cui 6501* (BJFC 004354), *Cui 6504* (BJFC 004357); Qiongzhong County, Diaoluoshan Forest Park, on fallen angiosperm trunk, 13 June 2014, Y.C. Dai, *Dai 13596A* (BJFC 017335), *Dai 13597A* (BJFC 017336), *Dai 13598A* (BJFC 017337); Wanning County, on fallen angiosperm trunk, 15 May 2009, B.K. Cui, *Cui 6704* (BJFC 004558). Papua New Guinea. 28 December 1969, *F 234519* (TNS, holotype). Zambia. Livingstone, on rotten angiosperm wood, 1 January 2018, Y.C. Dai, *Dai 18486* (BJFC 026006). 

(15) *Auricularia fuscosuccinea* (Mont.) Henn. Figure 20b and Figure 22.

≡*Exidia fuscosuccinea* Mont. [as ‘*fusco-succinea*’] Annls Sci. Nat., Bot., sér. 2 17: 125, 1842.

Basidiomata—Gelatinous when fresh, fawn to cinnamon brown, solitary or caespitose, sessile or substipitate; pileus discoid or auriculate, margin entire, projecting up to 6.5 cm, 1–3 mm thick, 0.15–0.3 mm thick and cinnamon brown to vinaceous brown when dry; upper surface pilose, sometimes with folds; hymenophore surface smooth, sometimes with folds.

Internal features—Medulla present near the abhymenium; crystals present, scattered in the hymenium; abhymenial hairs with a slightly swollen base, hyaline, thick-walled, with a wide or narrow lumen, sometimes with a septate lumen in acute or obtuse apical tips, single or tufted, 75–165 × 5–8 µm; hyphae with clamp connections and simple septa, 1–6 µm in diam in KOH; basidia clavate, transversely 3-septate, with oil guttules, 42–66 × 4–7 µm, sterigmata rarely observed; cystidioles absent.

Spores—Basidiospores allantoid, hyaline, thin-walled, smooth, usually with one or two large guttules, IKI–, CB–, (11.8–)12–14(–14.5) × (4–)4.3–5.2(–5.5) µm, L = 13 µm, W = 4.92 µm, Q = 2.62–2.66 (n = 60/2).

Distribution—Tropical and subtropical America.

Notes—*Auricularia fuscosuccinea* is characterized by macroscopically cinnamon brown basidiomata when dry and microscopically by a medulla distinctly present near the abhymenium. It is similar to *A. fibrillifera* and *A. thailandica* but differs by having slightly thicker basidiomata and hyphae with simple septa. In the phylogeny, *A. fuscosuccinea* is distantly related to *A. fibrillifera* and *A. thailandica* (Figure 1 and Figure 2). The species was previously reported in China [44]. Vouchers of Chinese materials were studied, and they are actually *A. fibrillifera* or *A. thailandica.* Because the three species share similar morphology, *A. fuscosuccinea* is temporarily considered as a species complex even if the phylogenetic relations are distant (Figure 1 and Figure 2).

Specimens examined—Brazil. Acre, Sena Madureira, Floresta Nacional de São Francisco, on dead angiosperm tree, 28 January 2016, A. Meiras-Ottoni, *AMO 538* (URM 93429), Floresta Nacional do Macauã, 29 January 2016, A. Meiras-Ottoni, *AMO 623* (URM 93435), Reserva Extrativista do Cazumbá-Iracema, 30 January 2016, A. Meiras-Ottoni, *AMO 672* (URM 93430). Goiás, Floresta Nacional de Silvânia, 12 January 2016, R.L.M. Alvarenga, *RLMA 395* (URM 93434); Paraíba, Areia, Parque Estadual Mata do Pau Ferro, 2013, C.R.S. Lira, *CRSL* 563 (URM 84766, duplicate, BJFC 018583); Pernambuco, Gravatá, RPPN Serra do Contente, on dead angiosperm tree, 18 May 2017, Y.C. Dai, *Dai 17451* (BJFC 024982). São Paulo, São Paulo, on dead angiosperm tree, 7 June 2015, A.M. Gugliotta, Y. Gafforov, *AG 1548* (BJFC 020597), A.M. Gugliotta, Y. Gafforov, *AG 1549* (BJFC 020598), A.M. Gugliotta, Y. Gafforov, *AG 1550* (BJFC020599), 2013, 12 May 2017, Y.C. Dai, *Dai 17406* (BJFC 024940), *Dai 17422* (BJFC 024953); Paraíba, Areia, Parque Estadual Mata do Pau Ferro, 2013, C.R.S. Lira, *CRSL* 563 (URM 84766, duplicate, BJFC 018583); Rondônia, Itapuã do Oeste, Floresta Nacional do Jamari, 06 February 2016, A. Meiras-Ottoni, *AMO 699* (URM 93431). **USA**. Florida, on rotten angiosperm wood, 24 November 2013, O. Miettinen, *OM 17909* (H). Louisiana, on fallen angiosperm trunk, 1990, T.J. Volk, *FP-102573-SP* (CFMR). Puerto Rico, on fallen angiosperm trunk, 1993, D.J. Lodge, H.H. Burdsall, Jr., & M. Boyd, *PR-1378* (CFMR).

(16) *Auricularia hainanensis* L.J. Li Figure 5d and Figure 23.

Basidiomata—Gelatinous when rehydrated, fuscous to vinaceous brown, solitary, sessile or substipitate; pileus auriculate, margin entire, projecting up to 3.5 cm, 2–3.1 mm thick, 0.6–0.7 mm thick and mouse-grey to black when dry; upper surface scantly pilose, sometimes with folds; hymenophore surface with obvious folds.

Internal features—Medulla present in the middle of the cross-section; crystals absent; abhymenial hairs with a slightly swollen base, hyaline, thick-walled, with a narrow lumen, sometimes septate, apical tips acute or obtuse, single, 30–80 × 4–5 µm; hyphae with clamp connections, 0.5–5 µm in diam in KOH; basidia clavate, transversely 3-septate, with oil guttules, 70–80 × 3–4 µm, sterigmata rarely observed; cystidioles absent.

Spores—Basidiospores not observed in the type, but reported as 7.5–8.5 × 3–4 µm by Li [18].

Distribution—China.

Notes—*Auricularia hainanensis* is characterized by very thick basidiomata when dried and very small basidiospores. According to the original description and our observation of the type, the species is morphologically similar to *A. minor* Kobayasi, but that species has thinner basidiomata (0.1–0.2 mm vs. 0.6–0.7 mm) and hymenophore surface without distinct folds. Because the type materials of both *A. hainanensis* and *A. minor* are in poor condition, and DNA extraction from the types failed, their phylogenetic relations remain uncertain.

Specimen examined—China. Hainan Province, Qiongzhong County, Limu Mountain, on fallen angiosperm trunk, 12 April 1977, L.J. Li, *HPNHM 103* (BJM, holotype).

(17) *Auricularia heimuer* F. Wu, B.K. Cui, Y.C. Dai Figure 5e,f and Figure 24.

Basidiomata—Gelatinous when fresh, fawn to reddish brown, solitary or caespitose, sessile or substipitate; pileus auriculate or petaloid, margin entire, projecting up to 12 cm, 0.8–1.5 mm thick, 0.04–0.24 mm thick and greyish brown to vinaceous brown when dry; upper surface pilose, sometimes with folds; hymenophore surface smooth, sometimes with shallow folds.

Internal features—Medulla present in the middle of the cross-section; crystals absent; abhymenial hairs with a slightly swollen base, hyaline, thick-walled, with a narrow lumen, apical tips acute or obtuse, single or tufted, 50–150 × 4–6.5 µm; hyphae with clamp connections, 0.5–3 µm in diam in KOH; basidia clavate, transversely 3-septate, with oil guttules, 40–67 × 3–6.5 µm, sterigmata frequently observed; cystidioles absent.

Spores—Basidiospores allantoid, hyaline, thin-walled, smooth, usually with one or two large guttules, IKI–, CB–, 11–13(–13.1) × 4–5(–5.1) µm, L = 12 µm, W = 4.71 µm, Q = 2.43–2.74 (n = 120/4).

Distribution—Temperate Asia.

Notes—Morphologically *A. heimuer* is similar to *A. villosula* Malysheva, but the latter has larger basidiospores (13–15.5 × 5–6.1 µm). *Auricularia heimuer* has a wide distribution in temperate Asia and grows mostly on *Quercus,* but basidiomata from the wild are uncommon. Previously the Asian *A. heimuer* was considered the same species as the European *A. auricula-judae* [56,57]. The cultivated and the wild *A. heimuer* (Figure 5e,f) were different from *A*. *auricula-judae* in both morphology (refer to the notes on *A. auricula-judae*) and phylogeny, in which they clustered into two different lineages, each one with high support (Figure 2). The morphological description of *A. heimuer* taken fromWu et al. [5].

Specimens examined—China. Heilongjiang Province, Donglin County, on fallen trunk of *Quercus*, 4 August 2014, Y.C. Dai, *Dai 13785* (BJFC 017516), *Dai 13788* (BJFC 017519); Hailin County, on fallen trunk of *Quercus*, 2 August 2014, Y.C. Dai, *Dai 13765* (BJFC 017496, holotype); Mudanjiang, Wenchun County, on fallen trunk of *Quercus*, 12 August 2009, B.K. Cui, *Cui 7105* (BJFC 005592); Yabuli County, Hongxing Farm, on fallen trunk of *Quercus*, 12 September 2013, Y.C. Dai, *Dai 13503* (BJFC 014965). Jilin Province, Dunhua County, Huangnihe, cultivation, 8 August 1997, Y.C. Dai, *Dai 2291* (BJFC 019121); Hunchun, Hadamen Village, on fallen trunk of *Quercus*, 7 August 2009, B.K. Cui, *Cui 7101* (BJFC 005588); Taiyang Mountain, on fallen trunk of *Quercus*, 25 July 2013, Y.C. Dai, *Dai 13647* (BJFC 015110). Shanxi Province, Lingchuan County, on fallen angiosperm trunk, 9 November 2009, B.K. Cui, *Cui 8644* (BJFC 007133). Tianjin, Ji County, on fallen trunk of *Quercus*, 1 August 2009, B.K. Cui, *Cui 7051* (BJFC 005538). Ningxia Autonomous Region, Liupan Mountain, on fallen angiosperm trunk, 11 July 2010, Y.C. Dai, *Dai 11939* (BJFC 009040). Russia. Khankaisky District, Novo-Kachalinsk, on fallen trunk of *Quercus*, 26 July1996, Leley, *Leley 296423* (LE). Primorsky Territory, Sikhote-Alin Nature Reserve, vicinities of Ust’-Serebryany, on fallen trunk, 4 September 1996, Morozova, (LE). Republic of Buryatia, Kabansky District, on fallen trunk of *Populus*, 9 August 1998, Morozova, *262806* (LE).

(18) *Auricularia lateralis* Y.C. Dai & F. Wu, sp. nov. Figure 10d and Figure 25.

MycoBank number: MB 825098.

Type—China. Yunnan Province, Yongde County, Daxueshan Nature Reserve, E 99°38′, N 24°9′, on rotten angiosperm wood, 27 August 2015, *Dai 15670* (BJFC 019774, holotype).

Etymology—*Lateralis* (Lat.): refers to the wide abhymenial hairs with slightly swollen center.

Basidiomata—Gelatinous when fresh, reddish brown to fawn, solitary or caespitose, sessile or substipitate; pileus discoid or auriculate, sometimes with lobed margin, projecting up to 5 cm, 1–3 mm thick, 0.2–0.3 mm thick and fawn to fuscous when dry; upper surface distinctly pilose, sometimes with a few folds; hymenophore surface porose-reticulate.

Internal features—Medulla present in the middle of the cross-section; crystals absent; abhymenial hairs with a slightly swollen center, hyaline, thick-walled, with a wide septate lumen, apical tips acute or obtuse, single or tufted, 95–250 × 9–14 µm; hyphae with clamp connections and simple septa, 1–6 µm in diam in KOH; basidia clavate, transversely 3-septate, with oil guttules, 50–70 × 5–6.5 µm, sterigmata rarely observed; cystidioles absent.

Spores—Basidiospores allantoid, hyaline, thin-walled, smooth, usually with one or two large guttules, IKI–, CB–, (12.3–)12.9–14.2(–15) × (5–)5.2–6(–6.2) µm, L = 13.56 µm, W = 5.73 µm, Q = 2.37 (n = 30/1).

Distribution—China.

Notes—*Auricularia lateralis* is characterized macroscopically by a conspicuously porose-reticulate hymenophore, and microscopically by wide abhymenial hairs with slightly swollen center. The species belongs to the *A. delicata* complex and it is morphologically close to *A. pilosa* Y.C. Dai & F. Wu sharing abhymenial hairs > 100 μm long, but *A. pilosa* has shorter basidia (35–45 × 4–5.5 µm) and is distributed in Africa. Phylogenetically, the species is distantly related to *A. pilosa* and forms a distinct lineage with high support (Figure 1 and Figure 2).

Additional specimens (paratypes) examined—China. Yunnan Province, Yongde County, Daxueshan Nature Reserve, on rotten angiosperm wood, 27 August 2015, Y.C. Dai, *Dai 15669* (BJFC 019773); 5 August 2014, Y.C. Dai, *Dai 16416* (BJFC 022535), *Dai 16417* (BJFC 022536), *Dai 16418* (BJFC 022537), *Dai 16419* (BJFC 022538), *Dai 16420* (BJFC 022539).

(19) *Auricularia mesenterica* (Dicks.) Pers. Figure 3e and Figure 26.

≡*Helvella mesenterica* Dicks., Fasc. pl. crypt. brit. (London) 1: 20, 1785.

=*Auricularia tremelloides* Bull., Herb. Fr. (Paris) 7: Table 290, 1787.

=*Auricularia lobata* Sommerf. Mag. Naturvididensk. 6: 295, 1826.

Basidiomata—Gelatinous when fresh, greyish brown to fuscous or buff to white, caespitose, resupinate to effused-reflexed; pileus free lobed, margin undulate, projecting up to 7 cm, 1.5–3 mm thick, 0.2–0.3 mm thick when dry; upper surface hispid, distinctly and concentrically zoned with canescent zones and dark bands, becoming olivaceous buff upon drying; hymenophore surface venose with obvious folds, becoming fawn to reddish brown upon drying.

Internal features—Medulla absent; crystals absent; abhymenial hairs with a slightly swollen base, hyaline, thick-walled, with a narrow lumen, apical tips acute or obtuse, tufted, 1000–2000 × 2–4 µm; hyphae with clamp connections, obviously inflated with a lumen in KOH, up to 14 µm in diam; basidia clavate, transversely 3-septate, with oil guttules, 48–68 × 4–6.5 µm, sterigmata rarely observed; cystidioles present, 5–9 × 2.5–5 µm.

Spores—Basidiospores allantoid, hyaline, thin-walled, smooth, usually with one or two large guttules, IKI–, CB–, (13.8–)14–17(–17.6) × (4.5–)4.7–5.2(–5.3) µm, L = 15.45 µm, W = 5.01 µm, Q = 3.08–3.09 (n = 60/2).

Distribution—Europe and Uzbekistan. 

Notes—*Auricularia mesenterica* is a common species in Europe and grows on species of different angiosperm wood usually in summer to autumn. The species was originally described from Europe, but is also reported from the Americas and Asia [14,20,43]. However, *A. mesenterica* was reported as a species complex with three species (*A. asiatica*, *A. brasiliana* and *A. orientalis* Y.C. Dai & F. Wu) [25,26]. In the present study, the other two new species, *A. africana* and *A. submesenterica* Y.C. Dai & F. Wu, are described. In addition, *A. pusio*, originally described from Australia [58], is re-studied and phylogenetically analyzed. We confirmed that it belongs to the *A. mesenterica* complex and is an independent species.

Morphologically, the *A. mesenterica* complex is distinct from other species in the genus by its resupinate to effused-reflexed basidiomata, the upper surface usually with distinct and concentric zones and a venose hymenophore surface with obvious folds. *Auricularia mesenterica* has the largest basidiospores among the species in the complex. Phylogenetically, *A. mesenterica* forms a single lineage with strong support and is closely related to *A. orientalis* and *A. submesenterica* (Figure 1 and Figure 2).

Specimens examined—Czech Republic. Bukovinka, Rakovecký les forest, on fallen angiosperm trunk, 15 September 2007, A. Vágner, (BRNM 706955, duplicate, BJFC 018552); Vranovice upon Svratka, Plačkův les Nature Reserve, on the trunk of *Fraxinus*, 19 November 1999, A. Vágner, (BRNM 648573, duplicate, BJFC 018554). Estonia. Viljandi District, on fallen angiosperm wood, 26 August 1989, I. Kytovuori, *Kytovuori 89-333* (H). France. Péage de Roussillon, on the trunk of *Populus*, 15 April 2014, B. Rivoire, *LYBR* 5353 (BJFC 018549). Italy. Parma, La Cittadella Park, on rotten angiosperm stump, 27 February 1987, V. Härkönen, *Härkönen 8045* (H). Sweden. Närke, Almby, Hjälmarsberg, on rotten wood of *Ulmus*, 28 September 1998, *Nilsson*, *F 12670* (S). Switzerland. Sparse, on fallen trunk of *Quercus*, 1 June 2008, O. Miettinen, *Miettinen 12680* (H). UK. Wales, Gwent, Abergavenny Riverside, on rotten angiosperm trunk, 5 March 1990, V. Härkönen, *Härkönen 11208* (H). Uzbekistan. Profenicel garden, Taslvencf, on angiosperm trunk, 2 September 2011, Y. Gofforov, *YG 029* (BJFC 020591); Xumson village, Suglens forest, on fallen trunk of *Juglans*, 26 September 2011, Y. Gofforov, *YG 037* (BJFC 020592). 

(20) *Auricularia minor* Kobayasi Figure 5g and Figure 27.

Basidiomata—Gelatinous when rehydrated, greyish brown to fawn, solitary, sessile; pileus discoid or auriculate, margin entire, projecting up to 6.5 cm, 0.65–0.8 mm thick, 0.1–0.2 mm thick when dry; upper surface scantly pilose, becoming mouse-grey upon drying; hymenophore surface smooth, sometimes with indistinct folds, becoming black upon drying.

Internal features—Medulla present in the middle of the cross-section; crystals absent; abhymenial hairs with a slightly swollen base, hyaline or clay buff, thick-walled, with a narrow lumen, sometimes septate, apical tips acute or obtuse, single or tufted, 90–120 × 6–10 µm; hyphae with clamp connections, 1–2 µm in diam in KOH; basidia clavate, transversely 3-septate, with oil guttules, 35–43 × 2–4 µm, sterigmata rarely observed; cystidioles absent.

Spores—Only one possible true basidiospore was observed in the type, and was reported as measuring 7–8 × 3–4 µm by Kobayasi [12].

Distribution—Asia.

Notes—*Auricularia minor* was originally described from tropical Asia [12], and is characterized by its solitary and very small basidiomata. We failed to extract DNA because the type material and other samples are in poor condition, so its phylogenetic relations with other species are uncertain. Morphologically, *A. minor* is very similar to *A. hainanensis* described from tropical Asia, but the abhymenial hairs in *A. minor* (90–120 × 6–10 µm) are longer than those in *A. hainanensis* (30–80 × 4–5 µm).

Specimens examined—New Guinea. 1981, *F-434* (TNS, type). Japan. 4 October 1981, *F-51392* (TNS).

(21) *Auricularia minutissima* Y.C. Dai, F. Wu & Malysheva Figure 5h,i and Figure 28.

Basidiomata—Gelatinous when fresh, vinaceous grey to fuscous or clay buff to pinkish buff, solitary or caespitose, sessile; pileus discoid or auriculate, sometimes with lobed margin, projecting up to 4 cm, 0.5–2 mm thick, 0.04–0.2 mm thick when dry; upper surface pilose, sometimes with folds, becoming vinaceous grey upon drying; hymenophore surface smooth or with obvious folds, becoming fuscous or black upon drying.

Internal features—Medulla present in the middle of the cross-section; crystals present, usually scattered in the hymenium; abhymenial hairs with a slightly swollen base, hyaline, thick-walled, with a narrow lumen, sometimes septate, apical tips acute or obtuse, single or tufted, 50–85(–105) × 5–6 µm; hyphae with clamp connections, 1–3.5 µm in diam in KOH; basidia clavate, transversely 3-septate, with oil guttules, 46–67 × 5–8 µm, sterigmata occasionally observed; cystidioles absent.

Spores—Basidiospores allantoid, hyaline, thin-walled, smooth, usually with one or two large guttules, IKI–, CB–, (11.8–)13–15(–16.2) × 5–6(–7) µm, L = 14.28 µm, W = 5.83 µm, Q = 2.44–2.46 (n = 80/3).

Distribution—China and Russia.

Notes—*Auricularia minutissima* belongs to the *A. auricula-judae* complex and is characterized by its small basidiomata. The species is similar to *A. heimuer* and *A. minor,* but the latter two species have smaller basidiospores (11–13 × 4–5 µm in *A. heimuer* and 7–8 × 3–4 µm in *A. minor*). *A. minutissima* has two kinds of basidiomata (Figure 5h,i) that are different in macro-morphology, but have the similar basidia, basidiospores and DNA data. In the phylogeny, *A. minutissima* is closely related to *A. heimuer*, but it forms a single lineage with strong support (Figure 1 and Figure 2). The morphological description of *A. minutissima* taken from Wu et al. [23].

Specimens examined—China. Gansu Province, Pingliang, Kongtongshan, on rotten branch of *Acer*, 3 August 2015, Y.C. Dai, *Dai 15452* (BJFC 019557), *Dai 15453* (BJFC 019558), *Dai 15454* (BJFC 019559), *Dai 15455* (BJFC 019560), *Dai 15456* (BJFC 019561). Hebei Province, Laishui County, Yesanpo Nature Reserve, on fallen branch of *Quercus*, 18 November 2014, Y.C. Dai, *Dai 14880* (BJFC 017993), *Dai 14881* (BJFC 017994, holotype). Russia. Primorye Territory, Kedrovaya Pad’Reserve, 10 October 1994, Govorova, (LE 296424).

(22) *Auricularia nigricans* (Sw.) Birkebak, Looney & Sánchez-García

Figure 15k and Figure 29.

≡*Peziza nigricans* Sw., Fl. Ind. Occid.3: 1938, 1806.

=*Exidia polytricha* Mont., Voy. Indes Or., Bot. 2: 154, 1834.

Basidiomata—Gelatinous when rehydrated, reddish brown to orange-brown, solitary or caespitose, sessile or substipitate; pileus cupulate or auriculate, sometimes with lobed margin, projecting up to 6 cm, 1–3 mm thick, 0.36–0.4 mm thick when dry; upper surface usually densely hispid, becoming vinaceous grey upon drying; hymenophore surface usually smooth, without folds, becoming dark-grey upon drying.

Internal features—Medulla obviously present near the abhymenium; crystals present, usually scattered in the hymenium; abhymenial hairs with a slightly swollen base, hyaline, thick-walled, with a narrow lumen, apical tips acute or obtuse, tufted, 300–600 × 7–9 µm; hyphae with clamp connections and simple septa, 0.5–4 µm in diam in KOH; basidia clavate, transversely 3-septate, with oil guttules, 50–60 × 4–6 µm, sterigmata rarely observed; cystidioles absent.

Spores—Basidiospores not observed in the studied samples, but reported as measuring 14.5–17 × 5–7 µm by Looney et al. [20].

Distribution—North America.

Notes—*Auricularia polytricha* is a name that has been applied to this species for a long time [12,17,51], but Looney et al. recently found that *Exidia polytricha* Mont. and *Peziza nigricans* represent the same species [20]. Since *P. nigricans* was an earlier name and thus had the priority, they proposed the combination *A. nigricans* (Sw.) Birkebak & al., and *A. polytricha* is treated as a synonym of *A. nigricans*.

*Auricularia nigricans* is characterized by the densely hispid upper surface and obvious medulla near the abhymenium. It is easily confused with *A. cornea* because of the variable morphology of *A. cornea*, but *A. nigricans* usually has a more hispid upper surface and is distributed in North America only. Phylogenetically, both species are distantly related (Figure 1 and Figure 2).

Specimens examined—Costa Rica. On fallen angiosperm trunk, 4 January 1979, *Ahti 36234* (H). Mexico. On fallen angiosperm trunk, 18 October 1997, T. Ahti, *Ahti 55718* (H). USA. Louisiana, on fallen trunk of *Quercus*, 1993, T.J. Volk, *TJV-93-242* (CFMR).

(23) *Auricularia novozealandica* Y.C. Dai & F. Wu, sp. nov. Figure 15l and Figure 30.

Mycobank number: MB 825099.

Type—New Zealand. Auckland, S 36°50′, E 174°44′, on stump, 12 May 2005, B. Dee, (PDD 83897, holotype).

Etymology—*Novozealandica* (Lat.): refers to the species from New Zealand.

Basidiomata—Gelatinous when rehydrated, reddish brown to fuscous, solitary or caespitose, sessile or substipitate; pileus cupulate or auriculate, sometimes with lobed margin, projecting up to 15 cm, 0.8–1.5 mm thick, 0.3–0.4 mm thick when dry; upper surface densely pilose, becoming greyish brown to clay buff upon drying; hymenophore surface usually smooth, without folds, becoming vinaceous grey upon drying.

Internal features—Medulla obviously present in the middle of the cross-section or near the abhymenium; crystals absent; abhymenial hairs with a slightly swollen base, hyaline, thick-walled, with a wide or narrow lumen, apical tips with several septate lumens, obtuse, tufted, 100–320 × 7–9 µm; hyphae with clamp connections and simple septa, 0.5–3 µm in diam in KOH; basidia clavate, transversely 3-septate, with oil guttules, 70–86 × 5–6 µm, sterigmata rarely observed; cystidioles absent.

Spores—Basidiospores allantoid, hyaline, thin-walled, smooth, usually with one or two large guttules, IKI–, CB–, (15–)16–19(–19.5) × (5.1–)5.3–6.1(–6.5) µm, L = 17.47 µm, W = 5.83 µm, Q = 2.94–3.05 (n = 90/3).

Distribution—New Zealand.

Notes—*Auricularia novozealandica* belongs to the *A. cornea* complex and is characterized by dense hairs on the upper surface, obvious medulla in the middle of the cross-section or near the abhymenium and very big basidiospores. It is morphologically and phylogenetically related to *A. cornea*, but *A. cornea* has smaller basidia (60–75 × 4–6 µm) and smaller basidiospores (13.8–16.5 × 4.5–6 µm). 

Additional specimens (paratypes) examined—New Zealand. Auckland, 29 January 1998, R. Leschen, (PDD 70405); 14 March 2001, S.R. Whitton, C. Delleey, R. Fuller, (PDD 75110); 29 May 1999, C. Shirley, (PDD 81195); on dead wood, 8 May 2006, P. Catcheside, D. Catcheside, (PDD 88998). Nelson, 11 May 2004, D. Catcheside, (PDD 94628). Wellington, 25 March 1997, A. Freeston, R. Freeston, (PDD 94817). 

(24) *Auricularia orientalis* Y.C. Dai & F. Wu Figure 3f and Figure 31.

Basidiomata—Gelatinous when fresh, greyish brown to fuscous, solitary or caespitose, resupinate to effused-reflexed; pileus free lobed, margin undulate, projecting up to 2 cm, 2–3 mm thick, 0.2–0.3 mm thick when dry; upper surface densely villose, sometimes concentrically zoned with canescent zones and dark bands, becoming white to olivaceous buff upon drying; hymenophore surface venose with folds, becoming greyish blue to dark greyish blue upon drying.

Internal features—Medulla absent; crystals absent; abhymenial hairs with a slightly swollen base, hyaline, thick-walled, with a narrow lumen, apical tips acute or obtuse, tufted, 900–1500(–1800) × 1.5–2.5 µm; hyphae with clamp connections, obviously inflated with a lumen in KOH, up to 7 µm in diam; basidia clavate, transversely 3-septate, with oil guttules, 50–72 × 3.5–6 µm, sterigmata rarely observed; cystidioles present, 6–8 × 3–3.5 µm.

Spores—Basidiospores allantoid, hyaline, thin-walled, smooth, usually with one to three large guttules, IKI–, CB–, (12–)12.5–14.2(–15) × 5–6 µm, L = 13.44 µm, W = 5.48 µm, Q = 2.45 (n = 30/1).

Distribution—China.

Notes—*Auricularia orientalis* was described as a new species in the *A. mesenterica* complex [25]. Compared to other species in this complex, only *A. orientalis* and *A. mesenterica* have cystidioles, but *A. orientalis* has smaller basidiospores than those of *A. mesenterica* (12.5–14.2 × 5–6 µm vs. 14–17 × 4.7–5.2 µm). Phylogenetically, samples of *A. orientalis* clustered in a single lineage with robust support (Figure 1 and Figure 2). The morphological description of *A. orientalis* taken from Wu et al. [25].

Specimens examined—China. Beijing, Baihuashan Nature Reserve, on fallen angiosperm trunk, 2 November 1993, Y.C. Dai, *Dai 1831* (BJFC 019131). Hebei Province, Laishui County, Yesanpo National Park, on rotten stump of *Juglans*, 17 November 2014, Y.C. Dai, *Dai 14875* (BJFC 017988, holotype). Inner Mongolia Autonomous Region, Daqinggou Nature Reserve, on rotten angiosperm wood, 25 August 2015, Y.C. Dai, *Dai 15813* (BJFC 019916). 

(25) *Auricularia papyracea* Yasuda Figure 20c and Figure 32.

Basidiomata—Gelatinous when rehydrated, reddish brown to cinnamon, caespitose, sessile; pileus discoid, margin entire, projecting up to 2 cm, 0.65–1 mm thick, 0.02–0.04 mm thick and yellowish brown when dry; upper surface scantly pilose, distinctly and concentrically zoned; hymenophore surface smooth, sometimes with folds.

Internal features—Medulla absent; crystals absent; abhymenial hairs with a slightly swollen base, hyaline or clay buff, thick-walled, with a narrow lumen, apical tips acute, single, 20 × 7 µm; hyphae with clamp connections, 2.8–4 µm in diam in KOH; basidia clavate, transversely 3-septate, with oil guttules, 60–70 × 8–10 µm, sterigmata rarely observed; cystidioles absent.

Spores—Basidiospores allantoid, hyaline, thin-walled, smooth, usually with one to three small guttules, IKI–, CB–, 16–18(–18.2) × (4–)4.2–5.3(–5.5) µm, L = 17.17 µm, W = 4.86 µm, Q = 3.53 (n = 30/1).

Distribution—Japan.

Notes—*Auricularia papyracea* is characterized by the extremely thin and papery basidiomata when dry. It is easy to distinguish from other species in the genus by the concentric white zones on the upper surface. No fresh samples were available, and DNA extraction from the type and another sample failed. Its relation within the genus are uncertain.

Specimens examined—Japan. 17 December 1911, *F-234520* (TNS, type); 7 December 1917, *F-203284* (TNS).

(26) *Auricularia pilosa* Y.C. Dai, L.W. Zhou & F. Wu, sp. nov. Figure 33.

MycoBank number: MB 840812.

Type—Ethiopia. Jimma, Shebe, N 7°34′, E 36°25′, on angiosperm trunk, 21 April 2019, L.W. Zhou, LWZ20190421-7 (HMAS, holotype).

Etymology—*Pilosa* (Lat.): refers to the distinctly pilose abhymenial surface.

Basidiomata—Gelatinous when fresh, reddish brown to vinaceous brown, solitary or caespitose, sessile or substipitate; pileus discoid or auriculate, sometimes with lobed margin, projecting up to 4 cm, 1–2 mm thick, 0.1–0.2 mm thick and reddish brown to fuscous when dry; upper surface distinctly pilose; hymenophore surface porose-reticulate.

Internal features—Medulla absent; crystals absent; abhymenial hairs with a slightly swollen base, hyaline, thick-walled, with a wide septate lumen, apical tips acute or obtuse, single or tufted, 90–207 × 8–16 µm; hyphae with clamp connections and simple septa, 1–5 µm in diam in KOH; basidia clavate, transversely 3-septate, with oil guttules, 35–45 × 4–5.5 µm, sterigmata rarely observed; cystidioles absent.

Spores—Basidiospores allantoid, hyaline, thin-walled, smooth, usually with one or two large guttules, IKI–, CB–, (10.5–)11–13.8(–14.8) × (3.8–)4.2–5.8(–6.1) µm, L = 12.46 µm, W = 4.95 µm, Q = 2.52 (n = 60/1).

Distribution—Ethiopia and Tanzania.

Notes—*Auricularia pilosa* is characterized by the pilose upper surface, porose-reticulate hymenophore, and wide abhymenial hairs with slightly swollen center. The species belongs to the *A. delicata* complex and is easily confused with *A. lateralis* because of the distinctly pilose abhymenial surface, but it differs by the slightly shorter hairs and basidia (in *A. lateralis* 95–250 µm long and 50–70 µm long, respectively) and is distributed in Africa. Phylogenetically, *A. pilosa* forms a single lineage distant from other species in the *A. delicata* complex (Figure 1 and Figure 2).

(27) *Auricularia pusio* Berk. Figure 3g and Figure 34.

Basidiomata—Gelatinous when rehydrated, fawn to greyish brown, caespitose, resupinate to effused-reflexed; pileus free lobed, margin undulate, projecting up to 3 cm, 1.5–2 mm thick, 0.2–0.4 mm thick when dry; upper surface tomentose, sometimes concentrically zoned with canescent zones and dark bands, becoming clay pink upon drying; hymenophore surface venose with folds, becoming dark greyish blue upon drying.

Internal features—Medulla absent; crystals present, usually scattered in the hymenium; abhymenial hairs with a slightly swollen base, hyaline, thick-walled, with a narrow lumen, apical tips acute or obtuse, tufted, 400–800 × 1.5–2 µm; hyphae with clamp connections and simple septa, slightly inflated with a lumen in KOH, up to 6 µm in diam; basidia clavate, transversely 3-septate, with oil guttules, 50–72 × 5–7.5 µm, sterigmata frequently observed; cystidioles absent.

Spores—Basidiospores allantoid, hyaline, thin-walled, smooth, usually with one or two large guttules, IKI–, CB–, 12–14(–14.2) × (4.8–)5–6 µm, L = 12.97 µm, W = 5.43 µm, Q = 2.39 (n = 30/1).

Distribution—Australia and Zambia.

Notes—*Auricularia pusio* was originally described from Queensland (Australia) and was considered a synonym of *A. mesenterica* in the light of morphological characters (http://www.indexfungorum.org/Names/NamesRecord.asp?RecordID=156648, accessed on 3 September 2021), but *A. mesenterica* is a species complex [25]. Basidiospores of *A. pusio* are distinctly shorter and wider than those in *A. mesenterica sensu stricto* (14–17 × 4.7–5.2 µm). Additionally, our phylogenies show that the two species form two distinct and distantly related lineages (Figure 1 and Figure 2). Therefore, we accept *A. pusio* as an independent species in the *A. mesenterica* complex. 

Specimens examined—Australia. Western Australia, Kimberley District, Kaugaroo Pool, on dead log, 30 August 1999, B.M. Spooner, *AK 547* (K 26101); Napier Range, Secret Valley, on dead standing shrub, 13 April 1988, B.M. Spooner, *AK 174* (K 26100). Zambia. Mpika, North Luangwa, on 20 September 1994, D.S. Smith, *Smith 18* (K 28316).

(28) *Auricularia scissa* Looney, Birkebak & Matheny Figure 35.

Basidiomata—Gelatinous when rehydrated, greyish brown to reddish brown, solitary or caespitose, sessile or substipitate; pileus discoid or auriculate, with lobed margin, projecting up to 7 cm, 1–3 mm thick, 0.1–0.2 mm thick and yellowish brown when dry; upper surface pilose; hymenophore surface conspicuously porose-reticulate.

Internal features—Schizomedulla present in the middle of the cross-section; crystals absent; abhymenial hairs with a slightly swollen base, hyaline, thick-walled, with a narrow lumen, apical tips acute or obtuse, single or tufted, 40–100 × 5–10 µm; hyphae with clamp connections, 0.5–5 µm in diam in KOH; basidia clavate, transversely 3-septate, with oil guttules, 40–60 × 4–5 µm, sterigmata rarely observed; cystidioles absent.

Spores—Basidiospores allantoid, hyaline, thin-walled, smooth, usually with one to a few large guttules, IKI–, CB–, (8–)9–12 × 4–6(–6.5) µm, L = 10.73 µm, W = 5.43 µm, Q = 1.97 (n = 30/1).

Distribution—Dominican Republic and USA.

Notes—*Auricularia scissa* was recently described from the Dominican Republic by Looney et al. [20], and is characterized macroscopically by conspicuously porose-reticulate hymenophores and microscopically by the presence of schizomedulla. It is like *A. subglabra* as both have porose-reticulate hymenophores and a schizomedulla, but the latter species has an almost smooth abhymenium and very short hairs (up to 45 µm). Phylogenetically, *A. scissa* forms a single lineage with high support (Figure 1 and Figure 2) and is not closely related to *A. subglabra*. 

Specimens examined—Dominican Republic. 31 December 1990, T. Ahti, *Ahti 49388* (H); 1998, O.P. Perdomo, *DR 777* (CFMR).

(29) *Auricularia sinodelicata* Y.C. Dai & F. Wu, sp. nov. Figure 10e–j and Figure 36.

MycoBank number: MB 825100.

Type—China. Chongqing, Jiangjin District, Simianshan Nature Reserve, E 104°4′, N 28°6′, on fallen angiosperm trunk, 22 July 2014, Y.C. Dai, *Dai 13926* (BJFC 017656, holotype).

Etymology—*Sinodelicata* (Lat.): refers to the species being like *A. delicata* and with the distribution in China.

Basidiomata—Gelatinous when fresh, fawn to reddish brown or cinnamon to yellowish brown, solitary or caespitose, sessile or substipitate; pileus discoid or auriculate, sometimes with lobed margin, projecting up to 8 cm, 1–2.5 mm thick, 0.16–0.25 mm thick and reddish brown to black when dry; upper surface scantly pilose, sometimes with a few folds; hymenophore surface conspicuously porose-reticulate.

Internal features—Medulla indistinctly present near the hymenium; crystals absent; abhymenial hairs with a slightly swollen base, hyaline, thick-walled, with a wide or narrow septate lumen, apical tips acute or obtuse, single, 30–80 × 6–9 µm; hyphae with clamp connections, 1–5 µm in diam in KOH; basidia clavate, transversely 3-septate, with oil guttules, 30–45 × 4–5.5 µm, sterigmata rarely observed; cystidioles absent.

Spores—Basidiospores allantoid, hyaline, thin-walled, smooth, usually with one to three large guttules, IKI–, CB–, (9.8–)10–12(–12.2) × (4–)4.3–5.1(–5.5) µm, L = 10.87 µm, W = 4.84 µm, Q = 2.2–2.27 (n = 120/4).

Distribution—China.

Notes—Previously, Chinese samples of *A. sinodelicata* were identified as *A. delicata* because of the porose-reticulate hymenophore [42,44]). However, *A. delicata* was originally described from West Africa [12,17], and phylogenetically *A. sinodelicata* and *A. delicata* cluster in two different lineages (Figure 1). Besides, *A. delicata* has longer basidia than those of *A. sinodelicata* (48–65 × 4–6 µm vs. 30–45 × 4–5.5 µm).

*Auricularia lateralis* is described as a new species in the present study and also has a porose-reticulate hymenophore. *Auricularia lateralis* and *A. sinodelicata* have an overlapping distribution in southern China, but the basidiospores are distinctly larger in *A. lateralis* (12.9–14.2 × 5.2–6 µm) than in *A. sinodelicata* (10–12 × 4.3–5.1 µm). *Auricularia sinodelicata* has variable macro-morphological characters (Figure 10e–j) and is extremely similar to *A. tremellosa* in some specimens (e.g., Cui 12236, Figure 10e), but *A. sinodelicata* usually has more or less radial folds on the hymenophore, while *A. tremellosa* usually has poroid hymenophore. More importantly, *A. sinodelicata* and *A. tremellosa* are distantly related in phylogenies (Figure 1 and Figure 2), and they are distributed in Asia and the Americas, respectively.

Additional specimens (paratypes) examined—China. Chongqing, Jiangjin District, Simianshan Nature Reserve, on fallen angiosperm trunk, 22 July 2014, Y.C. Dai, *Dai 13927* (BJFC 017657). Guizhou Province, Guiyang, Qianlingshan Park, on rotten angiosperm wood, 17 July 2014, Y.C. Dai, *Dai 13919* (BJFC 017649), *Dai 13920* (BJFC017650). Hainan Province, Lingshui County, Diaoluoshan Forest Park, on fallen angiosperm trunk, 13 June 2014, Y.C. Dai, *Dai 13593A* (BJFC 017332). Xizang Autonomous Region (Tibet), Motuo County, on fallen angiosperm trunk, 20 September 2014, B.K. Cui, *Cui 12236* (BJFC 017150); 9 September 2014, Y.C. Dai, *Dai 15083* (BJFC 018195). Yunnan Province, Jinghong, Wangtianshu Nature Reserve, on fallen angiosperm trunk, 2 November 2009, B.K. Cui, *Cui 8594* (BJFC 007083), *Cui 8596* (BJFC 007085), 19 July 2014, Y.C. Dai, *Dai 13746* (BJFC 017477), *Dai 13748* (BJFC 017479); Puer, Taiyanghe Forest Park, on fallen angiosperm trunk, 6 June 2011, Y.C. Dai, *Dai 12242* (BJFC 010525), 24 July 2014, Y.C. Dai, *Dai 13758* (BJFC 017489); Tengchong County, Yinghua Valley, on fallen angiosperm trunk, 28 October 2009, B.K. Cui, *Cui 8315* (BJFC 006804), *Cui 8316* (BJFC 006805);Yingjiang County, Tongbiguan Nature Reserve, on rotten angiosperm wood, 29 October 2010, Y.C. Dai, *Dai 13112* (BJFC 013332).

(30) *Auricularia srilankensis* Y.C. Dai & F. Wu, sp. nov. Figure 3h and Figure 37.

Mycobank number: MB 840813.

Type—Sri Lanka. Wadduwa, South Bolgoda Lake, N 6°46′, E 79°54′, on dead angiosperm tree, 28 February 2019, Y.C. Dai, *Dai 19522* (BJFC031201, holotype).

Etymology—*Srilankensis* (Lat.): refers to the species distribution in Sri Lanka.

Basidiomata—Gelatinous when fresh, greyish blue to vinaceous grey, caespitose, resupinate to effused-reflexed; pileus free lobed, margin undulate, projecting up to 1 cm, 1–2 mm thick, 0.08–0.16 mm thick when dry; upper surface villose, sometimes concentrically zoned with canescent zones and dark bands, becoming white to greyish violet upon drying; hymenophore surface venose with folds, becoming dark greyish blue upon drying.

Internal features—Medulla absent; crystals present; abhymenial hairs with a slightly swollen base, hyaline, thick-walled, with a narrow lumen, apical tips acute or obtuse, tufted, 400–800 × 1.5–3 µm; hyphae with clamp connections, obviously inflated with a lumen in KOH, up to 7 µm in diam; basidia clavate, transversely 3-septate, with some small oil guttules, 50–70 × 5–7 µm, sterigmata usually observed; cystidioles absent.

Spores—Basidiospores allantoid, hyaline, thin-walled, smooth, usually with one or two large guttules, IKI–, CB–, (11–)12–14.7(–15) × (4.2–)4.5–6 µm, L = 13.18 µm, W = 5.26 µm, Q = 2.31–2.82 (n = 90/3).

Distribution—Sri Lanka.

Notes—*Auricularia srilankensis* is phylogenetically closely related to *A. asiatica*, but it forms one single lineage with strong support (Figure 1 and Figure 2) and differs from *A. asiatica* by larger basidia (40–52 × 3–6 µm in *A. asiatica*) and slightly longer basidiospores (11.2–12.3 µm long in *A. asiatica*). Morphologically, the species is easily confused with *A. orientalis* from China, but *A. orientalis* has cystidioles and longer hairs (900–1500 µm). 

Specimens examined—Sri Lanka. Wadduwa, South Bolgoda Lake, on dead angiosperm tree, 28 February 2019, Y.C. Dai, *Dai 19519* (BJFC031198), *Dai 19520* (BJFC031199); Kandy, Udawatta Kele Royal Forest Park, on fallen angiosperm trunk, 2 March 2019, Y.C. Dai, *Dai 19575* (BJFC031254).

(31) *Auricularia subglabra* Looney, Birkebak & Matheny Figure 10k and Figure 38.

Basidiomata—Gelatinous when fresh, greyish violet to fawn, solitary or caespitose, sessile or substipitate; pileus discoid or auriculate, margin entire, projecting up to 6.5 cm, 1–2 mm thick, 0.24–0.4 mm thick and cinnamon buff to vinaceous grey when dry; upper surface scantly pilose, sometimes with a few folds; hymenophore surface slightly porose-reticulate.

Internal features—Schizomedulla present in the middle of the cross-section; crystals absent; abhymenial hairs with a slightly swollen base, hyaline, thick-walled, with a wide or narrow septate lumen, apical tips acute or obtuse, single, 20–45 × 5–6 µm; hyphae with clamp connections and simple septa, 1–5 µm in diam in KOH; basidia clavate, transversely 3-septate, with oil guttules, 30–45 × 3.5–4.5 µm, sterigmata rarely observed; cystidioles absent.

Spores—Basidiospores allantoid, hyaline, thin-walled, smooth, usually with one or two large guttules, IKI–, CB–, (8.5–)9–10.8(–11) × (3–)3.5–4.5 µm, L = 9.72 µm, W = 3.79 µm, Q = 2.56 (n = 30/1).

Distribution—Tropical North and South America.

Notes—*Auricularia subglabra* was described from Costa Rica by Looney et al. [20], and is characterized by slightly porose-reticulate hymenophores and the presence of schizomedulla. The species is similar to *A. scissa*, which, however, has a pilose upper surface and longer abhymenial hairs (40–100 µm). Phylogenetically, both species nest in two distinct lineages with high support (Figure 1 and Figure 2).

Specimens examined—Brazil. Amazonas, Manaus, on rotten angiosperm wood, 12 May 2017, Y.C. Dai, *Dai 17394* (BJFC 026424), *Dai 17403* (BJFC 026425), *Dai 17417* (BJFC 026427), F. Wu, *Wu 08* (BJFC 026014), *Wu 09* (BJFC 026014). French Guiana. Coralie, on rotten angiosperm trunk, 29 August 2018, J. Vlasák, *JV 1808/125* (JV).

(32) *Auricularia submesenterica* Y.C. Dai & F. Wu, sp. nov. Figure 3i and Figure 39.

MycoBank number: MB 825101.

Type—China. Jilin Province, Antu County, Changbaishan Nature Reserve, E 127°45′, N 41°38′, on fallen angiosperm trunk, 16 July 2015, *Dai 15450* (BJFC 019555, holotype).

Etymology—*Submesenterica* (Lat.): refers to the species being similar to *Auricularia mesenterica*.

Basidiomata—Gelatinous when fresh, greyish brown to fuscous, caespitose, resupinate to effused-reflexed; pileus free lobed, margin undulate, projecting up to 7 cm, 1–2 mm thick, 0.12–0.2 mm thick when dry; upper surface hispid, distinctly and concentrically zoned with canescent zones and dark bands, becoming greyish brown to deep olive upon drying; hymenophore surface venose with obvious folds, becoming dark bluish grey to dark grey upon drying.

Internal features—Medulla absent; crystals present, usually scattered in hymenium; abhymenial hairs with a slightly swollen base, hyaline, thick-walled, with a narrow lumen, apical tips acute or obtuse, tufted, 600–1000 × 2–2.5 µm; hyphae with clamp connections, obviously inflated in KOH with a wide lumen, up to 7.5 µm in diam; basidia clavate, transversely 3-septate, with oil guttules, 37–50 × 4–6.5 µm, sterigmata rarely observed; cystidioles absent.

Spores—Basidiospores allantoid, hyaline, thin-walled, smooth, usually with one or two large guttules, IKI–, CB–, 12.5–14.8 × 5.5–6 µm, L = 13.75 µm, W = 5.72 µm, Q = 2.4 (n = 4/1).

Distribution—China.

Notes—*Auricularia submesenterica* is characterized by caespitose, resupinate to effused-reflexed basidiomata with a lobed pileus, distinctly and concentrically zoned with canescent zones and dark bands on the upper surface, absence of medulla and cystidioles, and small basidia and basidiospores. *Auricularia submesenterica* definitely belongs to the *A. mesenterica* complex, but *A. mesenterica sensu stricto* differs from *A. submesenterica* by longer abhymenial hairs (1000–2000 µm), presence of cystidioles and larger basidiospores (14–17 × 4.7–5.2 µm). *Auricularia orientalis* is another member in the *A. mesenterica* complex and was recently described from China [25]; it resembles *A. submesenterica* by sharing similar abhymenial hairs and basidiospores, but *A. orientalis* usually has resupinate basidiomata and cystidioles in the hymenium. In addition, *A. submesenterica*, *A. orientalis* and *A. mesenterica* form three different lineages in our phylogenies (Figure 1 and Figure 2).

Additional specimens (paratypes) examined—China. Jilin Province, Antu County, Huangsongpu, on fallen angiosperm trunk, 28 September 1993, Y.C. Dai, *Dai 792* (BJFC 019132); Changbaishan Nature Reserve, on fallen angiosperm trunk, 16 July 2015, Y.C. Dai, *Dai 15449* (BJFC 019554), *Dai 15451* (BJFC 019556); on fallen trunk of *Quercus*, 11 September 2014, Y.C. Dai, *Dai 14773* (BJFC 017885). Liaoning Province, Qingyuan County, Qingyuan Ecological Station, on fallen angiosperm trunk, 26 August 2015, Y.C. Dai, *Dai 15820* (BJFC 019923).

(33) *Auricularia thailandica* Bandara & K.D. Hyde Figure 20d,e and Figure 40.

Basidiomata—Gelatinous when fresh, vinaceous to clay pink, solitary or caespitose, sessile or substipitate; pileus more or less circular, margin entire, projecting up to 6 cm, 0.4–0.7 mm thick when fresh, 0.04–0.24 mm thick and fawn to fuscous when dry; upper surface scantly pilose; hymenophore surface usually smooth, without folds, and usually with obvious white spots.

Internal features—Medulla present near the abhymenium; crystals present, very small, scattered throughout the cross-section; abhymenial hairs with a distinctly swollen base, irregular, hyaline, thick-walled with a wide septate lumen, apically acute or obtuse, single, 50–80(–100) × 6–14 µm; hyphae with clamp connections, 1–3 µm in diam in KOH; basidia clavate, transversely 3-septate, with oil guttules, 40–60 × 3–6 µm, sterigmata sometimes observed; cystidioles absent.

Spores—Basidiospores allantoid, hyaline, thin-walled, smooth, usually with one or two large guttules, IKI–, CB–, (9–)9.5–13(–14) × 4–5.6(–6) µm, L = 11.03 µm, W = 4.69 µm, Q = 1.83–2.68 (n = 210/7).

Distribution—China and Thailand.

Notes—*Auricularia thailandica* is characterized by very thin and fuscous basidiomata when dry. It is closely related to *A. fibrillifera* in the phylogenies (Figure 1 and Figure 2). For the differences between these two species, refer to the notes on *A. fibrillifera.*

Specimens examined—China. Guangxi Autonomous Region, Jinxiu County, Shengtang Mountain, on rotten angiosperm wood, 17 June 2014, Y.C. Dai, *Dai 15080* (BJFC 018192), *Dai 15081* (BJFC 018193). Hainan Province, Baoting County, on fallen angiosperm trunk, 1 July 2010, Y.L. Wei, *Wei 5513* (BJFC 009236); Changjiang County, Bawangling National Forest Park, on fallen angiosperm trunk, 16 June 2014, Y.C. Dai, *Dai 13687* (BJFC 017427); Danzhou County, Danzhou Tropical Botanical Garden, on fallen angiosperm trunk, 15 June 2014, Y.C. Dai, *Dai 13661* (BJFC 017401); Ledong County, Jianfengling Nature Reserve, on fallen angiosperm trunk, 17 June 2014, Y.C. Dai, *Dai 13706* (BJFC 017443); Qiongzhong County, Limu Mountain, on fallen angiosperm trunk, 15 June 2014, Y.C. Dai, *Dai 13629A* (BJFC 017368), *Dai 13635A* (BJFC 017374), *Dai 13654A* (BJFC 017393), *Dai 13655A* (BJFC 017394); Tunchang County, on fallen angiosperm trunk, 23 November 2010, Y.C. Dai, *Dai 11973* (BJFC 009265). Yunnan Province, Cangyuan County, on fallen angiosperm trunk, 11 July 2013, B.K. Cui, *Cui 11062* (BJFC 015178); Lancang County, on fallen angiosperm trunk, 9 July 2013, B.K. Cui, *Cui 11031* (BJFC 015147); Menhai County, Mangao Nature Reserve, on fallen trunk of *Castanea*, 8 June 2011, Y.C. Dai, *Dai 12332* (BJFC 010612); Puer County, Taiyanghe National Forest Park, on fallen angiosperm trunk, 24 July 2014, Y.C. Dai, *Dai 13759* (BJFC 017490).

(34) *Auricularia tibetica* Y.C. Dai & F. Wu Figure 5j and Figure 41.

Basidiomata—Gelatinous when fresh, orange-brown to reddish brown, usually solitary, sometimes caespitose, sessile or substipitate; pileus discoid or auriculate, usually with lobed margin, projecting up to 8.5 cm, 1–5 mm thick, 0.4–0.7 mm thick when dry; upper surface pilose, usually without folds, becoming mouse grey to greyish brown upon drying; hymenophore surface smooth, sometimes with several folds, becoming almost black upon drying.

Internal features—Medulla present in the middle of the cross-section; crystals absent; abhymenial hairs with a slightly swollen base, hyaline, thick-walled, with a narrow lumen, sometimes septate, apical tips acute or obtuse, single or tufted, 75–135(–150) × 5–8 µm; hyphae with clamp connections, 0.5–4.5 µm in diam in KOH; basidia clavate, transversely 3-septate, with oil guttules, 70–103 × 4–7 µm, sterigmata rarely observed; cystidioles absent.

Spores—Basidiospores allantoid, hyaline, thin-walled, smooth, usually with one to three large guttules, IKI–, CB–, (14–)15–18.5(–20) × 5.8–6.2(–6.5) µm, L = 16.22 µm, W = 6.0 µm, Q = 2.57–2.96 (n = 90/3). 

Distribution—China.

Notes—*Auricularia tibetica* is characterized morphologically by the thick basidiomata with obvious lobed margin and ecologically by the growth on gymnosperm wood. *Auricularia americana* is another species in the genus growing on gymnosperm species, and it can be distinguished by shorter basidia (55–71 × 4–5 µm vs. 70–103 × 4–7 µm) and thinner basidiospores (14–16.5 × 4.5–5.4 µm vs. 15–18.5 × 5.8–6.2 µm). Phylogenetically, *A. tibetica* forms a single lineage with strong support and intraspecific genetic variations exist among samples of the species (Figure 1 and Figure 2). The morphological description of *A. tibetica* taken from Wu et al. [25].

Specimens examined—China. Shanxi Province, Ningwu County, Qiutiangou Forest Farm, on dead tree of *Picea*, 7 August 2013, Y.C. Dai, *Dai 13336* (BJFC 014799). Sichuan Province, Daocheng County, Sandui Village, 18 July 2015, Y.C. Dai, *Dai 15604* (BJFC 019708). Xizang Autonomous Region (Tibet), Linzhi County, Kadinggou Park, on fallen gymnosperm trunk, 23 September 2014, B.K. Cui, *Cui 12337* (BJFC 017251); Sejilashan, on fallen trunk of *Abies*, 23 September 2014, B.K. Cui, *Cui 12266* (BJFC 17180), *Cui 12267* (BJFC 017181, holotype), *Cui 12268* (BJFC 017181). Yunnan Province, Yulong County, Laojunshan National Park, on dead tree of *Abies*, 31 August 2015, Y.C. Dai, *Dai 15734* (BJFC 019838); 1 September 2015, Y.C. Dai, *Dai 15752* (BJFC 019856).

(35) *Auricularia tremellosa* (Fr.) Pat. Figure 10l and Figure 42.

≡*Laschia tremellosa* Fr., Summa vegetabilium Scandinaviae 2: 325, 1849.

Basidiomata—Gelatinous when fresh, clay pink to clay buff, solitary or caespitose, sessile or substipitate; pileus auriculate with slightly lobed margin, projecting up to 6 cm, 0.5–1.5 mm thick, 0.08–0.2 mm thick and cinnamon to clay buff when dry; upper surface pilose, sometimes with a few folds; hymenophore surface conspicuously porose-reticulate to almost poroid especially when dry.

Internal features—Medulla present or absent; crystals absent; abhymenial hairs with a slightly swollen base, hyaline, thick-walled, with a wide or narrow septate lumen, apical tips acute or obtuse, single, 39–80 × 5–7 µm; hyphae with clamp connections and simple septa, 1–6.5 µm in diam in KOH; basidia clavate, transversely 3-septate, with oil guttules, 36–42 × 5–5.5 µm, sterigmata rarely observed; cystidioles absent.

Spores—Basidiospores allantoid, hyaline, thin-walled, smooth, usually with one or two large guttules, IKI–, CB–, (10–)10.2–12(–12.5) × 4–5 µm, L = 11.17 µm, W = 4.39 µm, Q = 2.5–2.59 (n = 60/2).

Distribution—Mexico and tropical South America.

Notes—*Auricularia tremellosa* was originally described from Mexico and then considered as synonym of *A. delicata* based on morphology by Lowy and Kobayasi [12,17]. However, in the present study, we accept *A. tremellosa* as an independent species because it is different from *A. delicata sensu stricto* not only in morphology but also in the phylogenies (Figure 1 and Figure 2). Six Brazilian specimens were studied and their morphological characters fit the description of *A. tremellosa. Auricularia tremellosa* differs from *A. delicata* by its poroid hymenophore, smaller basidia (36–42 × 5–5.5 µm vs. 48–65 × 4–6 µm) and distribution in Mexico and tropical South America. In the phylogeny, *A. tremellosa* is very distantly related to *A. delicata* (Figure 1 and Figure 2).

Specimens examined—Brazil. Acre, Sena Madureira, Floresta Nacional de São Francisco, on angiosperm wood, 16 January 2016, A. Meiras-Ottoni, *AMO 557* (URM 93424), Floresta Nacional do Macauã, on angiosperm wood, 29 January 2016, A. Meiras-Ottoni, *AMO 589* (URM 93425), Reserva Extrativista do Cazumbá-Iracema 30 January 2016, A. Meiras-Ottoni, *AMO 667* (URM 93426); Amazonas, Manaus, on rotten angiosperm wood, 12 May 2017, Y.C. Dai, *Dai 17415* (BJFC 026426), *Dai 17419* (BJFC 026428). Goiás, Pirenópolis, RPPN Vagafogo, on rotten angiosperm wood, 31 December 2016, R.L.M. Alvarenga, *RLMA 370* (URM 93427).

(36) *Auricularia villosula* Malysheva Figure 5k–l and Figure 43.

Basidiomata—Gelatinous when fresh, fawn to fuscous or buff yellow to orange yellow, solitary or caespitose, sessile or substipitate; pileus auriculate or petaloid, sometimes with lobed margin, projecting up to 5 cm, 1–2 mm thick, 0.08–0.3 mm thick and greyish brown to vinaceous brown when dry; upper surface pilose, sometimes with folds; hymenophore surface with obvious folds.

Internal features—Medulla absent; crystals absent; abhymenial hairs with a slightly swollen base, hyaline, thick-walled, with a narrow lumen, sometimes septate, apical tips acute or obtuse, single, 40–90 × 4.5–6 µm; hyphae with clamp connections, 1–3 µm in diam in KOH; basidia clavate, transversely 3-septate, with oil guttules, 40–61 × 4–5 µm, sterigmata rarely observed; cystidioles absent.

Spores—Basidiospores allantoid, hyaline, thin-walled, smooth, usually with one or two large guttules, IKI–, CB–, (12.6–)13–15.5(–16) × 5–6.1(–6.3) µm, L = 14.38 µm, W = 5.57 µm, Q = 2.39–2.53 (n = 90/3).

Distribution—Asia.

Notes—*Auricularia villosula* is characterized by its fawn and usually thin basidiomata when dry, and obvious folds on the hymenophore. It is easily confused with *A. heimuer* in the field, but it has larger basidiospores (13–15.5 × 5–6.1 µm vs. 11–13 × 4–5 µm) and is only distantly related to *A. heimuer* in the phylogenies (Figure 1 and Figure 2). *Auricularia villosula* has a wide distribution and is cultivated in China [23], but it was considered as *A. heimuer* in commercial markets.

Specimens examined—China. Anhui Province, Chuzhou, Shahe, on fallen trunk of *Paulownia*, 15 May 2009, B.K. Cui, *Cui 6760* (BJFC 004614). Hunan Province, Changsha, Yuelushan, on fallen angiosperm trunk, 10 August 2015, Y.C. Dai, *Dai 15994* (BJFC 020094). Inner Mongolia Autonomous Region, Chifeng, on fallen trunk of *Sophora*, 12 May 2014, T.Z. Liu, *CFSZ 10473* (CFSZ). Jiangxi Province, Fuzhou, on living tree of *Cinnamomum*, 5 August 2013, Y.C. Dai, *Dai 13453* (BJFC 014914); on living tree of *Cercis*, 21 July 2013, Y.C. Dai, *Dai 13450* (BJFC 014911). Sichuan Province, Bazhong County, Bazhong Park, on dead tree of *Ficus*, 31 May 2010, Y.C. Dai, *Dai 11633* (BJFC 009254); Yajiang County, on angiosperm trunk, 23 July 2015, Y.C. Dai, *Dai 15605* (BJFC 019709). Russia. Primorye Territory, 20 June 1991, Mikulin, (LE 296422); Ussuriyskiy Nature Reserve, on fallen angiosperm trunk, 3 July 1990, Govorova, (VLA M-11348), 30 July 1992, Govorova, (VLA M-11291), 15 August 2011, V. Malysheva, (LE 262989).

(37) *Auricularia xishaensis* L.J. Li Figure 20f and Figure 44.

Basidiomata—Gelatinous when fresh, cinnamon to reddish brown, solitary or caespitose, sessile; pileus discoid or auriculate, margin entire, projecting up to 3 cm, 0.2–0.5 mm thick, 0.14–0.16 mm thick when dry; upper surface pilose, sometimes with small folds, becoming vinaceous grey to mouse grey upon drying; hymenophore surface with obvious folds, becoming light vinaceous grey to ash-grey upon drying.

Internal features—Medulla present; crystals present, abundantly scattered throughout the cross-section; abhymenial hairs with a slightly swollen base, hyaline, thick-walled, with a narrow lumen, apical tips acute or obtuse, single, sometimes tufted, 80–120 × 4–5 µm; hyphae with clamp connections, 1–3 µm in diam in KOH; basidia clavate, transversely 3-septate, with oil guttules, 50–72 × 5–7 µm, sterigmata rarely observed; cystidioles absent.

Spores—Basidiospores not observed in the type but were reported as measuring 15–16 × 6–7 µm by Li (1984).

Distribution—China.

Notes—*Auricularia xishaensis* has very thin basidiomata when fresh like *A. fibrillifera*, but it has distinctly larger basidiospores than those of *A. fibrillifera* (11–14 × 4–5 µm). In addition, *A. xishaensis* is easily identified by obvious folds and white particles on the hymenophore surface and plenty of crystals scattered throughout the cross-section. We failed to extract DNA because the type is in poor condition, so the placement of *A. xishaensis* in *Auricularia* remain uncertain. 

Specimen examined—China. Hainan Province, Sansha, East of Xisha Islands, on rotten branch of *Messerschmidia*, 11 April 1982, L.J. Li, *HPNHM 1338* (BJM, holotype).

## 4. Discussion

Currently 37 *Auricularia* species belonging to five species complexes (*A. auricula-judae*, *A. cornea*, *A. delicata*, *A. fuscosuccinea* and *A. mesenterica*) are recognized based on morphological and/or molecular data. DNA sequences of 31 *Auricularia* species were obtained, while the samples of another six species (*A. eburnea*, *A. eminii*, *A. hainanensis*, *A. minor*, *A. papyracea* and *A. xishaensis*) were poorly dried or contaminated during sampling and DNA extraction failed. The molecular data of these six species will tentatively be accessed when new samples from their type locality are collected in the future.

*Auricularia* was considered a polyphyletic group based on nLSU sequences [27,29], but it is shown to be monophyletic in our phylogeny based on the concatenated ITS+nLSU dataset (Figure 1), which confirms the results of recent phylogenetic analyses [30,31]. The phylogenetic analyses inferred from both datasets result in similar topology of *Auricularia* with three clades (Clade A, Clade B and Clade C). Clade A is somewhat complicated and includes the *A. cornea*, the *A. delicata* and the *A. fuscosuccinea* complexes, while the *A. auricula-judae* and the *A. mesenterica* complexes are nested in Clade B and Clade C, respectively (Figure 1 and Figure 2). Clade A includes 16 species in both phylogenies. Those taxa cluster in three groups unrelated to the species complexes in the phylogeny based on the concatenated ITS+nLSU dataset. Group I and Group III form two distinct clades with strong support in the phylogenies (Figure 1 and Figure 2). For the convenience of discussion, species in Group II in the phylogeny based on the concatenated ITS+nLSU dataset (Figure 1) are still regarded as one group, although these species do not cluster in one clade in the phylogeny based on the concatenated ITS+nLSU+rpb1+rpb2 dataset (Figure 2).

Group I is divided into two small clades which include two species respectively (*A. cornea* and *A. novozealandica*) in the *A. cornea* complex and five species (*A. sinodelicata*, *A. delicata*, *A. australiana*, *A. conferta*, and *A. lateralis*) in the *A. delicata* complex. *Auricularia cornea* was originally collected in the Marianna Islands [17], and it has been widely reported from Asian and Pacific areas [12,20,49,50,51]. Although the lineage of *A. cornea* is not strongly supported and includes some small lineages in both phylogenies, those small lineages are not stable in both phylogenies (Figure 1 and Figure 2). In addition, samples with mostly similar macro-morphology are distantly related, e.g., Wu 07 (Figure 15g) and Dai 14876 (Figure 15h), and samples with slightly different morphology are closely related, e.g., Dai 15336 (Figure 15e) and Cui 7517 (Figure 15f). However, all these samples have similar micro-morphology and share the characteristics of *A. cornea* [17,20], so they are treated as *A. cornea* in the present study. These variations in macro-morphology and molecular data may be due to a wide distribution in Africa, North and South America, Asia, and Europe. The new species *A. novozealandica* is closely related to *A. cornea* in our phylogenies (Figure 1 and Figure 2) and macro-morphologically similar to a few specimens of *A. cornea*, but it has distinctly larger basidiospores (Table 2) and distribution restricted to New Zealand so far. The lineage of *A. australiana* from Australia was defined as *A. delicata* clade I in Looney et al. [20], and it is not supported in our phylogeny based on the concatenated ITS+nLSU dataset (Figure 1), but it is strongly supported in our phylogeny based on the concatenated ITS+nLSU+rpb1+rpb2 dataset (Figure 2). *Auricularia conferta* also from Australia has dense thick folds on the hymenophore surface and wider hairs than *A. australiana* (Table 3) with a wide and regular septate lumen. *Auricularia sinodelicata* and *A. lateralis* form two distinct lineages separated from *A. delicata* in the phylogenies (Figure 1 and Figure 2), and they are different from *A. delicata* in morphology. Therefore, the four species are considered as new species in the *A. delicata* complex in the present study.

Group II includes seven species: *A. fuscosuccinea*, *A. subglabra*, *A. scissa*, *A. pilosa*, *A. nigricans, A. camposii*, and *A. tremellosa*. The lineages of the species in this group are strongly supported, but their morphologies are not corresponding to their phylogenetic relations. *Auricularia fuscosuccinea*, originally collected in Cuba [17], resembles *A. fibrillifera* and *A. thailandica* because of the red-brown fresh basidiomata and all three species belong to the *A. fuscosuccinea* complex (Figure 20). *Auricularia subglabra, A. scissa*, *A. pilosa*, and *A. tremellosa* have more or less fleshy basidiomata, pilose pileal surface and porose-reticulate hymenophore, and they belong to the *A. delicata* complex (Figure 10). *Auricularia camposii* and *A. nigricans* have more or less leathery basidiomata, tomentose to hispid pileal surface and smooth hymenophore and they belong to the *A. cornea* complex (Figure 15). The new species, *A. pilosa*, from Ethiopia is described here based on only one small specimen (LWZ20190421-7), but the specimen together with JMH 45 from Tanzania form one lineage with high support in the phylogeny (Figure 1). So, these two specimens are recognized as the new species. *Auricularia camposii* is morphologically similar to *A. cornea*, but it has tomentose upper surface and forms one lineage distantly from *A. cornea* in the phylogenies (Figure 1 and Figure 2). The lineage of *A. tremellosa* was defined as the “*A. delicata* clade II” in Looney et al. [20], and it is recognized as the known species *A. tremellosa* originally described from Mexico [53].

Group III includes two species: *A. fibrillifera* and *A. thailandica*. Both species have thin basidiomata when fresh becoming fragile when dry and are distributed in subtropical to tropical areas of Africa and Asia. The Chinese samples were previously considered as “*A. fuscosuccinea”* because of the similar morphology [42,44]. In the present study, we still list *A. fibrillifera* and *A. thailandica* as members of the *A. fuscosuccinea* complex although *A. fuscosuccinea* is only distantly related to *A. fibrillifera* and *A. thailandica* in the phylogenies (Figure 1 and Figure 2).

The species from the *A. cornea* complex differ from other species in the genus by the smooth hymenophore surface, densely pilose, tomentose or hispid upper surface, long abhymenium hairs (>150 µm) and the presence of a medulla. Another two species, *A. eburnea* and *A. eminii,* also have the afore-mentioned characteristics, and we think they belong to the same complex even if the molecular data is lacking. The main characteristics of the seven species in the *A. cornea* complex are summarized in Table 2.

The *Auricularia delicata* complex is characterized by its porose-reticulate hymenophore surface, the absence of crystals, usually short abhymenium hairs (<100 µm) and small basidiospores. A synoptic table of a comparison of species in the complex is provided in Table 3.

The species from the *A. fuscosuccinea* complex are different from other species in the genus by the thin and fragile basidiomata when dry, pilose upper surface and short abhymenium hairs (<150 µm). The main characteristics of members in the complex are summarized in Table 4. *A**uricularia papyracea* and *A. xishaensis* have very thin dry basidiomata and are similar to *A. fuscosuccinea*. We temporarily treat them as members of the *A. fuscosuccinea* complex although the phylogenetic relations among the three species are unknown.

Clade B includes seven species belonging to the *Auricularia auricula-judae* complex (Figure 1 and Figure 2). Samples of these seven species in the complex are nested in seven lineages with high support in our phylogenies, Intraspecific genetic variations are obviously present among samples of these taxa which is probably the reason why *A. auricula-judae* has considerable variability in the size of the basidiospores. *Auricularia auricula-judae* was originally described from Europe and was recognized as a species complex by Looney et al. [20]. They also confirmed that the so-called “*A. auricula-judae”* in the USA was actually *A. americana,* but two sublineages, represented by samples, on angiosperm and gymnosperm, respectively, clustered in their phylogeny based on sequences of ITS and rpb2. Later, based on a comprehensive phylogeny, Wu et al. considered that samples on gymnosperm represented *A. americana sensu stricto* because *A. americana* was originally described from gymnosperm [23], and samples on angiosperm were described as *A. angiospermarum*. Subsequently, *A. heimuer, A. minutissima, A. tibetica* and *A. villosula* were described based on phylogenetic analyses [5,20,22].

The species from the *A. auricula-judae* complex are characterized by usually reddish brown basidiomata, a pilose upper surface with short abhymenium hairs (<150 µm), and mainly smooth hymenophores (Figure 5). Although we do not know the phylogenetic relations of *A. hainanensis* and *A. minor* with other members in the *A. auricula-judae* complex, we list them in this complex because they share these afore-mentioned morphological features. A synoptic table of a comparison on species in the complex is provided in Table 5.

Clade C includes eight species belonging to the *A. mesenterica* complex (Figure 1 and Figure 2). All species in this complex have similar macro-morphology, and were considered *A. mesenterica* previously [12,20]. Among them, *A. asiatica, A. brasiliana* and *A. orientalis* were recently described [25,26], while *A*. *africana*, *A. srilankensis* and *A. submesenterica* are described as new species in the present study. These eight species form eight independent lineages with high support in the phylogeny based on the concatenated ITS+nLSU dataset except for *A. submesenterica* (Figure 1), but six samples of *A. submesenterica* form a single lineage with high support in the phylogeny based on the concatenated ITS+nLSU+rpb1+rpb2 dataset (Figure 2). The lineage of *A. africana* from Africa is strongly supported and distant from other species of the complex in both phylogenies. The lineage of *A. srilankensis* is closely related to *A. asiatica*, but *A. srilankensis* morphologically differs from *A. asiatica* by the usually resupinate basidiomata (Figure 3h), larger basidia and longer basidiospores (Table 6).

The species from the *A. mesenterica* complex are distinctly different from other species of the genus by resupinate to effused-reflexed basidiomata, a usually concentrically zoned upper surface (Figure 3), the absence of medulla and usually inflated hyphae in KOH. Characteristics of these seven species in the complex are summarized in Table 6.

The morphological characters of the members of the *A. auricula-judae* and the *Auricularia*
*mesenterica* complexes totally correspond to molecular data, but not so in the members of the *A. cornea*, the *A. delicata* and *A. fuscosuccinea* complexes. Species in these three complexes may be at a stage of speciation. So far, the most important morphological characters to delimit a species in the genus are the presence or absence of the medulla, the texture, length and diameter of hairs, and the lengths and widths of basidia and basidiospores, which is more or less the same as observed in previous studies [12,17,20]. Morphological features displaying evidence for evolution and phenotypic plasticity are worthy of being studied in the future.

In addition to the molecular data, the present study provides data about host and geographical distribution of 277 samples from 35 countries in Asia, Europe, North and South America, Africa, and Oceania. The two species growing on gymnosperm wood, *A. americana* and *A. tibetica*, cluster in the *A. auricula-judae* complex in the phylogeny, and they form two independent lineages separated from other species growing on angiosperms. Thus, host may also be important in the taxonomy of *Auricularia* as with some other wood-inhabiting genera [59,60]. Regarding the geographical distribution, most species are distributed in the same continent, while a few species are widely distributed in different continents, e.g., *A. cornea* (Figure 1), which, however, remains as an unresolved lineage. The species in Group II are mostly distributed in the Americas, while other species are mainly distributed in Asia (Figure 1). Therefore, the host and geographical distribution probably have an influence on speciation of *Auricularia*.


**A key to species accepted in *Auricularia***
1. Basidiomata resupinate to effused-reflexed, upper surface usually concentrically zoned with canescent zones and dark bands or wide whitish zones and thin black bands; hyphae with a distinct lumen in KOH21. Basidiomata sessile or substipitate, upper surface usually without concentric zones or concentrically zoned; hyphae without a lumen in KOH92. Basidia usually <50 µm long32. Basidia usually >50 µm long53. Upper surface hispid, distinctly and concentrically zoned with canescent zones and dark bands; hyphae <7.5 µm in diam in KOH
*A. submesenterica*
3. Upper surface villose or tomentose, distinctly and concentrically zoned with wide whitish zones and thin black bands; hyphae usually > 7.5 µm in diam in KOH44. Upper surface villose, distinctly and concentrically zoned with wide whitish zones and thin black bands in the central pileal surface; crystals present; distribution: Asia
*A. asiatica*
4. Upper surface tomentose, distinctly and concentrically zoned with wide whitish zones and thin black bands throughout pileal surface; crystals absent; distribution: Brazil
*A. brasiliana*
5. Distributed in Africa or Oceania65. Distributed in Europe or Asia76. Abhymenial hairs 300–500 µm long, basida 50–80 × 4.5–6 µm, basidiospores 4.9–5.4 µm wide; distribution: Kenya and Uganda
*A. africana*
6. Abhymenial hairs 400–800 µm long, basida 50–72 × 5–7.5 µm, basidiospores 5–6 µm wide; distribution: Australia and Zambia
*A. pusio*
7. Hyphae up to 14 µm in diam in KOH, basidiospores 14–17 × 4.7–5.2 µm; distribution: Europe and Uzbekistan
*A. mesenterica*
7. Hyphae up to 7 µm in diam in KOH, basidiospores 12–14.7 × 4.5–6 µm; distribution: Asia88. Abhymenial hairs usually <900 µm long, cystidioles absent; distribution: Sri Lanka
*A. srilankensis*
8. Abhymenial hairs usually >900 µm long, cystidioles present; distribution: China
*A. orientalis*
9. Upper surface concentrically zoned; abhymenial hairs up to 20 µm long
*A. papyracea*
9. Upper surface without concentric zones; abhymenial hairs usually >20 µm long1010. Hymenophore surface porose-reticulate1110. Hymenophore surface smooth or with several folds1911. Schizomedulla present1211. Schizomedulla absent1312. Upper surface pilose; abhymenial hairs 40–80 × 5–10 µm, basidia 40–60 × 4–5 µm
*A. scissa*
12. Upper surface scantly pilose; abhymenial hairs 20–45 × 5–6 µm, basidia 30–45 × 3.5–4.5 µm
*A. subglabra*
13. Upper surface distinctly pilose, abhymenial hairs usually >100 µm long1413. Upper surface pilose or scantly pilose, abhymenial hairs usually <100 µm long1514. Abhymenial hairs with a slightly swollen center, basidia 50–70 × 5–6.5 µm
*A. lateralis*
14. Abhymenial hairs with a slightly swollen base, basidia 35–45 × 4–5.5 µm 
*A. pilosa*
15. Basidia usually >45 µm long; distribution: Western Africa or Australia1615. Basidia usually <45 µm long; distribution: Mexico and tropical South America or China1816. Distributed in Western Africa
*A. delicata*
16. Distributed in Australia1717. Hymenophore surface with dense and thick folds, abhymenial hairs 8–15 µm wide
*A. conferta*
17. Hymenophore surface without dense and thick folds, abhymenial hairs 7–11 µm wide
*A. australiana*
18. Hymenophore surface conspicuously porose-reticulate to almost poroid especially when dry, with more or less poroid folds; distribution: Mexico and tropical South America
*A. tremellosa*
18. Hymenophore surface conspicuously porose-reticulate especially when dry, with more or less radial folds; distribution: China
*A. sinodelicata*
19. Basidiomata more or less leathery, upper surface densely pilose, tomentose or hispid; abhymenial hairs usually >150 µm long2019. Basidiomata never leathery, upper surface pilose or scantly pilose; abhymenial hairs usually <150 µm long2520. Abhymenial hairs usually >1 cm long
*A. eminii*
20. Abhymenial hairs usually <1 cm long2121. Upper surface densely pilose2221. Upper surface densely tomentose or hispid2422. Basidia usually >70 µm long; distribution: New Zealand
*A. novozealandica*
22. Basidia usually <70 µm long; distribution: Africa, North and South America, Asia or Europe2323. Basidiomata usually reddish brown, sometimes white; cystal present, hyphae with simple septa, basidia 60–75 × 4–6 µm, basidiospores 13.8–16.5 × 4.5–6 µm
*A. cornea*
23. Basidiomata usually cream to honey-yellow; cystal absent, hyphae without simple septa, basidia 50–65 × 4–5 µm, basidiospores 15.6–18 × 5.5–7.5 µm
*A. eburnea*
24. Upper surface densely tomentose to hispid; abhymenial hairs 300–600 × 7–9 µm, basidia 50–60 long
*A. nigricans*
24. Upper surface densely tomentose; abhymenial hairs 120–250 × 6–7 µm, basidia 60–75 long
*A. camposii*
25. Upper surface scantly pilose2625. Upper surface pilose2926. Basidiomata fawn to fuscous when dry; basidiospores >9 µm long2726. Basidiomata mouse-grey to black when dry; basidiospores <9 µm long2827. Basidiomata soft gelatinous when fresh, hymenophore surface usually with folds; medulla indistinctly present, abhymenial hairs 60–100 × 10–20 µm
*A. fibrillifera*
27. Basidiomata gelatinous when fresh, hymenophore surface usually smooth; medulla present, abhymenial hairs 50–80 × 6–14 µm
*A. thailandica*
28. Basidiomata 0.6–0.7 mm thick when dry; abhymenial hairs 30–80 × 4–5 µm, basidia 70–80 long
*A. hainanensis*
28. Basidiomata 0.1–0.2 mm thick when dry; abhymenial hairs 90–120 × 6–10 µm, basidia 35–43 long
*A. minor*
29. Hymenophore surface with obvious white particles when dry; abundant crystals scattered throughout the cross-section
*A. xishaensis*
29. Hymenophore surface without obvious white particles when dry; crystals absent or a few scattered in the hymenium3030. Crystals present, scattered in the hymenium3130. Crystals absent3331. Basidia usually >65 µm long, basidiospores usually >15 µm long
*A. auricula-judae*
31. Basidia usually <65 µm long, basidiospores usually <15 µm long3232. Basidiomata fawn to cinnamon brown when fresh; abhymenium hairs 75–165 × 5–8 µm, basidiospores 4.3–5.2 µm wide; distribution: Tropical and subtropical America
*A. fuscosuccinea*
32. Basidiomata vinaceous grey to fuscous when fresh; abhymenium hairs 50–85 × 5–6 µm, basidiospores 5–6 µm wide; distribution: China and Russia
*A. minutissima*
33. Basidia usually >70 µm long, basidiospores usually >5.5 µm wide
*A. tibetica*
33. Basidia usually <70 µm long, basidiospores usually <5.5 µm wide3434. Basidiospores usually <13 µm long
*A. heimuer*
34. Basidiospores usually >13 µm long3535. Basidiomata fawn to fuscous when fresh, hymenophore surface with obvious folds
*A. villosula*
35. Basidiomata orange-brown to reddish brown or vinaceous brown when fresh, hymenophore surface smooth without folds3636. Basidiomata orange-brown to reddish brown when fresh, medulla present or absent, basidia 55–71 µm long, basidiospores 14–16.5 µm long; growth on gymnosperms
*A. americana*
36. Basidiomata vinaceous brown when fresh; medulla absent, basidia 46–63 µm long, basidiospores 13–15 µm long; growth on angiosperms
*A. angiospermarum*



## Figures and Tables

**Figure 1 jof-07-00933-f001:**
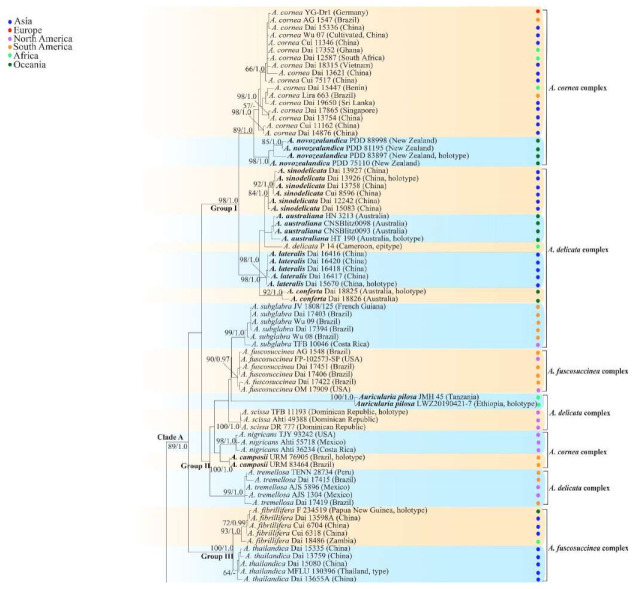
Maximum Likelihood (ML) tree illustrating the phylogeny of *Auricularia* based on the concatenated ITS+nLSU dataset. Branches are labeled with Maximum Likelihood bootstrap higher than 50%, and Bayesian Posterior Probabilities higher or equal to 0.90 respectively. The distribution of different specimens is marked by different colored dots.

**Figure 2 jof-07-00933-f002:**
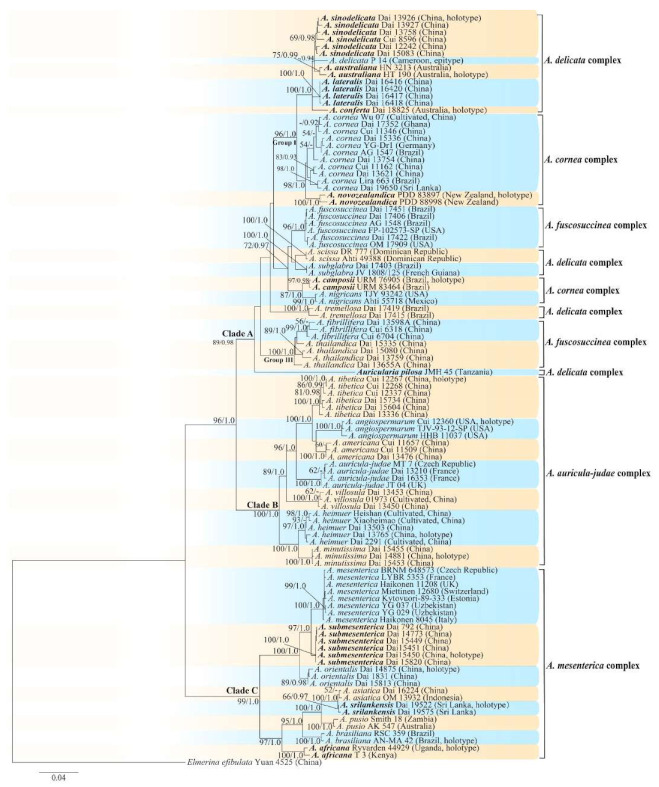
ML tree illustrating the phylogeny of *Auricularia* based on the concatenated ITS+nLSU+rpb1+rpb2 dataset. Branches are labeled with Maximum Likelihood bootstrap higher than 50%, and Bayesian Posterior Probabilities higher or equal to 0.90 respectively.

**Figure 3 jof-07-00933-f003:**
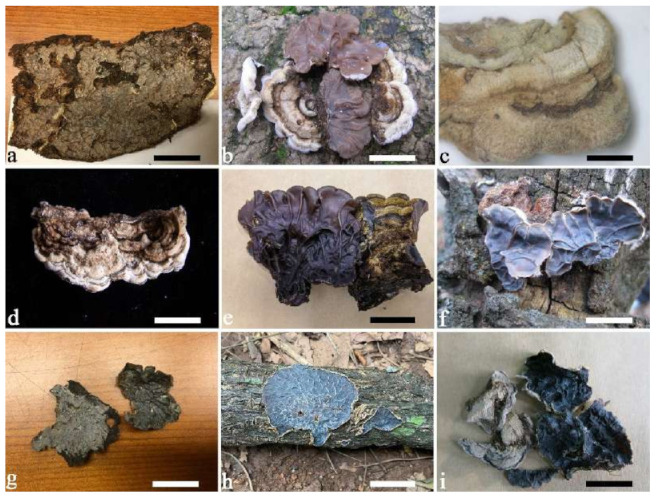
Basidiomata of the *Auricularia*
*mesenterica* complex: (**a**) *A. africana* (Ryvarden 44929); (**b**,**c**) *A. asiatica* (Dai 16149); (**d**) *A. brasiliana* (AN-MA 42); (**e**) *A. mesenterica* (LYBR 5353); (**f**) *A. orientalis* (Dai 14875); (**g**) *A. pusio* (AK 174); (**h**) *A. srilankensis*; (**i**) *A. submesenterica* (Dai 15450). Bars: (**a**) 1 cm; (**b**,**d**) 2.5 cm; (**c**) 5 mm; (**e**–**i**) 2 cm.

**Figure 4 jof-07-00933-f004:**
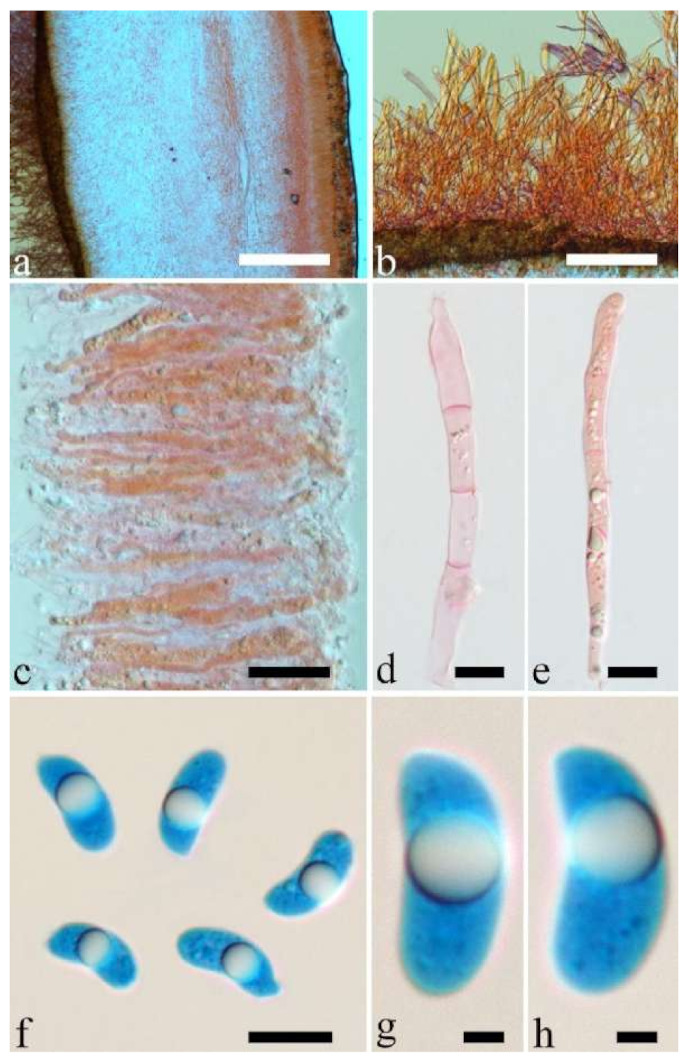
Microscopic structures of *Auricularia africana* (Ryvarden 44929, holotype). (**a**) Cross-section of a basidioma; (**b**) Abhymenial hairs; (**c**–**e**) Basidia and basidioles; (**f**–**h**) Basidiospores. Bars: (**a**) 300 μm; (**b**) 100 μm; (**c**) 20 μm; (**d**,**f**) 10 μm; (**g**,**h**) 2 μm.

**Figure 5 jof-07-00933-f005:**
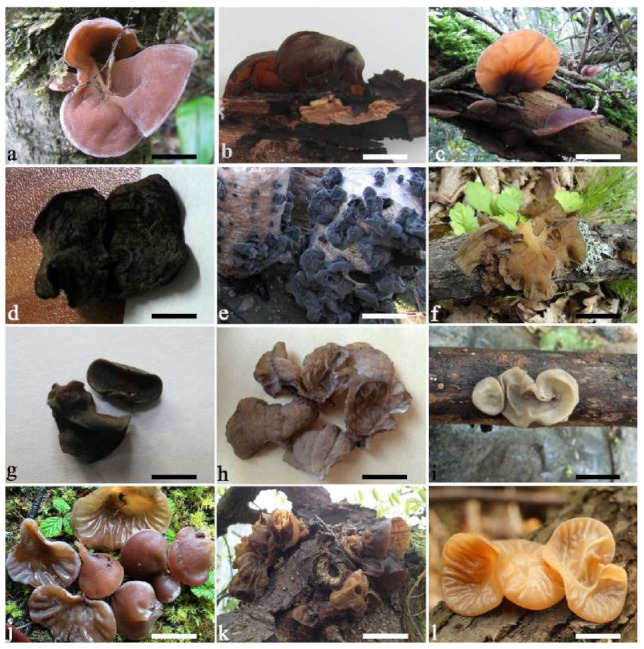
Basidiomata of the *Auricularia auricula-judae* complex: (**a**) *A. americana* (Dai 13461); (**b**) *A. angiospermarum* (HHB-11037); (**c**) *A. auricula-judae* (Dai 16353); (**d**) *A. hainanensis* (HPNHM 103); (**e**,**f**) *A. heimuer* (Dai 2291, Dai 13788); (**g**) *A. minor* (F-434); (**h**,**i**) *A. minutissima* (Dai 15452, Dai 15454); (**j**) *A. tibetica* (Dai 15752); (**k**,**l**) *A. villosula* (CFSZ 10473, Dai 15994). Bars: (**a**–**d**,**f**,**h**,**j**–**l**) 2 cm; (**e**) 5 cm; (**g**,**i**) 5 mm.

**Figure 6 jof-07-00933-f006:**
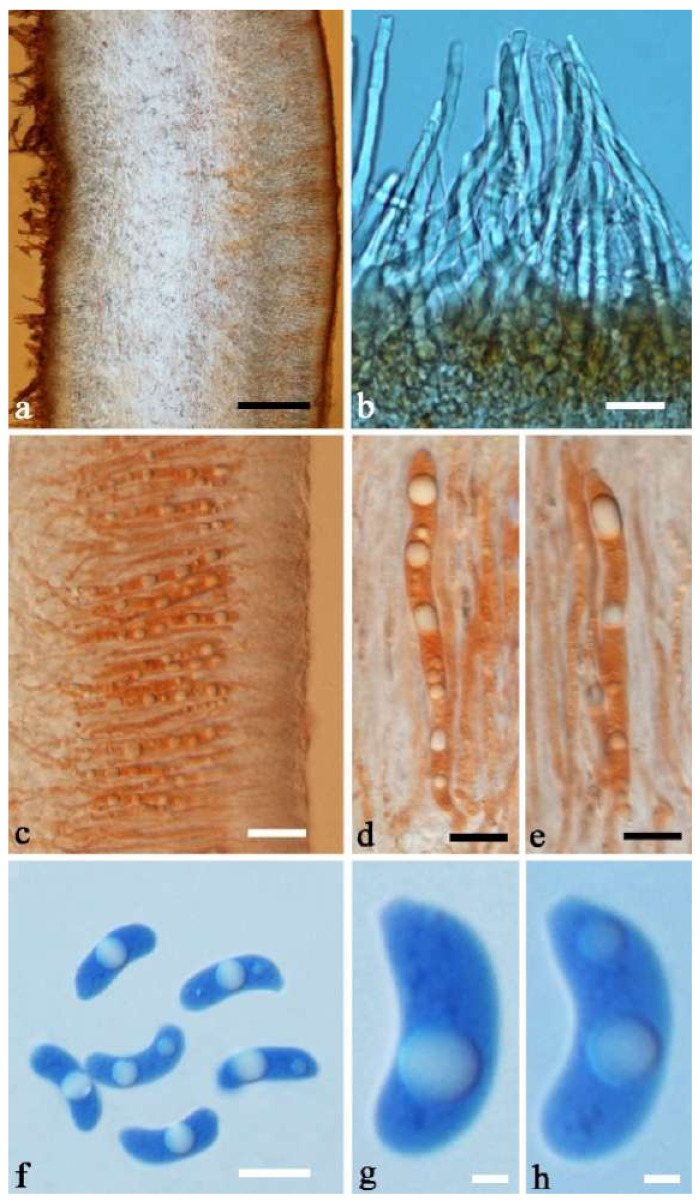
Microscopic structures of *Auricularia americana* (HHB-11374). (**a**) Cross-section of a basidioma; (**b**) Abhymenial hairs; (**c**–**e**) Basidia and basidioles; (**f**–**h**) Basidiospores. Bars: (**a**) 300 μm; (**b**,**c**) 20 μm; (**d**,**f**) 10 μm; (**g**,**h**) 2 μm.

**Figure 7 jof-07-00933-f007:**
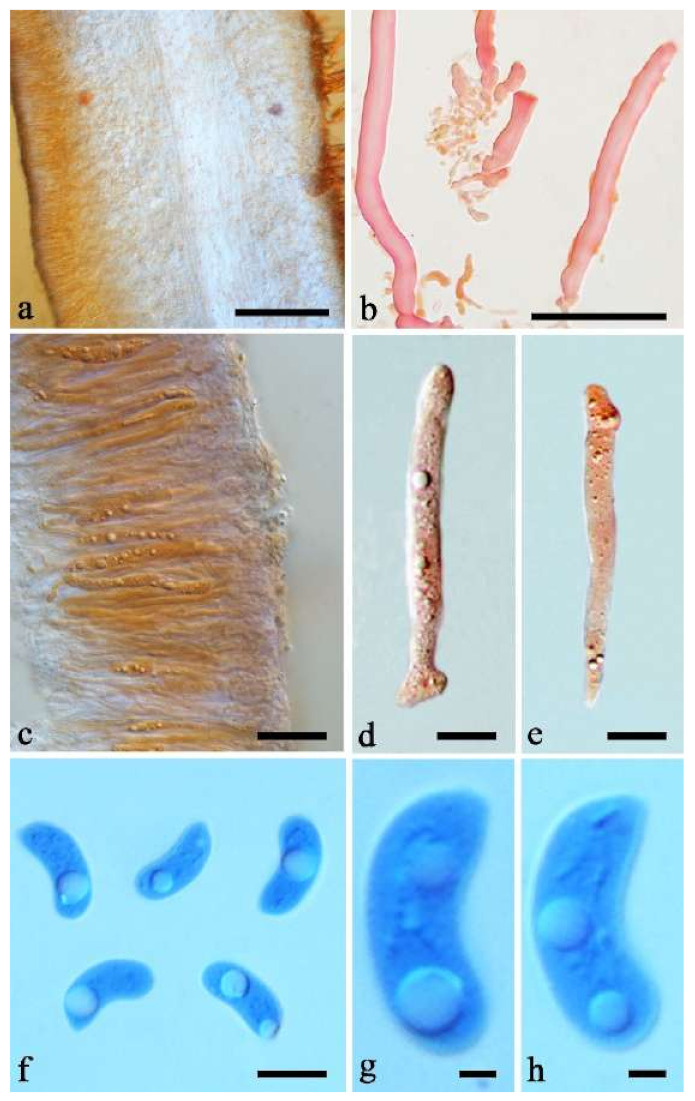
Microscopic structures of *Auricularia angiospermarum* (Cui 12360, holotype). (**a**) Cross-section of a basidioma; (**b**) Abhymenial hairs; (**c**–**e**) Basidia and basidioles; (**f**–**h**) Basidiospores. Bars: (**a**) 300 μm; (**b**,**c**) 30 μm; (**d**,**f**) 10 μm; (**g**,**h**) 2 μm.

**Figure 8 jof-07-00933-f008:**
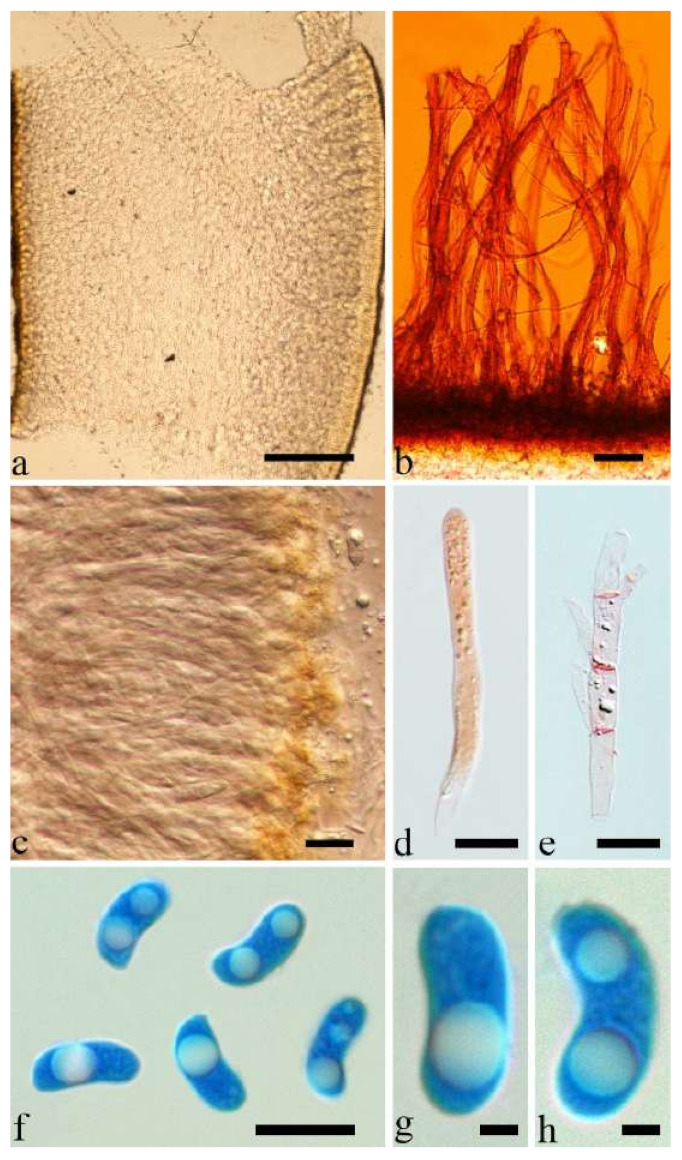
Microscopic structures of *Auricularia asiatica* (Dai 15285). (**a**) Cross-section of a basidioma; (**b**) Abhymenial hairs; (**c**–**e**) Basidia and basidioles; (**f**–**h**) Basidiospores. Bars: (**a**) 300 μm; (**b**) 100 μm; (**c**–**f**) 10 μm; (**g**,**h**) 2 μm.

**Figure 9 jof-07-00933-f009:**
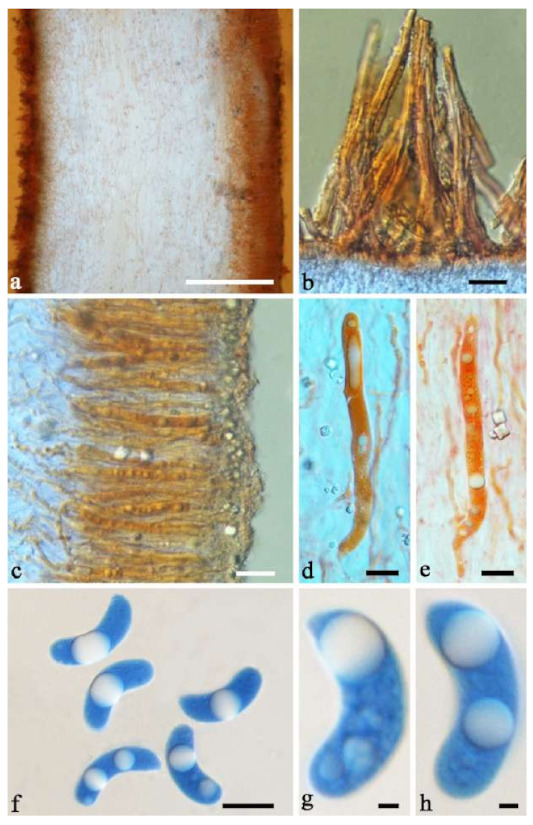
Microscopic structures of *Auricularia auricula-judae* (Dai 13189). (**a**) Cross-section of a basidioma; (**b**) Abhymenial hairs; (**c**–**e**) Basidia and basidioles; (**f**–**h**) Basidiospores. Bars: (**a**) 300 μm; (**b**,**c**) 20 μm; (**d**,**f**) 10 μm; (**g**,**h**) 2 μm.

**Figure 10 jof-07-00933-f010:**
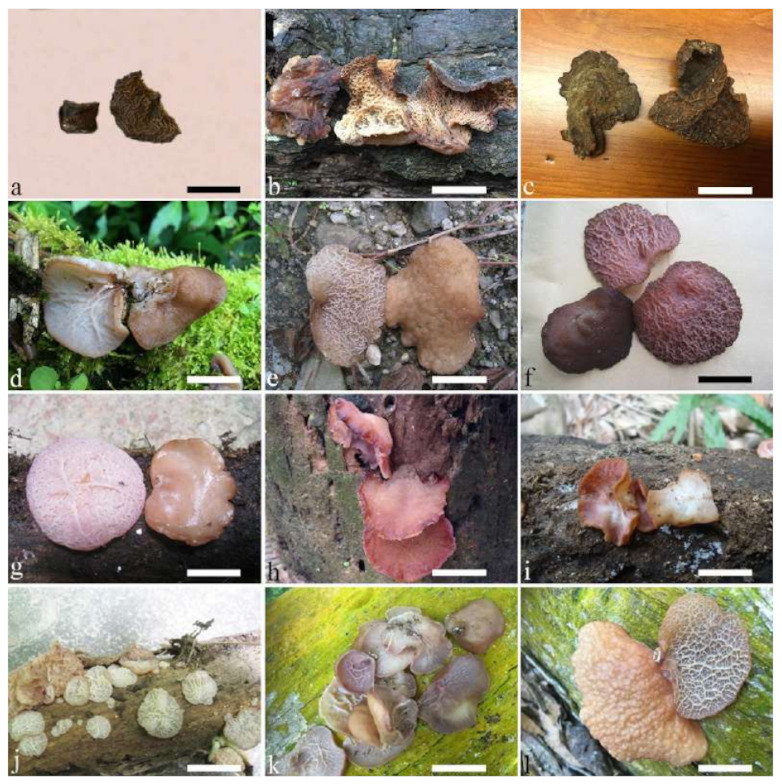
Basidiomata of the *Auricularia delicata* complex: (**a**) *A. australiana* (HT 190); (**b**) *A. conferta* (Dai 18825); (**c**) *A. delicata* (P 14); (**d**) *A. lateralis* (Dai 15670); (**e**–**j**) *A. sinodelicata* (Cui 12236, Dai 13593A, Dai 13748, Dai 13758, Dai 13919, Dai 13920); (**k**) *A. subglabra* (Dai 17417); (**l**) *A. tremellosa* (Dai 17415). Bars: (**a**–**l**) 2 cm.

**Figure 11 jof-07-00933-f011:**
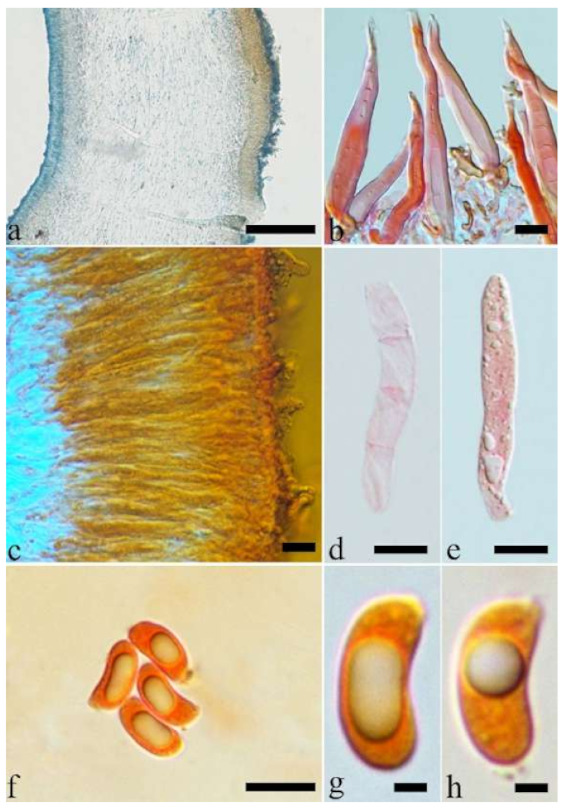
Microscopic structures of *Auricularia australiana* (HT 190, holotype). (**a**) Cross-section of a basidioma; (**b**) Abhymenial hairs; (**c**–**e**) Basidia and basidioles; (**f**–**h**) Basidiospores. Bars: (**a**) 200 μm; (**b**–**f**) 10 μm; (**g**,**h**) 2 μm.

**Figure 12 jof-07-00933-f012:**
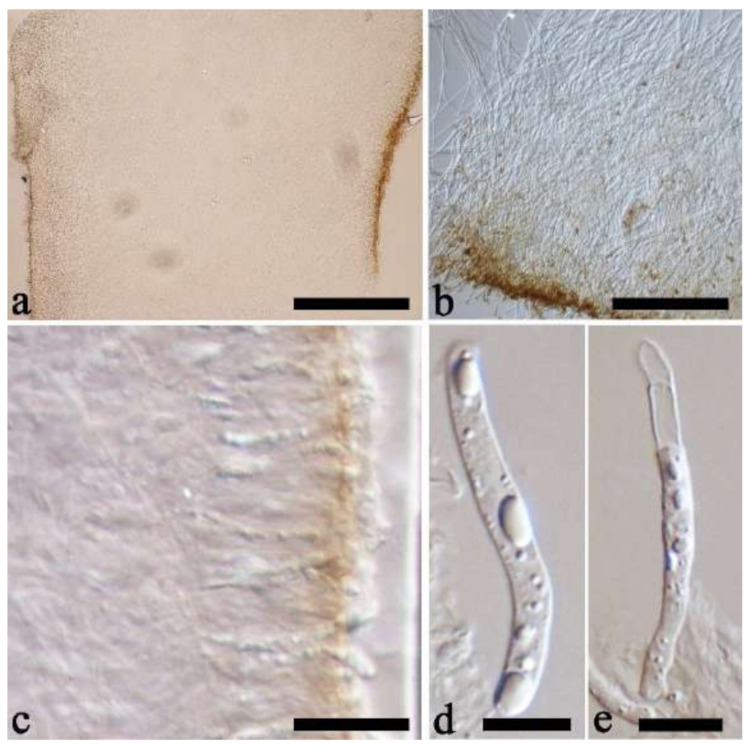
Microscopic structures of *Auricularia brasiliana* (AN-MA 42, holotype). (**a**) Cross-section of a basidioma; (**b**) Abhymenial hairs; (**c**–**e**) Basidia and basidioles. Bars: (**a**,**b**) 300 μm; (**c**) 30 μm; (**d**,**e**) 10 μm.

**Figure 13 jof-07-00933-f013:**
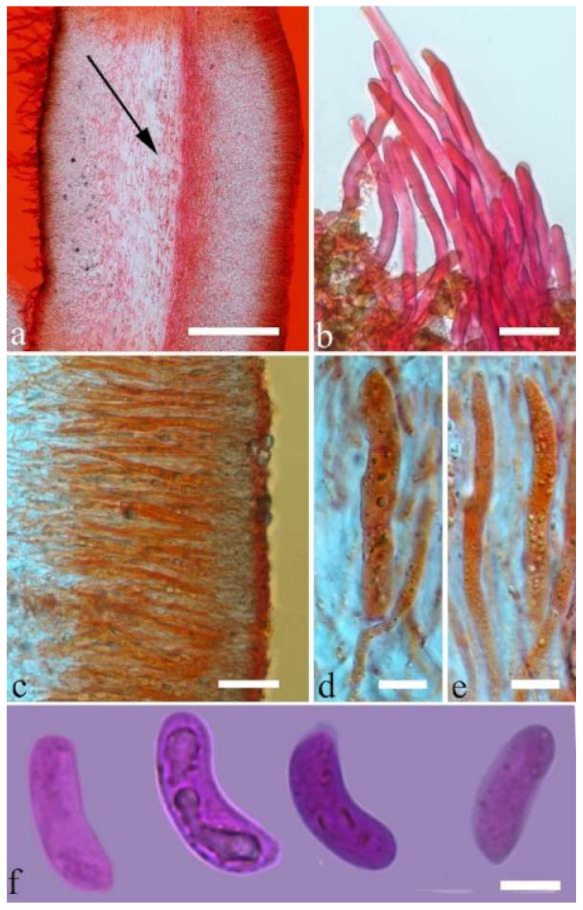
Microscopic structures of *Auricularia camposii* (URM 83464). (**a**) Cross-section of a basidioma (medulla is shown by the arrow); (**b**) Abhymenial hairs; (**c**–**e**) Basidia and basidioles in hymenium; (**f**) Basidiospores. Bars: (**a**) 300 μm; (**b**,**c**) 20 μm; (**d**,**e**) 10 μm; (**f**) 3 μm.

**Figure 14 jof-07-00933-f014:**
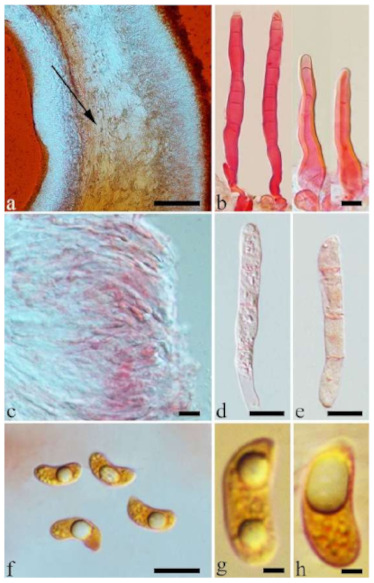
Microscopic structures of *Auricularia conferta* (Dai 18825, holotype). (**a**) Cross-section of a basidioma (medulla is shown by the arrow); (**b**) Abhymenial hairs; (**c**–**e**) Basidia and basidioles; (**f**–**h**) Basidiospores. Bars: (**a**) 200 μm; (**b**–**f**) 10 μm; (**g**,**h**) 2 μm.

**Figure 15 jof-07-00933-f015:**
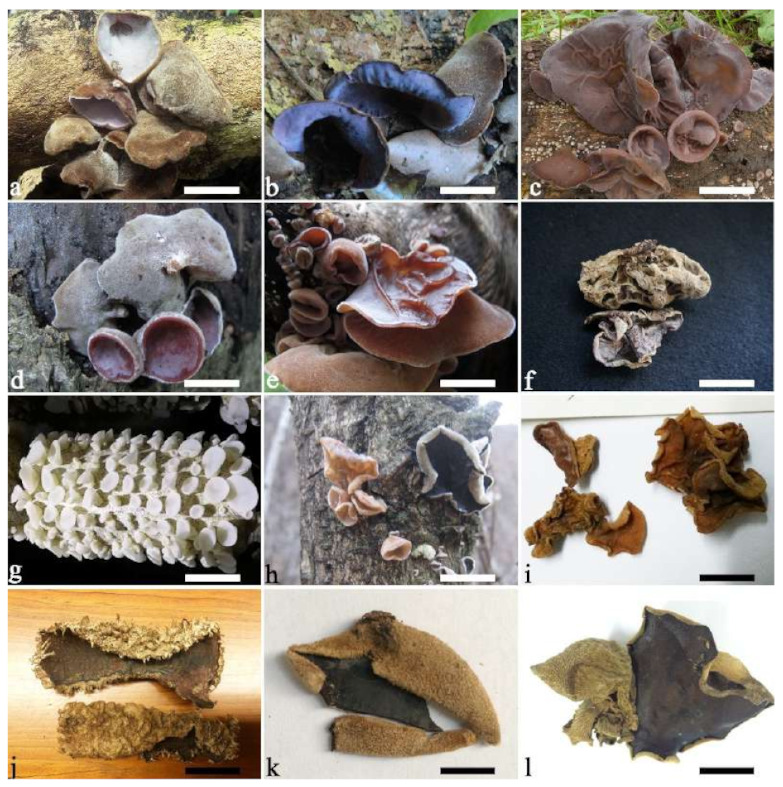
Basidiomata of the *Auricularia cornea* complex: (**a**–**h**) *A. cornea* (Dai 10845, Dai 13679, Dai 15447, Dai 15379, Dai 15336, Cui 7517, Wu 07, Dai 14876); (**i**) *A. eburnea* (HPNHW 1449); (**j**) *A. eminii* (K 27093); (**k**) *A. nigricans* (Ahti 36234); (**l**) *A. novozealandica* (PDD 83897). Bars: (**a**–**c**,**l**) 2.5 cm; (**d**–**f**,**h**–**j**) 2 cm; (**g**) 5 cm; (**k**) 5 mm.

**Figure 16 jof-07-00933-f016:**
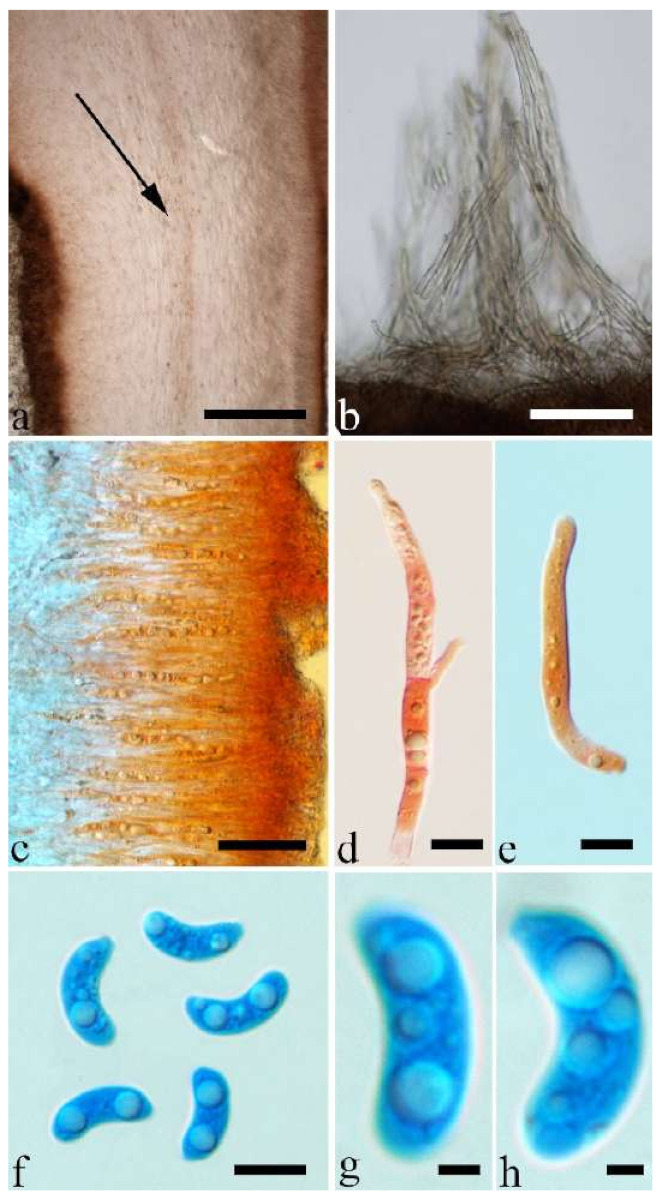
Microscopic structures of *Auricularia cornea* (Dai 12592). (**a**) Cross-section of a basidioma (medulla is shown by the arrow); (**b**) Abhymenial hairs; (**c**–**e**) Basidia and basidioles; (**f**–**h**) Basidiospores. Bars: (**a**) 300 μm; (**b**) 100 μm; (**c**) 20 μm; (**d**,**f**) 10 μm; (**g**,**h**) 2 μm.

**Figure 17 jof-07-00933-f017:**
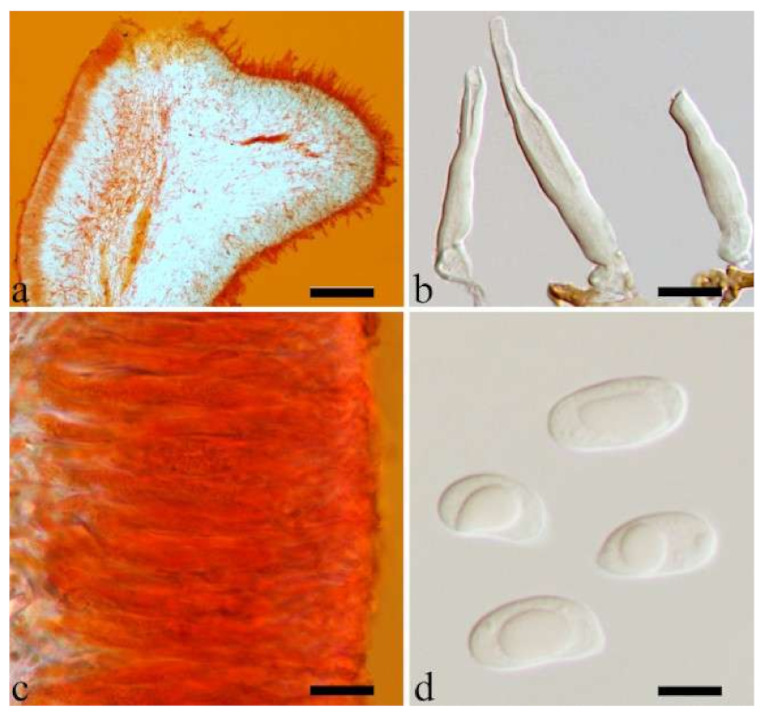
Microscopic structures of *Auricularia delicata* (P 14). (**a**) Cross-section of a basidioma; (**b**) Abhymenial hairs; (**c**) Basidia and basidioles in hymenium; (**d**) Basidiospores. Bars: (**a**) 100 μm; (**b**,**c**) 10 μm; (**d**) 5 μm.

**Figure 18 jof-07-00933-f018:**
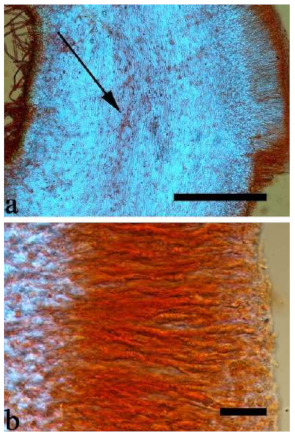
Microscopic structures of *Auricularia eburnea* (HPNHW 1449, holotype). (**a**) Cross-section of a basidioma (medulla is shown by the arrow); (**b**) Basidia and basidioles in hymenium. Bars: (**a**) 300 μm; (**b**) 20 μm.

**Figure 19 jof-07-00933-f019:**
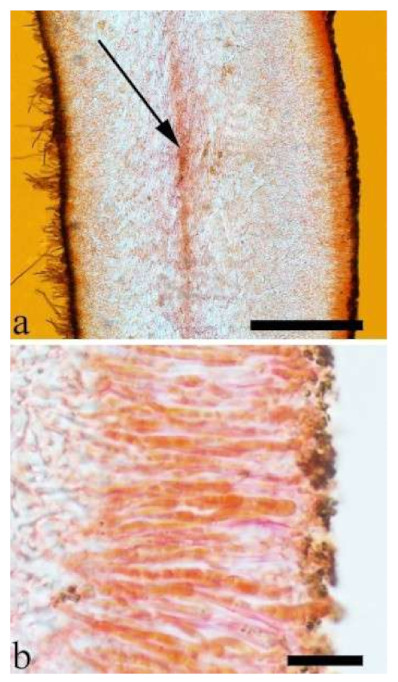
Microscopic structures of *Auricularia eminii* (F 19526). (**a**) Cross-section of a basidioma (medulla is shown by the arrow); (**b**) Basidia and basidioles in hymenium. Bars: (**a**) 300 μm; (**b**) 20 μm.

**Figure 20 jof-07-00933-f020:**
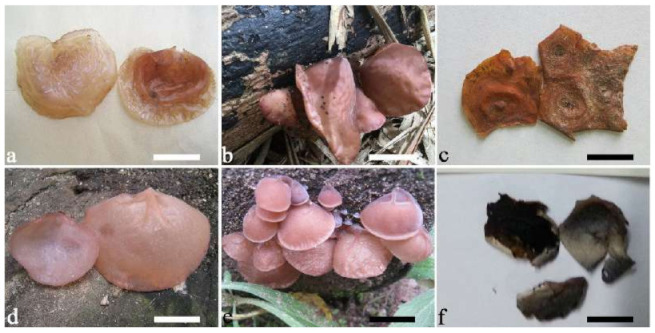
Basidiomata of the *Auricularia*
*fuscosuccinea* complex: (**a**) *A. fibrillifera* (Dai 13596A); (**b**) *A. fuscosuccinea* (Dai 17406); (**c**) *A. papyracea* (F-234520); (**d**,**e**) *A. thailandica* (Dai 13635A, Dai 13655A); (**f**) *A. xishaensis* (HPNHM 1338). Bars: (**a**,**b**,**d**–**f**) 2 cm; (**c**) 5 mm.

**Figure 21 jof-07-00933-f021:**
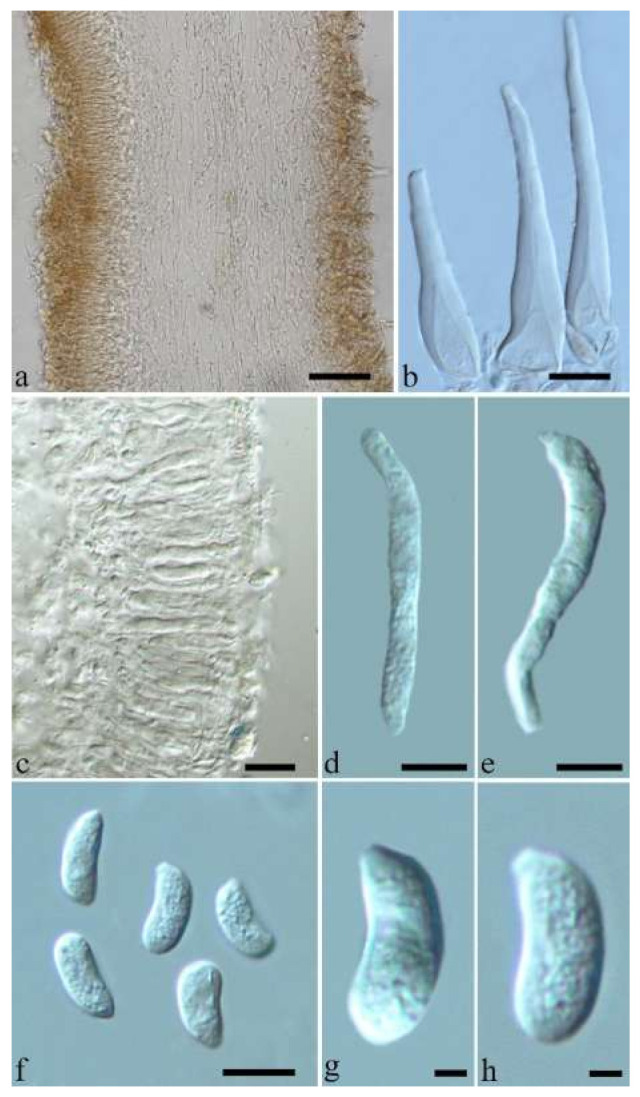
Microscopic structures of *Auricularia fibrillifera* (F 234519, holotype). (**a**) Cross-section of a basidioma; (**b**) Abhymenial hairs; (**c**–**e**) Basidia and basidioles; (**f**–**h**) Basidiospores. Bars: (**a**) 100 μm; (**c**) 20 μm; (**b**, **d**–**f**) 10 μm; (**g**,**h**) 2 μm.

**Figure 22 jof-07-00933-f022:**
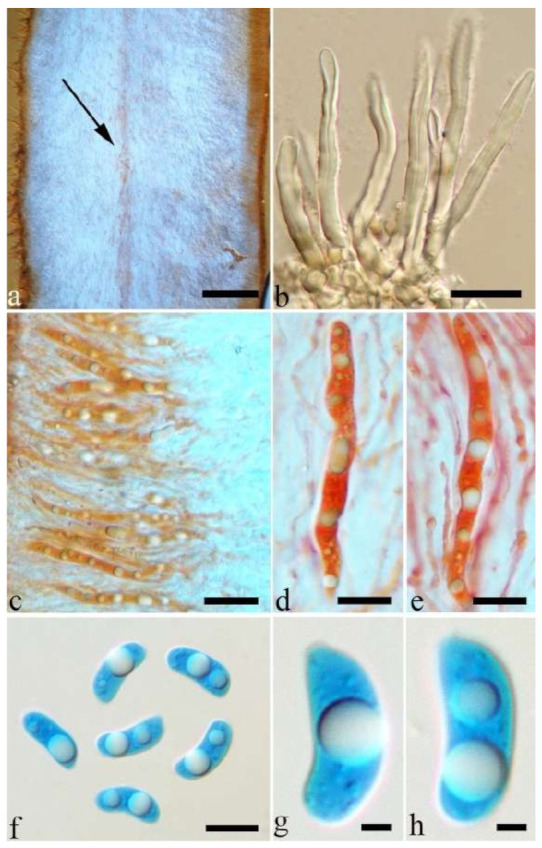
Microscopic structures of *Auricularia fuscosuccinea* (AG 1548). (**a**) Cross-section of a basidioma (medulla is shown by the arrow); (**b**) Abhymenial hairs; (**c**–**e**) Basidia and basidioles in hymenium; (**f**–**h**) Basidiospores. Bars: (**a**) 300 μm; (**b**,**c**) 20 μm; (**d**,**f**) 10 μm; (**g**,**h**) 2 μm.

**Figure 23 jof-07-00933-f023:**
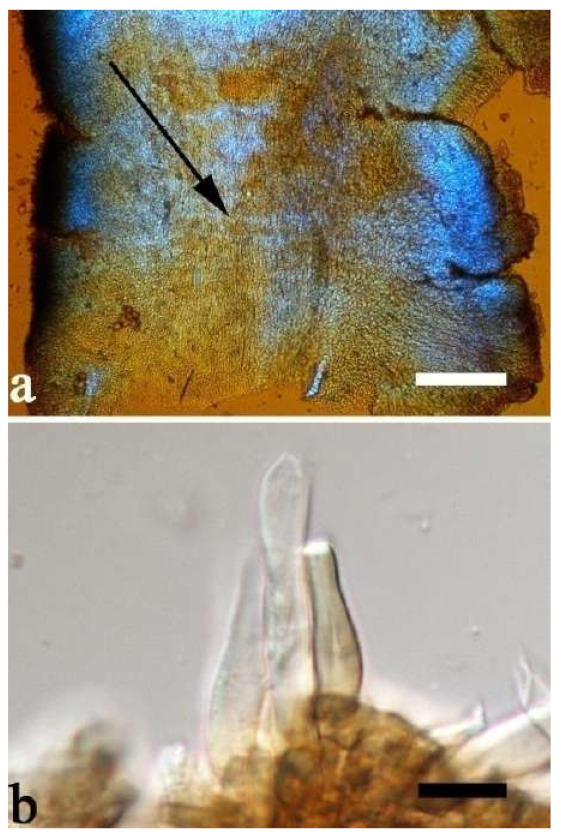
Microscopic structures of *Auricularia hainanensis* (HPNHM 103, holotype). (**a**) Cross-section of a basidioma (medulla is shown by the arrow); (**b**) Abhymenial hairs. Bars: (**a**) 300 μm; (**b**) 10 μm.

**Figure 24 jof-07-00933-f024:**
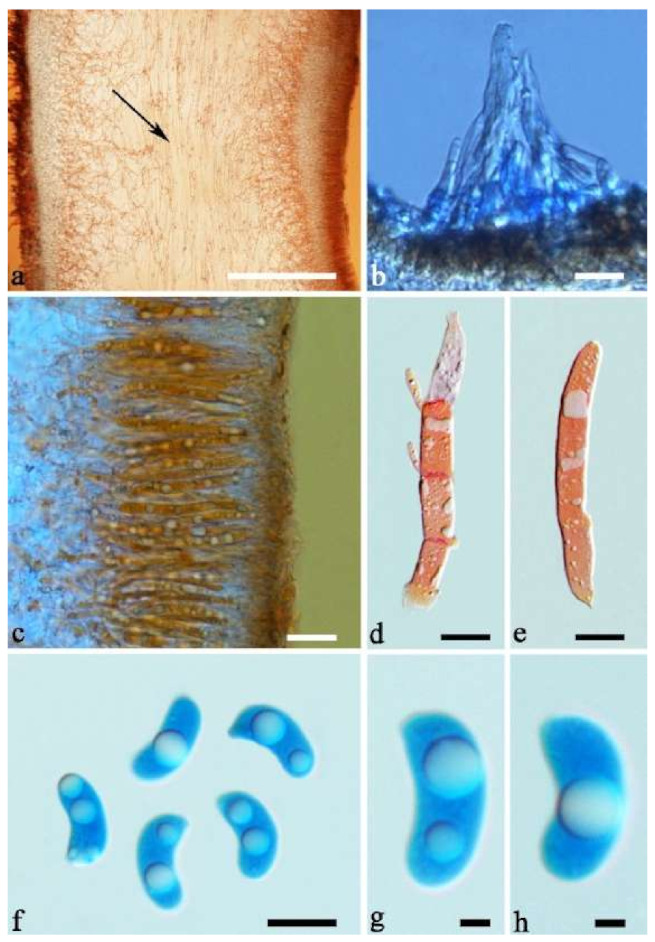
Microscopic structures of *Auricularia heimuer* (Cui 7101). (**a**) Cross-section of a basidioma (medulla is shown by the arrow); (**b**) Abhymenial hairs; (**c**–**e**) Basidia and basidioles; (**f**–**h**) Basidiospores. Bars: (**a**) 300 μm; (**b**,**c**) 20 μm; (**d**,**f**) 10 μm; (**g**,**h**) 2 μm.

**Figure 25 jof-07-00933-f025:**
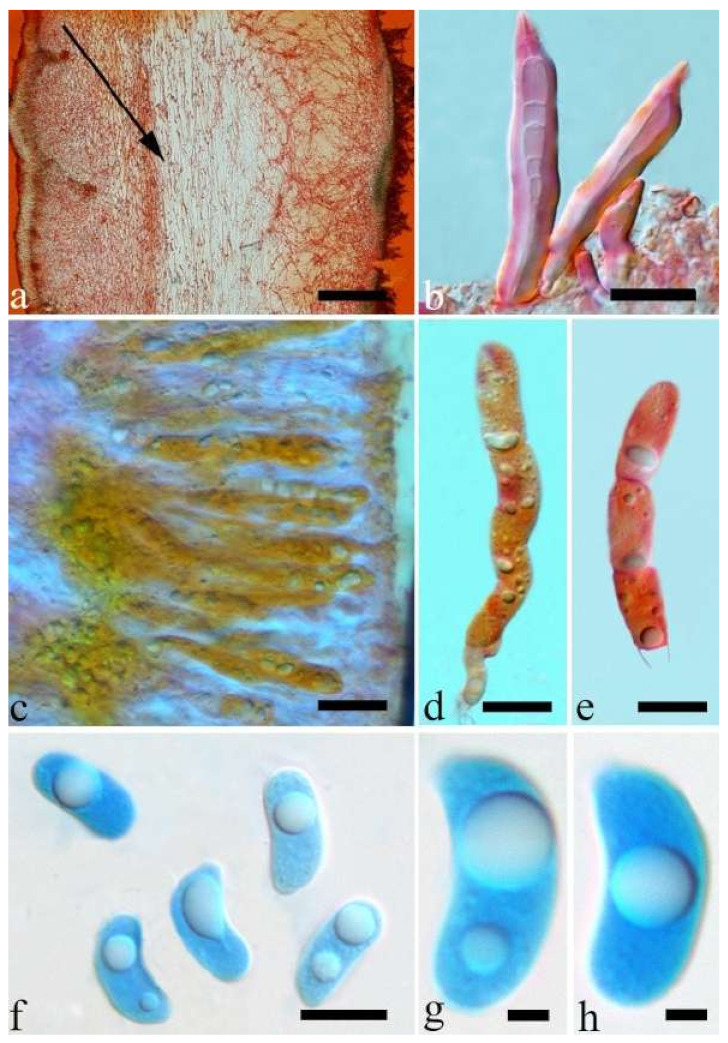
Microscopic structures of *Auricularia lateralis* (Dai 15670, holotype). (**a**) Cross-section of a basidioma (medulla is shown by the arrow); (**b**) Abhymenial hairs; (**c**–**e**) Basidia and basidioles; (**f**–**h**) Basidiospores. Bars: (**a**) 300 μm; (**b**,**c**) 20 μm; (**d**,**f**) 10 μm; (**g**,**h**) 2 μm.

**Figure 26 jof-07-00933-f026:**
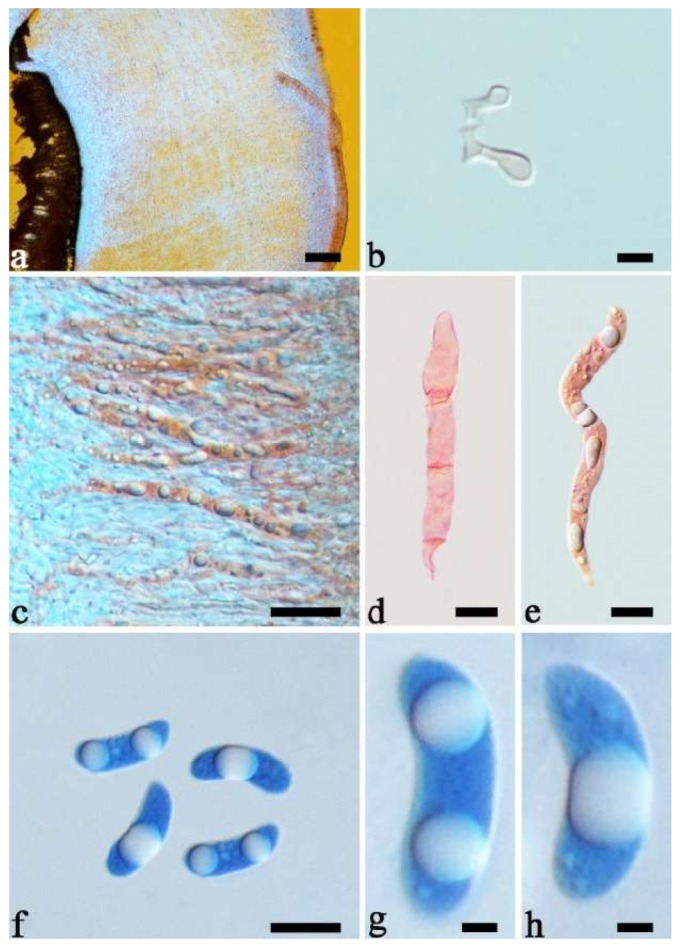
Microscopic structures of *Auricularia mesenterica* (Härkönen 11208). (**a**) Cross-section of a basidioma; (**b**) Cystidioles; (**c**–**e**) Basidia and basidioles; (**f**–**h**) Basidiospores. Bars: (**a**) 300 μm; (**b**) 4 μm; (**c**) 20 μm; (**d**,**f**) 10 μm; (**g**,**h**) 2 μm.

**Figure 27 jof-07-00933-f027:**
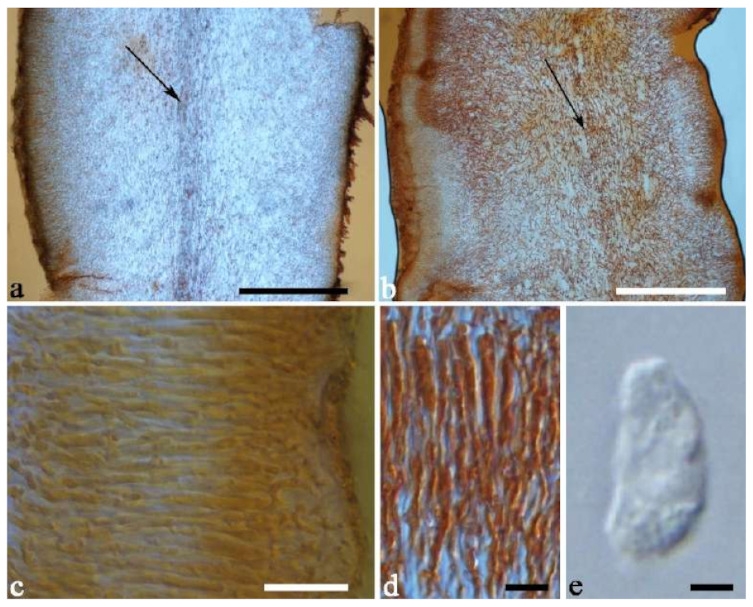
Microscopic structures of *Auricularia minor* (F-434, type). (**a**,**b**) Cross-section of a basidioma (medulla is shown by the arrow); (**c**,**d**) Basidia and basidioles in hymenium; (**e**) A basidiospore. Bars: (**a**,**b**) 300 μm; (**c**,**d**) 20 μm; (**e**) 2 μm.

**Figure 28 jof-07-00933-f028:**
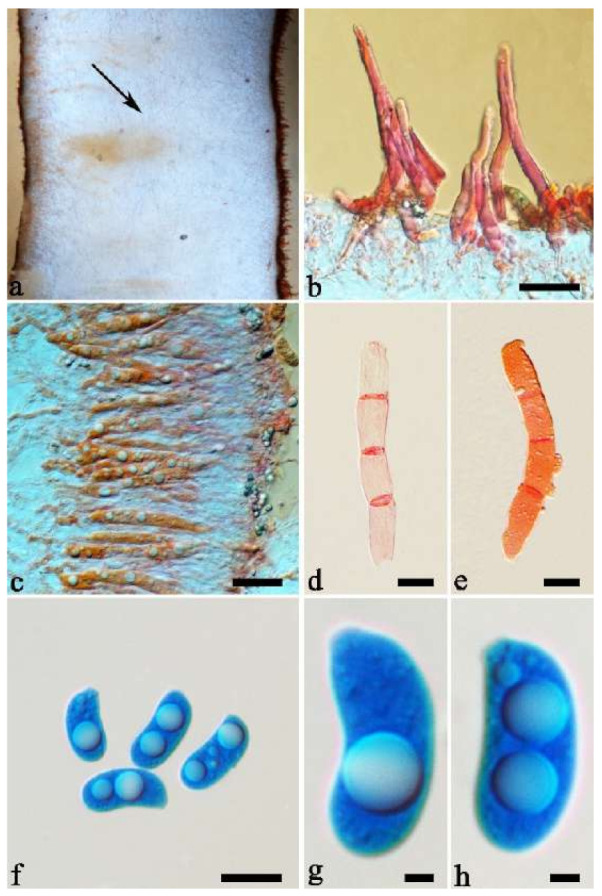
Microscopic structures of *Auricularia minutissima* (Dai 15452). (**a**) Cross-section of a basidioma (medulla is shown by the arrow); (**b**) Abhymenial hairs; (**c**–**e**) Basidia and basidioles; (**f**–**h**) Basidiospores. Bars: (**a**) 300 μm; (**b**) 30 μm; (**c**) 20 μm; (**d**,**f**) 10 μm; (**g**,**h**) 2 μm.

**Figure 29 jof-07-00933-f029:**
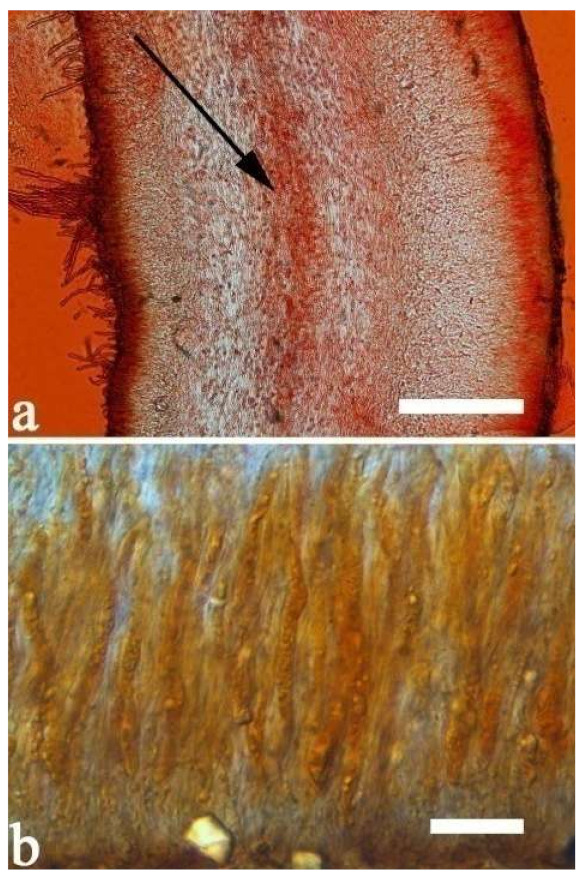
Microscopic structures of *Auricularia nigricans* (Ahti 36234). (**a**) Cross-section of a basidioma (medulla is shown by the arrow); (**b**) Basidia and basidioles in hymenium. Bars: (**a**) 300 μm; (**b**) 20 μm.

**Figure 30 jof-07-00933-f030:**
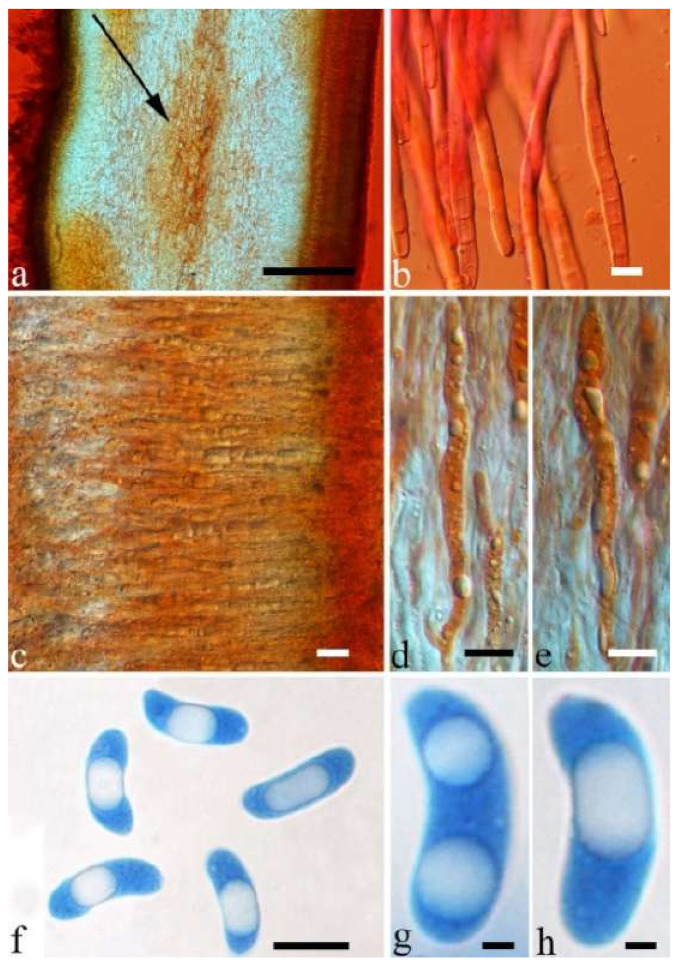
Microscopic structures of *Auricularia novozealandica* (PDD 83897, holotype). (**a**) Cross-section of a basidioma (medulla is shown by the arrow); (**b**) Abhymenial hairs; (**c**–**e**) Basidia and basidioles in hymenium; (**f**–**h**) Basidiospores. Bars: (**a**) 300 μm; (**b**–**f**); 20 μm; (**g**,**h**) 2 μm.

**Figure 31 jof-07-00933-f031:**
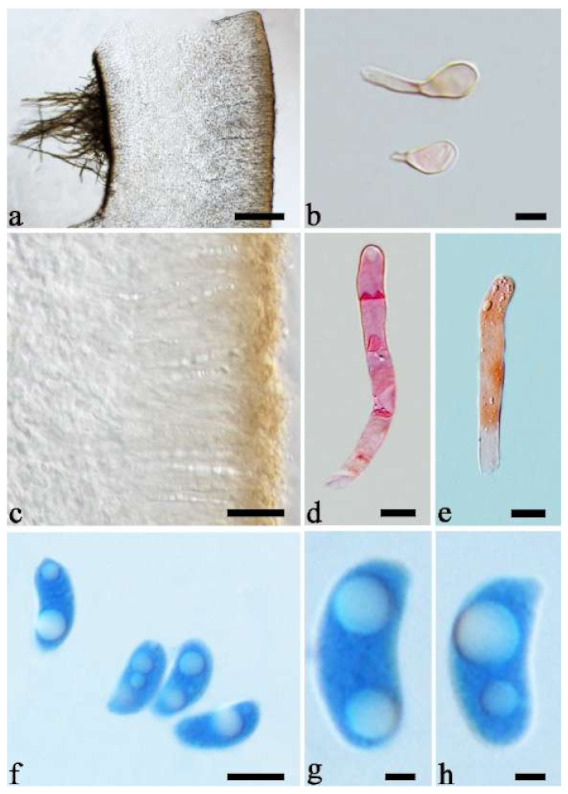
Microscopic structures of *Auricularia orientalis* (Dai 14875, holotype). (**a**) Cross-section of a basidioma; (**b**) Cystidioles; (**c**,**d**) Basidia and basidioles; (**f**–**h**) Basidiospores. Bars: (**a**) 300 μm; (**b**) 4 μm; (**c**) 20 μm; (**d**–**f**) 10 μm; (**g**,**h**) 2 μm.

**Figure 32 jof-07-00933-f032:**
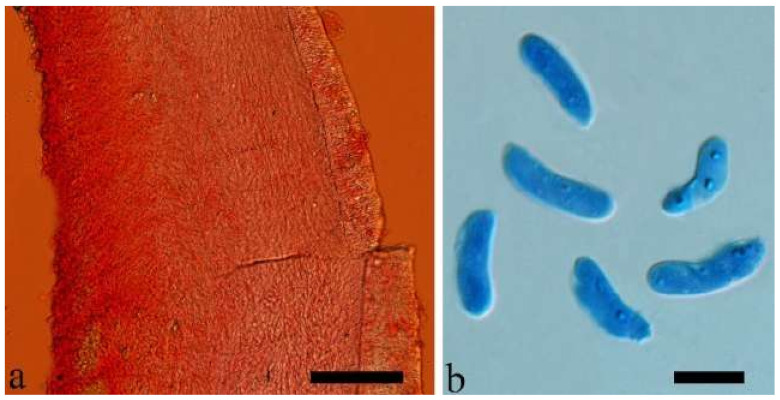
Microscopic structures of *Auricularia papyracea* (F-234520, type). (**a**) Cross-section of a basidioma; (**b**) Basidiospores. Bars: (**a**) 100 μm; (**b**) 10 μm.

**Figure 33 jof-07-00933-f033:**
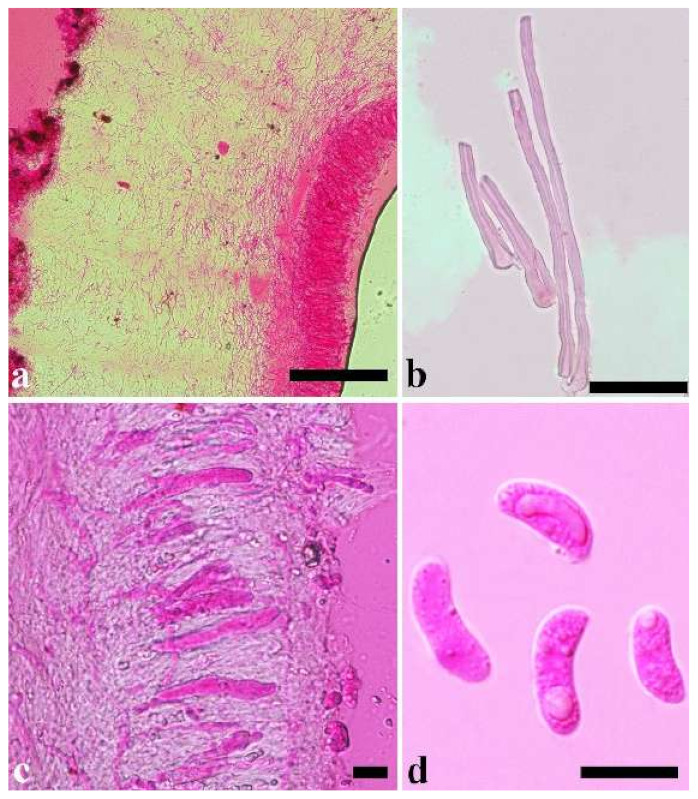
Microscopic structures of *Auricularia pilosa* (LWZ20190421-7, holotype). (**a**) Cross-section of a basidioma; (**b**) Abhymenial hairs; (**c**) Basidia and basidioles; (**d**) Basidiospores. Bars: (**a**) 200 μm; (**b**) 50 μm; (**c**,**d**) 10 μm.

**Figure 34 jof-07-00933-f034:**
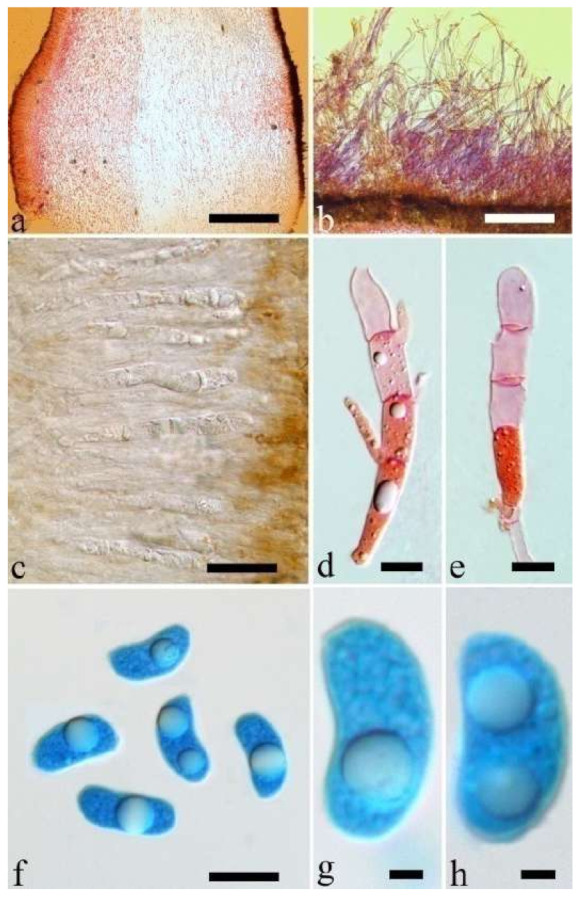
Microscopic structures of *Auricularia pusio* (AK 174). (**a**) Cross-section of a basidioma; (**b**) Abhymenial hairs; (**c**–**e**) Basidia and basidioles; (**f**–**h**) Basidiospores. Bars: (**a**) 300 μm; (**b**) 200 μm; (**c**) 20 μm; (**d**,**f**) 10 μm; (**g**,**h**) 2 μm.

**Figure 35 jof-07-00933-f035:**
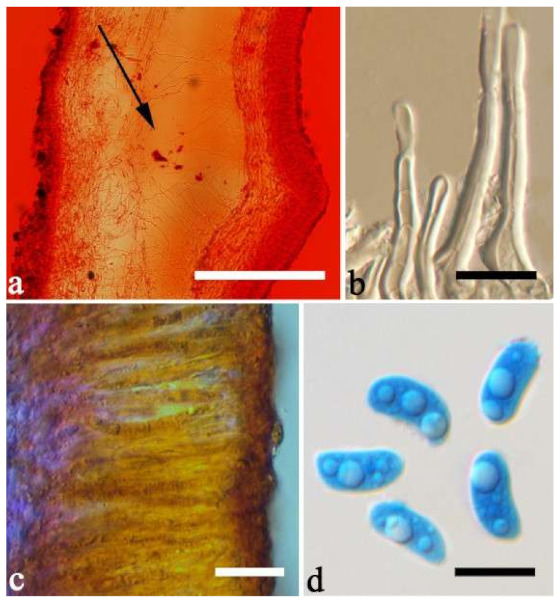
Microscopic structures of *Auricularia scissa* (Ahti 49388). (**a**) Cross-section of a basidioma (schizomedulla is shown by the arrow); (**b**) Abhymenial hairs; (**c**) Basidia and basidioles in hymenium; (**d**) Basidiospores. Bars: (**a**) 300 μm; (**b**,**c**) 20 μm; (**d**) 10 μm.

**Figure 36 jof-07-00933-f036:**
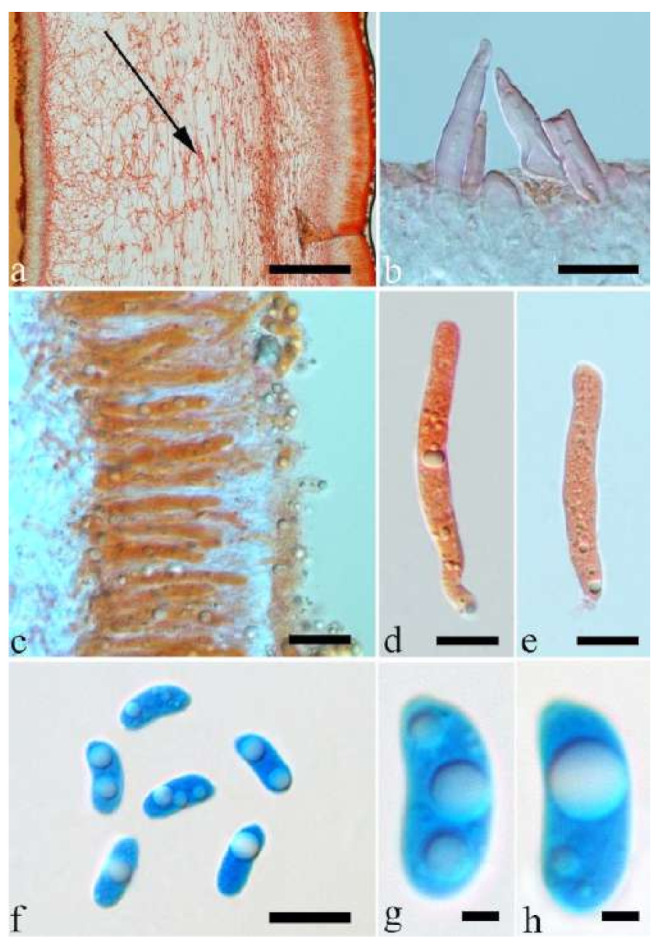
Microscopic structures of *Auricularia sinodelicata* (Dai 13926, holotype). (**a**) Cross-section of a basidioma (medulla is shown by the arrow); (**b**) Abhymenial hairs; (**c**–**e**) Basidia and basidioles in hymenium; (**f**–**h**) Basidiospores. Bars: (**a**) 200 μm; (**b**,**c**) 20 μm; (**d**,**f**) 10 μm; (**g**,**h**) 2 μm.

**Figure 37 jof-07-00933-f037:**
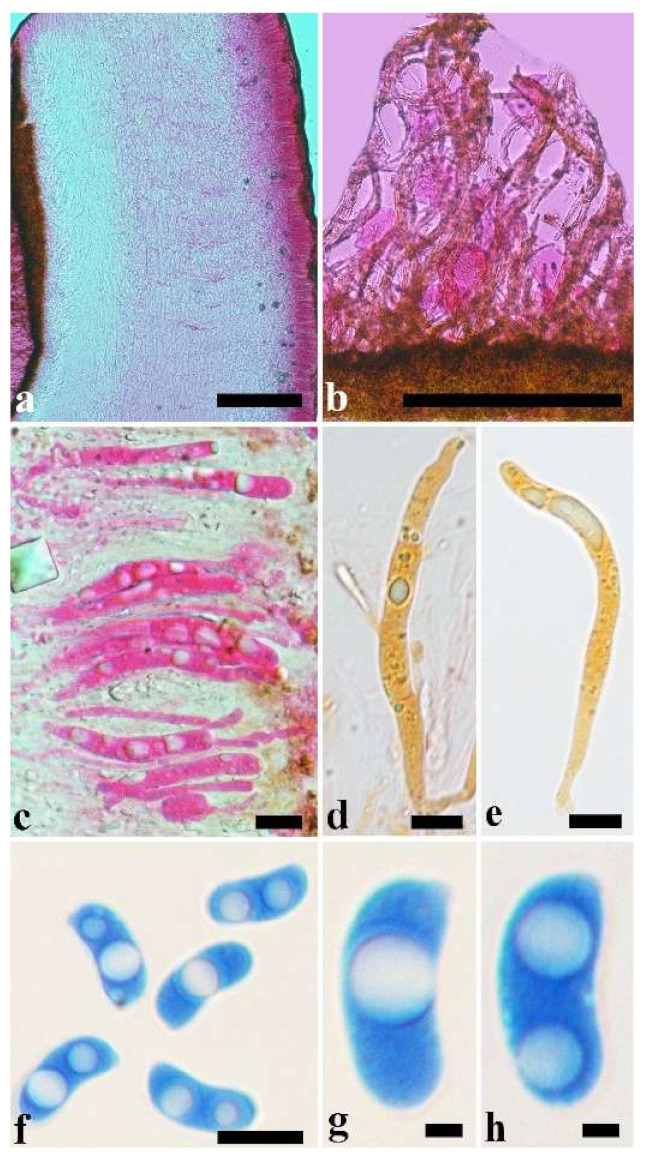
Microscopic structures of *Auricularia srilankensis* (Dai 19522, holotype). (**a**) Cross-section of a basidioma; (**b**) Abhymenial hairs; (**c**–**e**) Basidia and basidioles; (**f**–**h**) Basidiospores. Bars: (**a**,**b**) 200 μm; (**c**–**f**) 10 μm; (**g**,**h**) 2 μm.

**Figure 38 jof-07-00933-f038:**
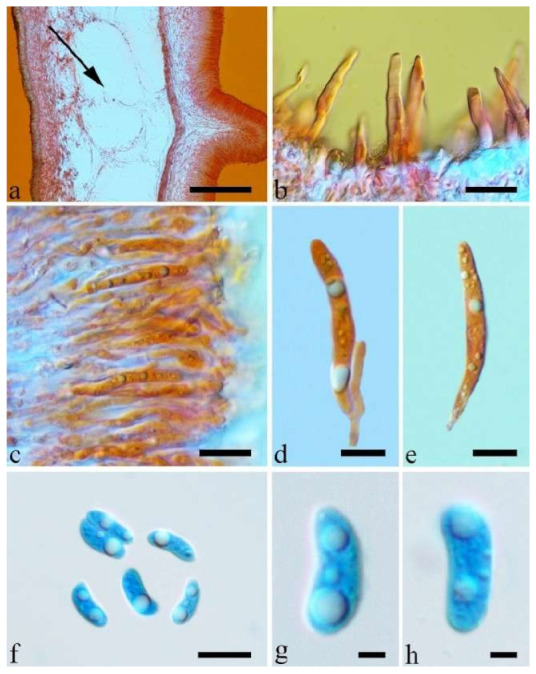
Microscopic structures of *Auricularia subglabra* (Dai 17417). (**a**) Cross-section of a basidioma (schizomedulla is shown by the arrow); (**b**) Abhymenial hairs; (**c**–**e**) Basidia and basidioles; (**f**–**h**) Basidiospores. Bars: (**a**) 300 μm; (**b**,**c**) 20 μm; (**d**,**f**) 10 μm; (**g**,**h**) 2 μm.

**Figure 39 jof-07-00933-f039:**
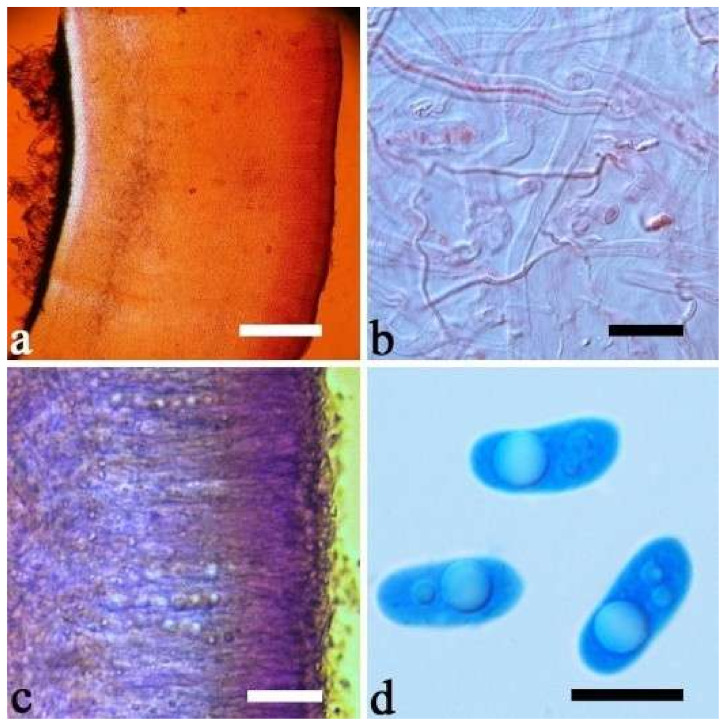
Microscopic structures of *Auricularia submesenterica* (Dai 15450, holotype). (**a**) Cross-section of a basidioma; (**b**) Hyphae; (**c**) Basidia and basidioles in hymenium; (**d**) Basidiospores. Bars: (**a**) 300 μm; (**b**,**c**) 20 μm; (**d**) 10 μm.

**Figure 40 jof-07-00933-f040:**
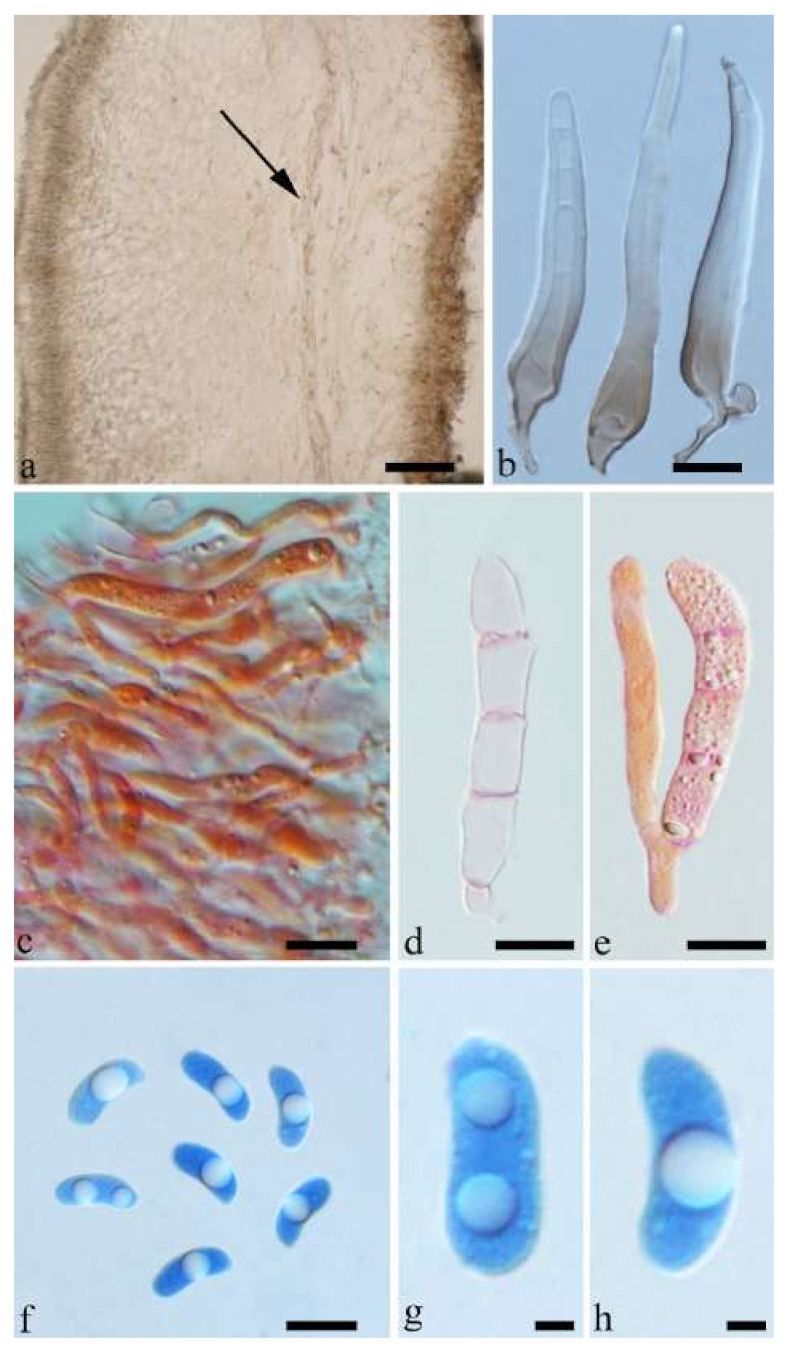
Microscopic structures of *Auricularia thailandica* (Dai 12332). (**a**) Cross-section of a basidioma (medulla is shown by the arrow); (**b**) Abhymenial hairs; (**c**–**e**) Basidia and basidioles; (**f**–**h**) Basidiospores. Bars: (**a**) 100 μm; (**b**–**f**) 10 μm; (**g**,**h**) 2 μm.

**Figure 41 jof-07-00933-f041:**
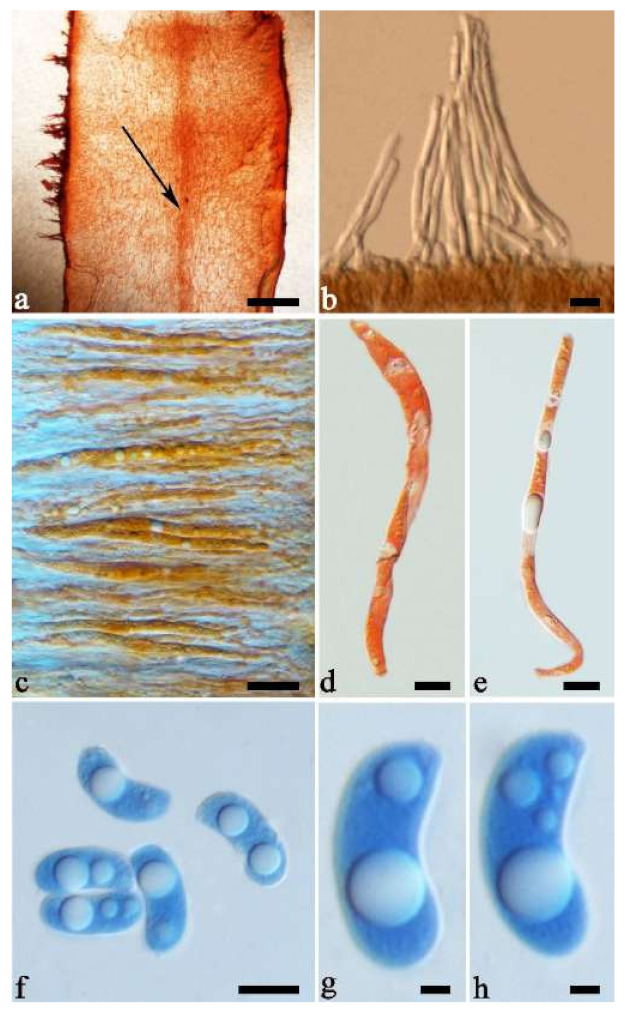
Microscopic structures of *Auricularia tibetica* (Cui 12267, holotype). (**a**) Cross-section of a basidioma (medulla is shown by the arrow); (**b**) Abhymenial hairs; (**c**–**e**) Basidia and basidioles; (**f**–**h**) Basidiospores. Bars: (**a**) 300 μm; (**b**,**c**) 20 μm; (**d**,**f**) 10 μm; (**g**,**h**) 2 μm.

**Figure 42 jof-07-00933-f042:**
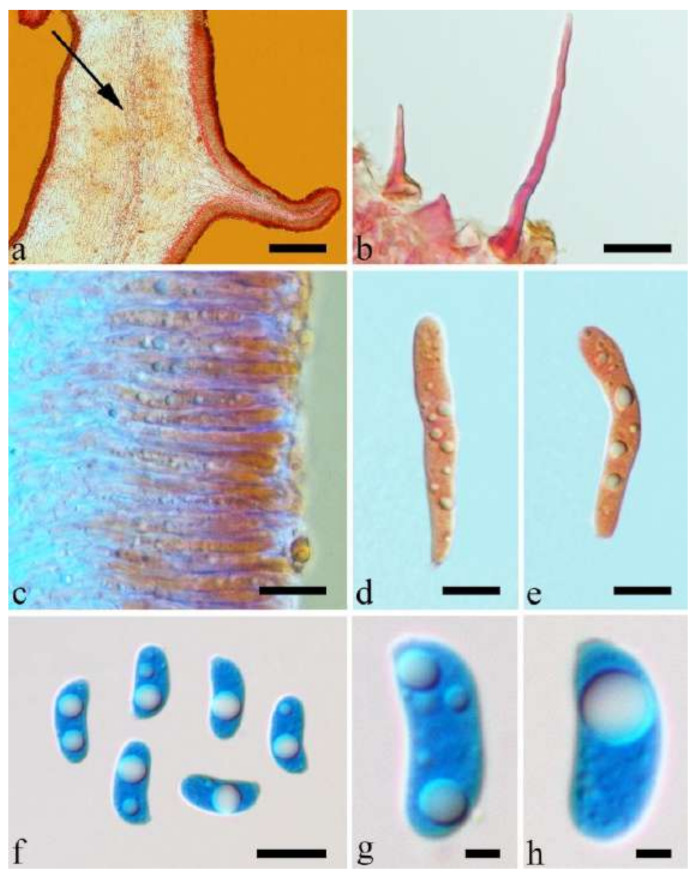
Microscopic structures of *Auricularia tremellosa* (Dai 17419). (**a**) Cross-section of a basidioma (medulla is shown by the arrow); (**b**) Abhymenial hairs; (**c**–**e**) Basidia and basidioles; (**f**–**h**) Basidiospores. Bars: (**a**) 300 μm; (**b**,**c**) 20 μm; (**d**,**f**) 10 μm; (**g**,**h**) 2 μm.

**Figure 43 jof-07-00933-f043:**
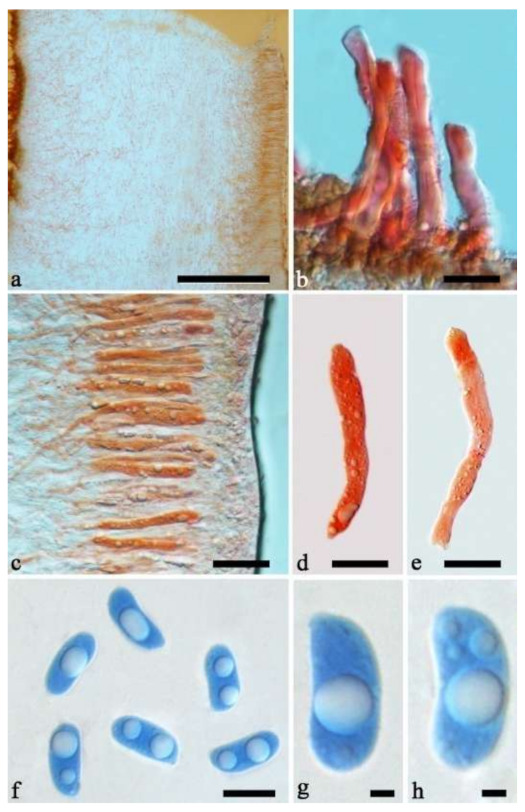
Microscopic structures of *Auricularia villosula* (Dai 13652). (**a**) Cross-section of a basidioma; (**b**) Abhymenial hairs; (**c**–**e**) Basidia and basidioles; (**f**–**h**) Basidiospores. Bars: (**a**) 300 μm; (**b**,**d**–**f**) 10 μm; (**c**) 20 μm; (**g**,**h**) 2 μm.

**Figure 44 jof-07-00933-f044:**
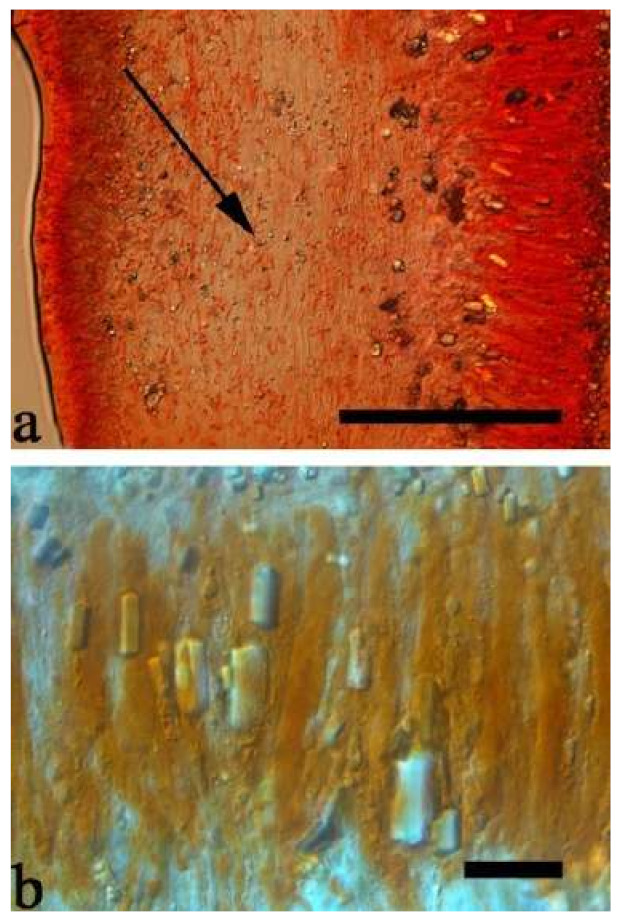
Microscopic structures of *Auricularia xishaensis* (HPNHM 1338, holotype). (**a**) Cross-section of a basidioma (medulla is shown by the arrow); (**b**) Basidia and basidioles in hymenium. Bars: (**a**) 200 μm; (**b**) 20 μm.

**Table 1 jof-07-00933-t001:** List of species, specimens and GenBank accession number of sequences used in this study.

Species Name	Sample No.	Country	GenBank Accessions			
			ITS	nLSU	rpb1	rpb2
*Amphistereum schrenkii*	HHB 8476	USA	KX262130	KX262178	—	—
*Auricularia africana*	T 3	Kenya	**MH213349**	**MZ669918**	—	—
*A. africana*	Ryvarden 44929, holotype	Uganda	**MH213350**	**MZ669897**	—	**MZ740061**
*A. americana*	Dai 13476	China	KM396766	KM396824	—	KT152126
*A. americana*	Dai 13461	China	KM396763	KM396821	—	—
*A. americana*	LE 296428	Russia	KJ698429	—	—	—
*A. americana*	HHB 11370	USA	KM396766	—	—	—
*A. americana*	HHB 14337	USA	KM396768	—	—	—
*A. americana*	Cui 11509	China	KT152094	KT152110	—	KT152127
*A. americana*	Cui 11657	China	KT152095	KT152111	**MZ753951**	KT152128
*A. angiospermarum*	TJV-93-12-SP	USA	KT152096	KT152112	—	KT152129
*A. angiospermarum*	Cui 12360, holotype	USA	KT152097	KT152113	**MZ753952**	KT152130
*A. angiospermarum*	HHB 11037	USA	KT152098	KT152114	—	KT152131
*A. asiatica*	Dai 15285	China	**KX022009**	**KX022040**	—	—
*A. asiatica*	Dai 16149	China	**KX022010**	**KX022041**	—	—
*A. asiatica*	Dai 16224	China	**KX022011**	**KX022042**	**MZ753953**	**MZ740045**
*A. asiatica*	BBH 895, holotype	Thailand	KX621160	—	—	—
*A. asiatica*	OM 13932	Indonesia	**MZ618931**	**MZ669899**	**MZ753954**	**MZ740046**
*A. auricula-judae*	Dai 13210	France	KM396769	KM396824	—	KP729312
*A. auricula-judae*	MT 7	Czech Republic	KM396771	KM396826	—	KP729314
*A. auricula-judae*	JT 04	UK	KT152099	KT152115	—	KT152132
*A. auricula-judae*	MW 446	Germany	AF291268	AF291289	—	—
*A. auricula-judae*	Dai 16353	France	**MZ618932**	**MZ669900**	**MZ753955**	**MZ740047**
*A. australiana*	CNSBlitz0098	Australia	JX065171	—	—	—
*A. australiana*	CNSBlitz0093	Australia	JX065165	—	—	—
*A. australiana*	HT 190, holotype	Australia	**MZ647503**	**MZ669920**	—	—
*A. australiana*	HN 3213	Australia	**MZ647504**	**MZ669921**	—	—
*A. brasiliana*	CRSL 886	Brazil	KP729274	KP729292	—	—
*A. brasiliana*	AN-MA 42, holotype	Brazil	KP729275	KP729293	—	—
*A. brasiliana*	RSC 359	Brazil	KP729276	KP729294	—	—
*A. brasiliana*	RSC 487	Brazil	KP729277	KP729295	—	—
*A. camposii*	URM 76905, holotype	Brazil	**MH213351**	**MH213395**	—	**MH213427**
*A. camposii*	URM 83464	Brazil	**MH213352**	**MH213396**	—	**MH213428**
*A. conferta*	Dai 18825, holotype	Australia	**MZ647500**	**MZ669901**	—	**MZ740048**
*A. conferta*	Dai 18826	Australia	**MZ647505**	—	—	—
*A. cornea*	YG-Dr1	Germany	**MH213353**	**MH213397**	—	**MH213429**
*A. cornea*	Dai 12587	South Africa	**KX022012**	**KX022043**	—	—
*A. cornea*	Dai 15336	China	**KX022014**	**KX022045**	—	**KX022074**
*A. cornea*	Wu 07	China	**MH213354**	**MH213398**	—	**MH213430**
*A. cornea*	Dai 17352	Ghana	**MH213355**	**MH213399**	—	**MH213431**
*A. cornea*	Cui 7517	China	**MH213356**	**MH213400**	—	—
*A. cornea*	Dai 14876	China	**MH213357**	**MH213401**	—	—
*A. cornea*	Lira 663	Brazil	**MH213359**	**MH213403**	—	**MH213433**
*A. cornea*	Dai 15447	Benin	**MH213360**	—	—	—
*A. cornea*	Dai 13754	China	**MH213361**	**MH213404**	**MZ753956**	**MH213434**
*A. cornea*	Dai 17865	Singapore	**MH213362**	**MH213405**	—	—
*A. cornea*	Dai 18315	Vietnam	**MH213363**	**MH213406**	—	—
*A. cornea*	Cui 11346	China	**MZ618933**	**MZ669902**	**MZ753957**	**MZ740049**
*A. cornea*	Cui 11162	China	**MZ618934**	**MZ669903**	**MZ753958**	**MZ740050**
*A. cornea*	AG 1547	Brazil	**KX022016**	**KX022047**	**MZ753959**	**KX022076**
*A. cornea*	Dai 13621	China	**MZ618936**	**MZ669905**	**MZ753960**	**MZ740052**
*A. cornea*	Dai 19650	Sri Lanka	**MZ618937**	**MZ669906**	—	**MZ740053**
*A. delicata*	P 14, epitype	Cameroon	**MH213364**	**MZ669933**	—	—
*A. fibrillifera*	Dai 13598A	China	KP765615	KP765629	**MZ753961**	**KX022084**
*A. fibrillifera*	Cui 6704	China	KP765613	KP765627	—	**MH213435**
*A. fibrillifera*	F 234519, holotype	Papua New Guinea	KP765610	KP765624	—	—
*A. fibrillifera*	Dai 18486	Zambia	**MH213365**	**MZ669923**	—	—
*A. fibrillifera*	Cui 6318	China	KP765611	KP765625	**MZ753962**	**MZ740054**
*A. fuscosuccinea*	FP-102573-SP	USA	**KX022027**	**KX022058**	—	**KX022088**
*A. fuscosuccinea*	Dai 17406	Brazil	**MH213366**	**MH213407**	—	**MH213436**
*A. fuscosuccinea*	Dai 17422	Brazil	**MH213367**	**MH213408**	—	**MH213437**
*A. fuscosuccinea*	Dai 17451	Brazil	**MH213368**	**MH213409**	—	**MH213438**
*A. fuscosuccinea*	OM 17909	USA	**KX022029**	**KX022060**	**MZ753963**	**KX022090**
*A. fuscosuccinea*	AG 1548	Brazil	**KX022028**	**KX022059**	**MZ753964**	**KX022089**
*A. heimuer*	Dai 2291	China	KM396785	KM396838	—	KP729315
*A. heimuer*	Dai 13503	China	KM396789	KM396840	—	KP729316
*A. heimuer*	AFM 21	Japan	AB615232	—	—	—
*A. heimuer*	LE 296433	Russia	KJ698423	—	—	—
*A. heimuer*	Heishan	China	KM396778	KM396831	KY418981	KY419034
*A. heimuer*	xiaoheimao	China	KM396781	KM396834	KY418982	KY419035
*A. heimuer*	Dai 13765, holotype	China	KM396793	KM396844	**MZ753965**	KP729317
*A. lateralis*	Dai 16416	China	**KX022023**	**KX022054**	—	**KX022081**
*A. lateralis*	Dai 16420	China	**KX022025**	**KX022056**	—	**KX022083**
*A. lateralis*	Dai 16417	China	**MH213369**	**MH213410**	—	**MH213440**
*A. lateralis*	Dai 16418	China	**MH213370**	**MH213411**	—	**MH213441**
*A. lateralis*	Dai 15670, holotype	China	**KX022022**	**KX022053**	—	—
*A. mesenterica*	LYBR 5353	France	KM396801	KM396849	—	KP729318
*A. mesenterica*	BRNM 648573	Czech Republic	KP729279	KP729297	**MZ753966**	KP729320
*A. mesenterica*	Kytovuori-89-333	Estonia	KP729284	KP729302	—	KP729321
*A. mesenterica*	Miettinen 12680	Switzerland	KP729286	KP729304	—	KP729322
*A. mesenterica*	Haikonen 11208	UK	KP729287	KP729305	—	KP729323
*A. mesenterica*	Haikonen 8045	Italy	KP729285	KP729303	**MZ753967**	—
*A. mesenterica*	YG 029	Uzbekistan	**MZ618938**	**MZ669907**	**MZ753968**	**MZ740055**
*A. mesenterica*	YG 037	Uzbekistan	**MZ618939**	**MZ669908**	**MZ753969**	**MZ740056**
*A. minutissima*	Dai 14881, holotype	China	KT152104	KT152120	—	KT152137
*A. minutissima*	Dai 15455	China	**KX022030**	**KX022061**	—	**KX022091**
*A. minutissima*	Dai 15453	China	**MH213371**	**MH213412**	**MZ753970**	**MH213442**
*A. minutissima*	LE 296424	Russia	KJ698434	—	—	—
*A. nigricans*	Ahti 36234	Costa Rica	KM396802	KM396850	—	—
*A. nigricans*	TJY-93-242	USA	KM396803	KM396851	—	—
*A. nigricans*	Ahti 55718	Mexico	**MH213372**	**MH213413**	—	—
*A. novozealandica*	PDD 81195	New Zealand	**KX022033**	**KX022064**	—	—
*A. novozealandica*	PDD 88998	New Zealand	**KX022035**	**KX022066**	—	—
*A. novozealandica*	PDD 75110	New Zealand	**KX022032**	**KX022063**	—	—
*A. novozealandica*	PDD 83897, holotype	New Zealand	**KX022034**	**KX022065**	—	—
*A. orientalis*	Dai 14875, holotype	China	KP729270	KP729288	—	KP729310
*A. orientalis*	Dai 1831	China	KP729271	KP729289	**MZ753971**	KP729311
*A. orientalis*	Dai 15813	China	**KX022036**	**KX022067**	**MZ753972**	**KX022093**
*A. pilosa*	LWZ 20190421-7, holotype	Ethiopia	**MZ647506**	—	—	—
*A. pilosa*	JMH 45	Tanzania	KM267731	KM267738	—	—
*A. pusio*	AK 174	Australia	**MH213373**	**MZ669925**	—	—
*A. pusio*	AK 547	Australia	**MH213374**	**MH213414**	—	**MH213443**
*A. pusio*	Smith 18	Zambia	**MH213375**	**MZ669926**	—	—
*A. scissa*	DR 777	Dominican Republic	KM396804	KM396852	—	—
*A. scissa*	Ahti 49388	Dominican Republic	KM396805	KM396853	—	KP729324
*A. scissa*	TFB 11193, holotype	Dominican Republic	JX065160	—	—	—
*A. sinodelicata*	Cui 8596	China	**MH213376**	**MH213415**	—	**MH213444**
*A. sinodelicata*	Dai 13758	China	**MH213378**	**MH213416**	**MZ753973**	**MH213445**
*A. sinodelicata*	Dai 13926, holotype	China	**MH213379**	**MZ669909**	**MZ753974**	—
*A. sinodelicata*	Dai 13927	China	**MH213380**	**MH213417**	**MZ753975**	**MH213446**
*A. sinodelicata*	Dai 12242	China	**MH213381**	**MH213418**	—	**MH213447**
*A. sinodelicata*	Dai 15083	China	**MZ618940**	**MZ669910**	**MZ753976**	**MZ740057**
*A. subglabra*	Dai 17403	Brazil	**MH213382**	**MH213419**	—	**MH213448**
*A. subglabra*	Dai 17394	Brazil	**MH213383**	**MZ669927**	—	—
*A. subglabra*	Wu 08	Brazil	**MH213384**	**MZ669928**	—	—
*A. subglabra*	Wu 09	Brazil	**MH213385**	**MZ669929**	—	—
*A. subglabra*	TFB10046	Costa Rica	JX524199	—	—	—
*A. subglabra*	JV 1808/125	French Guiana	**MZ618941**	**MZ669911**	**MZ753977**	—
*A. srilankensis*	Dai 19522, holotype	Sri Lanka	**MZ647501**	**MZ669912**	**MZ753978**	—
*A. srilankensis*	Dai 19575	Sri Lanka	**MZ647502**	**MZ669913**	—	**MZ740058**
*A. srilankensis*	Dai 19519	Sri Lanka	**MZ647507**	**MZ669930**	—	—
*A. srilankensis*	Dai 19520	Sri Lanka	**MZ647508**	**MZ669931**	—	—
*A. submesenterica*	Dai 792	China	**KX022037**	**KX022068**	—	**KX022094**
*A. submesenterica*	Dai 14773	China	**KX022038**	**KX022069**	**MZ753979**	**KX022095**
*A. submesenterica*	Dai 15449	China	**KX022039**	**KX022070**	—	**KX022096**
*A. submesenterica*	Dai 15450, holotype	China	**MH213386**	**MH213420**	—	**MH213449**
*A. submesenterica*	Dai 15451	China	**MZ618942**	**MZ669914**	**MZ753980**	**MZ740059**
*A. submesenterica*	Dai 15820	China	**MH213387**	**MH213421**	—	**MH213450**
*A. thailandica*	Dai 13655A	China	KP765619	KM396863	—	KP729329
*A. thailandica*	Dai 15335	China	**KX022026**	**KX022057**	—	**KX022086**
*A. thailandica*	Dai 13759	China	KP765621	KP765635	**MZ753981**	**MH213451**
*A. thailandica*	Dai 15080	China	KP765622	KP765636	—	**MH213452**
*A. thailandica*	MFLU 130396, type	Thailand	KR336690	—	—	—
*A. tibetica*	Cui 12267, holotype	China	KT152106	KT152122	MW290311	KT152139
*A. tibetica*	Cui 12337	China	KT152108	KT152124	—	KT152141
*A. tibetica*	Dai 15604	China	**MH213388**	**MH213422**	—	**MH213453**
*A. tibetica*	Dai 15734	China	**MH213389**	**MH213423**	—	**MH213454**
*A. tibetica*	Dai 13336	China	**MZ618943**	**MZ669915**	**MZ753982**	**MZ740060**
*A. tibetica*	Cui 12268	China	KT152107	KT152123	**MZ753983**	KT152140
*A. tremellosa*	AJS 5896	Mexico	JX065162	—	—	—
*A. tremellosa*	TENN 28734	Peru	JX065159	—	—	—
*A. tremellosa*	AJS 1304	Mexico	JX065158	—	—	—
*A. tremellosa*	Dai 17415	Brazil	**MH213390**	**MH213424**	—	**MH213455**
*A. tremellosa*	Dai 17419	Brazil	**MH213391**	**MZ669932**	—	—
*A. villosula*	Dai 13450	China	KM396812	KM396860	—	KP729327
*A. villosula*	Dai 13453	China	KM396813	KM396861	—	KP729328
*A. villosula*	Dai 15994	China	**MH213392**	—	—	—
*A. villosula*	LE 296422, holotype	Russia	NR137873	—	—	KJ698441
*A. villosula*	MFLU 162128	Thailand	KX621164	—	—	—
*A. villosula*	01973	China	**MZ618944**	**MZ669916**	**MZ753984**	**MZ740062**
*Elmerina dimidiata*	O 18261	Belize	JQ764664	JQ764641	—	—
*E. efibulata*	Yuan 4525	China	**MZ618945**	**MZ669917**	**MZ753985**	**MZ740063**
*Exidia candida*	LE 38198	Canada	KY801871	KY801896	—	—
*E. glandulosa*	MW 355	Germany	AF291273	AF291319	—	—
*Heterochaete shearii*	USJ 54609	Costa Rica	AF291284	AF291335	—	—
*Heteroradulum deglubens*	LE 38182	Sweden	KX262112	KX262162	—	—

Newly generated sequences for this study are indicated in **bold**.

**Table 2 jof-07-00933-t002:** A comparison of species in the *Auricularia cornea* complex.

Name	Upper Surface	Crystals	Hairs (µm)	Basidia (µm)	Basidiospores (µm)
*A. camposii*	Tomentose	Present	120–250 × 6–7	60–75 × 4–5.5	12.5–15 × 5–6
*A. cornea*	Pilose	Present	180–425 × 6–9	60–75 × 4–6	13.8–16.5 × 4.5–6
*A. eburnea*	Pilose	Absent	280–380 × 5–6.5	50–65 × 4–5	15.8–18 × 5.5–7.5
*A. eminii*	Pilose	Absent	>10 000	40–60 × 4–6	12–14 × 4–5
*A. nigricans*	Hispid	Present	300–600 × 7–9	50–60 × 4–6	14.5–17 × 5–7
*A. novozealandica*	Pilose	Absent	100–320 × 7–9	70–86 × 5–6	16–19 × 5.3–6.1

**Table 3 jof-07-00933-t003:** A comparison of species in the *Auricularia delicata* complex.

Name	Upper Surface	Medulla	Hairs (µm)	Basidia (µm)	Basidiospores (µm)
*A. australiana*	Pilose	Absent	60–100 × 7–11	45–55 × 4.5–6	11–12.8 × 4.4–5
*A. conferta*	Pilose	Indistinctly present	45–95 × 8–15	44–58 × 4.5–6.5	11–13 × 4.5–5.2
*A. delicata*	Pilose	Absent or indistinctly present	60–100 × 5–8	48–65 × 4–6	10–11.5 × 4.5–5.5
*A. lateralis*	Distinctly pilose	Present	95–250 × 9–14	50–70 × 5–6.5	12.9–14.2 × 5.2–6
*A. pilosa*	Distinctly pilose	Absent	90–207 × 8–16	35–45 × 4–5.5	11–13.8 × 4.2–5.8
*A. tremellosa*	Pilose	Present or absent	39–80 × 5–7	36–42 × 5–5.5	10.2–12 × 4–5
*A. sinodelicata*	Scantly pilose	Indistinctly present	30–80 × 6–9	30–45 × 4–5.5	10–12 × 4.3–5.1
*A. scissa*	Pilose	Schizomedulla present	40–100 × 5–10	40–60 × 4–5	9–12 × 4–6
*A. subglabra*	Scantly pilose	Schizomedulla present	20–45 × 5–6	30–45 × 3.5–4.5	9–10.8 × 3.5–4.5

**Table 4 jof-07-00933-t004:** A comparison of species in the *Auricularia fuscosuccinea* complex.

Name	Medulla	Crystals	Hairs (µm)	Basidia (µm)	Basidiospores (µm)
*A. fibrillifera*	Indistinctly present	Present	60–100 × 10–20	41–57 × 4–6	11–14 × 4–5
*A. fuscosuccinea*	Present	Present	75–165 × 5–8	42–66 × 4–7	12–14 × 4.3–5.2
*A. papyracea*	Absent	Absent	20 × 7	60–70 × 8–10	16–18 × 4.2–5.3
*A. thailandica*	Present	Present	50–80 × 6–14	40–60 × 3–6	9.5–13 × 4–5.6
*A. xishaensis*	Present	Present	80–120 × 4–5	50–72 × 5–7	15–16 × 6–7

**Table 5 jof-07-00933-t005:** A comparison of species in the *Auricularia auricula-judae* complex.

Name	Medulla	Crystals	Hairs (µm)	Basidia (µm)	Basidiospores (µm)	Host
*A. americana*	Present or absent	Absent	67–125 × 4.5–6	55–71 × 4–5	14–16.5 × 4.5–5.4	Gymnosperm
*A. angiospermarum*	Absent	Absent	80–140 × 5–6	47–63 × 3.5–5	13–15 × 4.8–5.5	Angiosperm
*A. auricula-judae*	Present or absent	Present	100–150 × 5–7.5	65–85 × 4–5.5	15–22 × 5–7	Angiosperm
*A. hainanensis*	Present	Absent	30–80 × 4–5	70–80 × 3–4	7.5–8.5 × 3–4	Angiosperm
*A. heimuer*	Present	Absent	50–150 × 4–6.5	40–67 × 3–6.5	11–13 × 4–5	Angiosperm
*A. minor*	Present	Absent	90–120 × 6–10	35–43 × 2–4	7–8 × 3–4	Angiosperm
*A. minutissima*	Present	Present	50–85 × 5–6	46–67 × 5–8	13–15 × 5–6	Angiosperm
*A. tibetica*	Present	Absent	70–135 × 5–8	70–103 × 4–7	15–18.5 × 5.8–6.2	Gymnosperm
*A. villosula*	Absent	Absent	40–90 × 4.5–6	40–61 × 4–5	13–15.5 × 4–5	Angiosperm

**Table 6 jof-07-00933-t006:** A comparison of the species in the *Auricularia mesenterica* complex.

Name	Upper Surface	Crystals	Hairs (µm)	Basidia (µm)	Basidiospores (µm)	Cystidioles
*A. africana*	Tomentose	Present	300–500 × 1–2	50–80 × 4.5–6	12–14 × 4.9–5.4	Absent
*A. asiatica*	Villose	Present	800–1200 × 1.5–3	40–52 × 3–6	11.2–12.3 × 4.5–5.2	Absent
*A. brasiliana*	Tomentose	Absent	1000–1500 × 2–3.5	30–47 × 3–5	11.5–12 × 4.5	Absent
*A. mesenterica*	Hispid	Absent	1000–2000 × 2–4	48–68 × 4–6.5	14–17 × 4.7–5.2	Present
*A. orientalis*	Villose	Absent	900–1500 × 1.5–2.5	50–72 × 3.5–6	12.5–14.2 × 5–6	Present
*A. pusio*	Tomentose	Present	400–800 × 1.5–2	50–72 × 5–7.5	12–14 × 5–6	Absent
*A. srilankensis*	Villose	Present	400–800 × 1.5–3	50–70 × 5–7	12–14.7 × 4.5–6	Absent
*A. submesenterica*	Hispid	Present	600–1000 × 2–2.5	37–50 × 4–6.5	12.5–14.8 × 5.5–6	Absent

## Data Availability

Publicly available datasets were analyzed in this study. This data can be found here: [https://www.ncbi.nlm.nih.gov/; https://www.mycobank.org/page/Simple%20names%20search; http://purl.org/phylo/treebase, accessed on 10 August 2021, submission ID 28649].

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
