# Peer review of "Global Diversity and Updated Phylogeny of Auricularia (Auriculariales, Basidiomycota)"

_jof, 2021, doi:10.3390/jof7110933_

Round 1
Reviewer 1 Report
The authors in this work analyzed loads of morphological and molecular data proposing and elaborating the new Auricularia state of the art implemented with some taxonomical novelties. Anyway, the manuscript needs to be deeply improved before considering for publication.
English must be improved, it could be useful the support of an English-speaking specialist to choose the most fitting/technical word. Sentences sometimes sound wordy and some words didn’t fit (e.g. all over the world instead of worldwide). Some examples are “morphological characteristics” used instead of morphological characters, when morphology changes with the environmental conditions we should talk of plasticity, “taxonomic feature” used instead of “identification key” etc.
Better, but not perfect, the section with descriptions, most probably is due to its schematic, synthetic and repetitive structure.
Below are reported some not exhaustive examples of the improvements needed.
Note: The progressive line numbering is missed. This made reviewing and notes contextualization more difficult than due.
The Abstract is the main window on your work, it should be clear and sharp to grab the reader’s attention. This section should be re-organized because the speech is fragmentary and confusing. Below the sequence of items there present
General information Auricularia….
Methods nr of specimens analysed and Phylogenetic analyses were based
Results 37 Auricularia species are determined. Ten new species,
Methods Phylogenetic relationships based on the concatenated ITS+nLSU dataset and the concatenated ITS+nLSU+rpb1+rpb2 dataset of 31
Results …discussed. …..discussed. synopsis...identification keys
As you can note the background information is very synthetic, but a clear aim/scope of the work is missed… The readers should deduce aims reading results. Please rework
Keywords - Multilocus or multigene analysis/phylogeny and not multi-marker
Auricularia has a wide distribution all over the world and is very important due to its edibility and medicinal properties.
Auricularia has a worldwide distribution…
Introduction.
- Auricularia Bull. (Auriculariaceae, Auriculariales), typified by A. mesenterica (Dicks.) Pers. [1], has species that most inhabit angiosperm wood, such as dead trees, stumps, fallen trunks and branches, and rotten wood, and a few grow on gymnosperm wood [2,3].
- The genus plays an important role in degradation in forest ecosystems, especially in tropical forests.
These two sentences should be inverted… or better if the opening is changed with something like “Members of Auricularia are widely distributed and are recognized for their ecological and economic values and medicinal properties. Most species play an important role in degradation… inhabiting…..Several species are widely used as important edible……”
Barrett [15] and Kobayasi [16] tried to identify the species based on macro-morphological characteristics, such as colour and size of basidiomata and length of hairs. Tried to identify is inappropriate…they indeed tried to find the character able to discriminate/distinguish the different species
Something like that could be better “Species identification has been performed for a long time using macro-morphological characters, such as colour and size of basidiomata and length of hairs as introduced by Barrett and Kobayasi [15,16].
Lowy [10] considered the hyphal structure of internal stratification of basidiomata as an important character to distinguish different species, and subsequently, this structure was accepted as a taxonomic feature by some mycologists [12,17,18]
Lowy introduced as species discriminant character the hyphal structure of internal stratification of basidiomata species, and subsequently, this taxonomical key was accepted by some (?) mycologists [12,17,18].
Recently, molecular analyses were applied in the taxonomy of Auricularia, and some important species were revised
It’s better to modify the above sentence with something like “The introduction of molecular methods evidenced a number of misidentifications and some important species were revised.” Anyway, the list of misidentifications should be synthetized now is too long and creates some confusion…
Thus, the aim of this study is to substantiate the current knowledge of phylogeny and species diversity…What does it mean substantiate? Did you mean validate?
Thus, the aim of this study is to substantiate the current knowledge of phylogeny and species diversity of Auricularia world wide by studying 278 samples (including 26 type specimens) from 35 countries in Asia, Europe, North America, South America, Africa and Oceania. Rework the sentence please correct the spelling of worldwide. 278 samples were analysed not studied.
Traditional taxonomic methods are used in morphological studies, and the stable morphological features are screened. The sentence is not clear because 1) stable morphological characters (not features) had not introduced before, 2) is not clear what is the aforementioned screening; 3) even phylogeny-based new species should be also morphologically described;
The last two sentences of the Introduction should be reworked.
Material and methods.
Being the specimens analysed coming from the Beijing Forestry University herbarium (BJFC) and other international herbaria, it is necessary to deeply modify table 1.
The table layout should be changed to landscape mode. A column with herbaria numbers, a column to indicate the host/substrate, country, collector and date. As a note, you must report the acronym of collection. Moreover, table 1 is not in the right place because it should be placed immediately after the section where is mentioned for the first time.
DNA Extraction, PCR and Sequencing
CTAB rapid plant genome extraction kit-DN14 (Aidlab Biotechnologies Co., Ltd, Beijing) was used to obtain PCR products... There is an error: you obtained DNA and not PCR products
The reaction mix description is missed – e.g volume reaction, Taq or mix used (+supplier) DNA used for reaction etc.
ITS region was amplified with primer pairs ITS5 and ITS4 [33] (White et al. 1990). Nuclear LSU region was amplified with primer pairs LR0R and LR7…please add primer sequences since there are at least two LR0R. The same is also for rpb1, rpb2. I suggest moving all conditions in a table reporting primers names, sequence, amplification program and references…in this way the section increase its readability. Add information on sequencing… did you use the big dye terminator? Was the sequencing bidirectional?
Phylogenetic Analyses
Please improve the readability of this section in particular it should be clear the goal you want to achieve from each set of data
For example: The following sentence” The phylogeny of the concatenated ITS+nLSU dataset was established based on sequences of 156 specimens of Auriculariales “should be changed to something like “ In order to define the high-rank phylogenetic relations within Auriculariales a dataset composed of concatenated ITS+nLSU sequences resulting in a final alignment comprising 156 different specimens. Heteroradulum deglubens (Berk. & Broome) Spirin & Malysheva was used as outgroup.”
because the rpb1 or rpb2 sequences of some species failed to be generated. How were considered the alignment gaps within your alignment?
How many characters were present in concatenated sequences? I mean How many bp were used for each locus ITS xxbp, rDNA LSU xybp; rpb1 zzbp; rpb2 yy bp….
Maximum likelihood (ML) and Bayesian inference (BI) methods were used for both datasets. It is not clear in which way these methods were used. I mean both methods for each dataset or one method each. Please rework the sentence and explain why you use two methods.
Results
The dataset has an aligned length of 1953 characters, of which 600 characters are ITS partions and 1353 characters are nLSU partions. 1) What does it mean partion??? 2) these data are included in material and methods because necessary for trees construction…phylogenetic trees are results
Maybe to highlight the presence of different clades it could be useful to prepare a condensed tree…
In any case Figure 1 is nearly unreadable and should be split to increase the font size. There are some incongruences between materials and methods and caption/figure: bootstrapping in the former was fixed to 75 and not 50 and the BPP fixed to 0.95 and not 0.90 reported below the picture. More, the legend should be enlarged too as the coloured icons next to the strain. Plain dots or squares are better than stars
Figure 2 should be enlarged because is unreadable, also here there are incongruences between Materials and Methods and figure caption.
Taxonomy
All sections reporting “Specimens examined” should be cancelled. The information should be moved/synthesized in Table 1. The same for additional specimens
Captions should be separated and distinguishable from the text (font size etc) please check this item along with the text.
Discussions
Some not exhaustive examples
Currently 37 Auricularia species are recognized and belong to five species complexes: the A. auricula-judae complex, the A. cornea complex, the A. delicata complex, the A. fuscosuccinea complex, and the A. mesenterica complex. Please rework the sentence avoiding repetitions too
Auricularia was considered as polyphyletic group based on nLSU sequences [27,29], and it is showed monophyletic in our phylogeny based on the concatenated ITS+nLSU dataset (Figure 1) which is same as the result of recent phylogenetic analysis. This sentence sounds confusing please rework.
What about classifications devoid of DNA data? There are possibilities to cover this gap in next future etc. etc.
Groups mentioned in this section are not indicated in the relative figures
Tables 2-6 should be better formatted avoiding measures distributed in two lines …. About the distribution column, there are some inconsistencies in geographical data sometimes reported as nations and others as a continent and so on. Being this information already given, I suggest cancelling that column and use better space enlarging the remaining columns. Mind a gap between tables and discussion text.
Minor notes
References should be given just before punctuation signs.
Author Response
Dear reviewer
Thank you for your carefuf review and coments.
We have revised the manuscript according to your comments.
Thank you.
Best regards.
Point 1: English must be improved, it could be useful the support of an English-speaking specialist to choose the most fitting/technical word. Sentences sometimes sound wordy and some words didn’t fit (e.g. all over the world instead of worldwide). Some examples are “morphological characteristics” used instead of morphological characters, when morphology changes with the environmental conditions we should talk of plasticity, “taxonomic feature” used instead of “identification key” etc.
Response 1: Thank you for your careful review. Yes, English should be improved. We have revised the manuscript according to your comments.
Point 2: Better, but not perfect, the section with descriptions, most probably is due to its schematic, synthetic and repetitive structure.
Response 2: The section with descriptions is revised. Actually, the structure is repetitive because the same microscopic structures are present almost all species of Auricularia but with variation of size.
Below are reported some not exhaustive examples of the improvements needed.
Point 3: Note: The progressive line numbering is missed. This made reviewing and notes contextualization more difficult than due.
Response 3: The progressive line numbering is added now.
Point 4: The Abstract is the main window on your work, it should be clear and sharp to grab the reader’s attention. This section should be re-organized because the speech is fragmentary and confusing. Below the sequence of items there present
General information Auricularia….
Methods nr of specimens analysed and Phylogenetic analyses were based
Results 37 Auricularia species are determined. Ten new species,
Methods Phylogenetic relationships based on the concatenated ITS+nLSU dataset and the concatenated ITS+nLSU+rpb1+rpb2 dataset of 31
Results …discussed. …..discussed. synopsis...identification keys
As you can note the background information is very synthetic, but a clear aim/scope of the work is missed… The readers should deduce aims reading results. Please rework
Response 4: The Abstract is re-organized according to your comments.
Point 5: Keywords - Multilocus or multigene analysis/phylogeny and not multi-marker
Response 5: Multi-marker is revised as multigene.
Point 6: Auricularia has a wide distribution all over the world and is very important due to its edibility and medicinal properties.
Auricularia has a worldwide distribution…
Response 6: Yes, it is revised.
Point 7: Introduction.
- Auricularia Bull. (Auriculariaceae, Auriculariales), typified by A. mesenterica (Dicks.) Pers. [1], has species that most inhabit angiosperm wood, such as dead trees, stumps, fallen trunks and branches, and rotten wood, and a few grow on gymnosperm wood [2,3].
- The genus plays an important role in degradation in forest ecosystems, especially in tropical forests.
These two sentences should be inverted… or better if the opening is changed with something like “Members of Auricularia are widely distributed and are recognized for their ecological and economic values and medicinal properties. Most species play an important role in degradation… inhabiting…..Several species are widely used as important edible……”
Response 7: These two sentences are inverted and revised as “Members of Auricularia Bull. (Auriculariaceae, Auriculariales), typified by A. mesenterica (Dicks.) Pers., are widely distributed and are recognized for their ecological and economic values and medicinal properties. Most species play an important role in degradation in forest ecosystems, especially in tropical forests, usually inhabiting angiosperm wood, such as dead trees, stumps, fallen trunks and branches, and rotten wood, and a few growing on gymnosperm wood.”.
Point 8: Barrett [15] and Kobayasi [16] tried to identify the species based on macro-morphological characteristics, such as colour and size of basidiomata and length of hairs. Tried to identify is inappropriate…they indeed tried to find the character able to discriminate/distinguish the different species
Something like that could be better “Species identification has been performed for a long time using macro-morphological characters, such as colour and size of basidiomata and length of hairs as introduced by Barrett and Kobayasi [15,16].
Response 8: Thanks! It is revised.
Point 9: Lowy [10] considered the hyphal structure of internal stratification of basidiomata as an important character to distinguish different species, and subsequently, this structure was accepted as a taxonomic feature by some mycologists [12,17,18]
Lowy introduced as species discriminant character the hyphal structure of internal stratification of basidiomata species, and subsequently, this taxonomical key was accepted by some (?) mycologists [12,17,18].
Response 9: This sentence is revised as Lowy [10] introduced the hyphal structure of internal stratification of basidiomata as species discriminant character , and subsequently, this taxonomical key was accepted by Kobayasi [12], Lowy [17], and Li [18].
Point 10: Recently, molecular analyses were applied in the taxonomy of Auricularia, and some important species were revised
It’s better to modify the above sentence with something like “The introduction of molecular methods evidenced a number of misidentifications and some important species were revised.” Anyway, the list of misidentifications should be synthetized now is too long and creates some confusion…
Response 10: The sentence is revised, and the list of misidentifications are synthetized.
Point 11: Thus, the aim of this study is to substantiate the current knowledge of phylogeny and species diversity…What does it mean substantiate? Did you mean validate?
Thus, the aim of this study is to substantiate the current knowledge of phylogeny and species diversity of Auricularia world wide by studying 278 samples (including 26 type specimens) from 35 countries in Asia, Europe, North America, South America, Africa and Oceania. Rework the sentence please correct the spelling of worldwide. 278 samples were analysed not studied.
Response 11: The sentence is revised as “Thus, the aim of this study is to validate the current knowledge of phylogeny and species diversity of Auricularia worldwide by analyzing 278 samples (including 26 type specimens) from 35 countries in Asia, Europe, North America, South America, Africa and Oceania.”
Point 12: Traditional taxonomic methods are used in morphological studies, and the stable morphological features are screened. The sentence is not clear because 1) stable morphological characters (not features) had not introduced before, 2) is not clear what is the aforementioned screening; 3) even phylogeny-based new species should be also morphologically described;
The last two sentences of the Introduction should be reworked.
Response 12: The two sentences are revised as “According to morphological examinations and phylogenetic analyses based on ITS, nLSU, rpb1 and rpb2 sequences of those samples, 37 species of Auricularia are recognized, including ten species described as new to science, and morphological differences and phylogenetic relationships of species in five Auricularia complexes, the A. auricula-judae complex, the A. cornea complex, the A. delicata complex, the A. fuscosuccinea complex and the A. mesenterica complex are elaborated.”
Material and methods.
Point 13: Being the specimens analysed coming from the Beijing Forestry University herbarium (BJFC) and other international herbaria, it is necessary to deeply modify table 1.
The table layout should be changed to landscape mode. A column with herbaria numbers, a column to indicate the host/substrate, country, collector and date. As a note, you must report the acronym of collection. Moreover, table 1 is not in the right place because it should be placed immediately after the section where is mentioned for the first time.
Response 13: Other herbaria are added in the Material and methods. The table 1 is changed to landscape mode and is placed immediately after the section where is mentioned for the first time. Although 277 samples were examined, sequences of only 149 specimens of Auricularia were used in the phylogenetic analysis because some materials are not successfully sequenced. Therefore, all newly generated sequences and additional related sequences downloaded from GenBank are listed in Table 1, and a column to indicate country is added in Table 1. “Specimens examined” and “Additional specimens” are reserved in Taxonomy.
DNA Extraction, PCR and Sequencing
Point 14: CTAB rapid plant genome extraction kit-DN14 (Aidlab Biotechnologies Co., Ltd, Beijing) was used to obtain PCR products... There is an error: you obtained DNA and not PCR products
The reaction mix description is missed – e.g volume reaction, Taq or mix used (+supplier) DNA used for reaction etc.
Response 14: Yes, “PCR products” is revised as DNA, and 2×EasyTaq PCR SuperMix (TransGen biotech, China) was used for reaction to obtain PCR products.
Point 15: ITS region was amplified with primer pairs ITS5 and ITS4 [33] (White et al. 1990). Nuclear LSU region was amplified with primer pairs LR0R and LR7…please add primer sequences since there are at least two LR0R. The same is also for rpb1, rpb2. I suggest moving all conditions in a table reporting primers names, sequence, amplification program and references…in this way the section increase its readability. Add information on sequencing… did you use the big dye terminator? Was the sequencing bidirectional?
Response 15: Thank you very much for your suggestion about the primer pairs. The primer sequences are added in the text, and the conditions including primers names, sequence, amplification program and references can be found in the text. A table actually increase its readability, but only four primer pairs were used in this study. It is not absolutely necessary to add a table for primers and there are already six tables in this paper. DNA sequencing was performed at Beijing Genomics Institute, China. Sequences of nLSU, rpb1 and rpb2 were the sequencing bidirectional.
Phylogenetic Analyses
Point 16: Please improve the readability of this section in particular it should be clear the goal you want to achieve from each set of data
For example: The following sentence” The phylogeny of the concatenated ITS+nLSU dataset was established based on sequences of 156 specimens of Auriculariales “should be changed to something like “ In order to define the high-rank phylogenetic relations within Auriculariales a dataset composed of concatenated ITS+nLSU sequences resulting in a final alignment comprising 156 different specimens. Heteroradulum deglubens (Berk. & Broome) Spirin & Malysheva was used as outgroup.”
Response 16: Thanks, this section is re-organized as follow.
In order to define the high-rank phylogenetic relations within Auriculariales a dataset composed of concatenated ITS+nLSU sequences resulting in a final alignment comprising 156 different specimens. Heteroradulum deglubens (Berk. & Broome) Spirin & Malysheva was used as outgroup. The concatenated ITS+nLSU dataset has an aligned length of 1953 characters, of which 600 characters are from locus ITS, and 1353 characters are from locus nLSU. Another dataset composed of concatenated ITS+nLSU+rpb1+rpb2 sequences resulting in a final alignment comprising 106 different specimens to define phylogenetic relations within Auricularia, especially within closely related Auricularia species. Elmerina efibulata (Y.C. Dai & Y.L. Wei) Y.C. Dai & L.W. Zhou was used as outgroup to root trees because Elmerina Bres. is closer to Auricularia than other genera of Auriculariales in the phylogeny [30]. The concatenated ITS+nLSU+rpb1+rpb2 dataset has an aligned length of 3817 characters, of which 594 characters are from locus ITS, 1353 characters are from locus nLSU, 1336 characters are from locus rpb1, and 534 characters are from locus rpb2. All characters were equally weighted and gaps were treated as missing data.
Point 17: because the rpb1 or rpb2 sequences of some species failed to be generated. How were considered the alignment gaps within your alignment?
Response 17: Because the rpb1 or rpb2 sequences of some species failed to be generated, only 105 Auricularia specimens selecting from 149 Auricularia specimens with ITS or nLSU sequences were used to construct phylogenetic tree based on the concatenated ITS+nLSU+rpb1+rpb2 dataset. Gaps were treated as missing data.
Point 18: How many characters were present in concatenated sequences? I mean How many bp were used for each locus ITS xxbp, rDNA LSU xybp; rpb1 zzbp; rpb2 yy bp….
Response 18: They are added in the manuscript.
Point 19: Maximum likelihood (ML) and Bayesian inference (BI) methods were used for both datasets. It is not clear in which way these methods were used. I mean both methods for each dataset or one method each. Please rework the sentence and explain why you use two methods.
Response 19: The sentence is revised as “Maximum likelihood (ML) and Bayesian inference (BI) methods were used for each dataset to enhance the credibility of phylogenetic analyses”.
Results
Point 20: The dataset has an aligned length of 1953 characters, of which 600 characters are ITS partions and 1353 characters are nLSU partions. 1) What does it mean partion??? 2) these data are included in material and methods because necessary for trees construction…phylogenetic trees are results
Response 20: These data are included in material and methods.
Point 21: Maybe to highlight the presence of different clades it could be useful to prepare a condensed tree…
Response 21: Yes, they are revised.
Point 22: In any case Figure 1 is nearly unreadable and should be split to increase the font size. There are some incongruences between materials and methods and caption/figure: bootstrapping in the former was fixed to 75 and not 50 and the BPP fixed to 0.95 and not 0.90 reported below the picture. More, the legend should be enlarged too as the coloured icons next to the strain. Plain dots or squares are better than stars
Response 22: Figure 1 is split to two figures to increase the font size. The bootstrapping in the former is revised to 50 and 0.90. The legend is enlarged as the coloured icons next to the strain and revised as dots.
Point 23: Figure 2 should be enlarged because is unreadable, also here there are incongruences between Materials and Methods and figure caption.
Response 23: Figure 2 is enlarged and the bootstrapping is revised.
Taxonomy
Point 24: All sections reporting “Specimens examined” should be cancelled. The information should be moved/synthesized in Table 1. The same for additional specimens
Response 24: Although 277 samples were examined, sequences of only 149 specimens of Auricularia were used in the phylogenetic analysis because some materials are not successfully sequenced. Therefore, all newly generated sequences and additional related sequences downloaded from GenBank are listed in Table 1, and a column to indicate country is added in Table 1. “Specimens examined” and “Additional specimens” are reserved in Taxonomy.
Point 25: Captions should be separated and distinguishable from the text (font size etc) please check this item along with the text.
Response 25: Yes, captions usually should be separated and distinguishable from the text. But current form follows template of Journal of fungi.
Discussions
Some not exhaustive examples
Point 26: Currently 37 Auricularia species are recognized and belong to five species complexes: the A. auricula-judae complex, the A. cornea complex, the A. delicata complex, the A. fuscosuccinea complex, and the A. mesenterica complex. Please rework the sentence avoiding repetitions too
Response 26: The sentence is revised as “Currently 37 Auricularia species belonging to five species complexes including A. auricula-judae, A. cornea, A. delicata, A. fuscosuccinea, and A. mesenterica, are recognized based on morphological and molecular data.”
Point 27: Auricularia was considered as polyphyletic group based on nLSU sequences [27,29], and it is showed monophyletic in our phylogeny based on the concatenated ITS+nLSU dataset (Figure 1) which is same as the result of recent phylogenetic analysis. This sentence sounds confusing please rework.
Response 27: The sentence is revised as “Auricularia was considered as a polyphyletic group based on nLSU sequences [27,29], but it is showed monophyletic in our phylogeny based on the concatenated ITS+nLSU dataset (Figure 1) which confirms the result of recent phylogenetic analyses [30,31]”
Point 28: What about classifications devoid of DNA data? There are possibilities to cover this gap in next future etc. etc.
Response 28: The molecular data of these six species will be covered by collecting new samples from their type locality in next future.
Point 29: Groups mentioned in this section are not indicated in the relative figures
Response 29: Group I, III are indicated in Figures 1 and 2, Group II is indicated in Figure 1 only, because species in Group II are not clustered into one clade in Figure 2. For the convenience of discussion, species in Group II in Figure are still regarded as one group in present paper.
Point 30: Tables 2-6 should be better formatted avoiding measures distributed in two lines …. About the distribution column, there are some inconsistencies in geographical data sometimes reported as nations and others as a continent and so on. Being this information already given, I suggest cancelling that column and use better space enlarging the remaining columns. Mind a gap between tables and discussion text.
Response 30: The distribution column is cancelled, and gaps between tables and discussion text are modified.
Minor notes
Point 31: References should be given just before punctuation signs.
Response 31: They are revised.
Reviewer 2 Report
Here is the review of the paper entitled "Global diversity and updated phylogeny of Auricularia (Auriculariales, Basidiomycota)" written by Fang Wu & co-authors.
The authors analyzed 278 samples of the genus Auricularia originated from 35 countries in Asia, Europe, North America, South America, Africa, and Oceania. Morphological and multigene (ITS, nLSU, rpb1 and rpb2 marker genes) phylogenetic studies were performed and 10 species are described as new to science. Five Auricularia morphological species complexes are recognized and discussed: A. auricula-judae , A. cornea, A. delicata, A. fuscosuccinea and A. mesenterica species complex.
The manuscript needs to be thoroughly revised. I have made more than 240 corrections/comments in the pdf file attached. The English language should be much better and some parts of the text could be misunderstood because the quality of the English language used in the manuscript is inadequate. The text in species descriptions is frequently copied from some of the previous species descriptions so the same errors are continuously repeated throughout the text.
Best, Reviewer

Author Response
Dear reviewer 2
Thank you for your carefuf review and coments.
We have revised the manuscript according to your comments.
Thank you.
Best regards.
Point 1: The manuscript needs to be thoroughly revised. I have made more than 240 corrections/comments in the pdf file attached. The English language should be much better and some parts of the text could be misunderstood because the quality of the English language used in the manuscript is inadequate. The text in species descriptions is frequently copied from some of the previous species descriptions so the same errors are continuously repeated throughout the text.
Response 1: We appreciate you for the elaborate reading and the thoughtful comments and suggestions.. We have carefully revised the manuscript according to your comments.
Point 2: Comment: In abstract you state that you described 10 new species not 9, please check.
Response 2: Sorry, it is revised as ten new species.
Point 3: Comment: Please list additional herbaria here.
Response 3: Additional herbaria are list.
Point 4: Comment: Please include geographical coordinates (latitude, longitude) for localities of all holotype collections. The same applies for all species newly described in the paper. It should became rule for all new species descriptions. Please refer to Aime & al. (2021) How to publish a new fungal species, or name, version 3.0. IMA Fungus.
Response 4: The geographical coordinates (latitude, longitude) for localities of holotypes of all new species are added.
Point 5: Comment: There is no bar added in the photo h.
Response 5: A bar is added in photo h.
Point 6: some repeated errors, A cross-section from basidiomata > A cross-section of basidiomata; dying > drying; usually in middle of the cross-section > usually in the middle of the cross-section; usually smooth without folds > usually smooth, without folds; and distribution in -> and is distributed in
Response 6: Those repeated errors are revised throught the text, and also some other revisions in English languish.
Point 7: Please include full description here, designate holotype etc. There is no point to have monograph of Auricularia here and have to look for a description of this species in other work.
Response 7: The morphological descriptions of recently described species including A. angiospermarum, A. brasiliana, A. heimuer, A. minutissima, A. orientalis and A. tibetica are added, and the holotype of A. angiospermarum is designated.
Point 8: Comment: Include the name of the person who found type material.
Response 8: The collectors of the almost all specimens are listed except some loaned specimens that their collectors can not be found.
Point 9: reticulata (word is not used in English). Maybe reticulate or reticulated?
Response 9: reticulata is revised as reticulate.
Point 10: Comment: smaller basidiospores? In comparison with some other species? Maybe it would be better to say "relatively small basidiospores".
Response 10: Yes, it is revised as relatively small basidiospores.
Point 11: Comment: Did you try to obtain sequences and failed? Or you simply claim that the specimens are too old without even trying. Please comment accordingly.
Response 11: We actually have tried to obtain sequences and failed. The sentence is revised as "We failed to extract DNA becuase the type material is in poor condition. The similar texts in other species are revised, too.
Point 12: Question: morphological or molecular, or both? Please define.
Response 12: "this difference" is revised as "morphological difference".
Point 13: up drying -> drying up
Response 13: "up drying" is revised as "upon drying" because "upon drying" is used in other species description.
Point 14: Comment: which size?
Response 14: "big basidiospores" is deleted.
Point 15: Comment: According to your phylogram It is a type collection, so please mark this here as type. Go through the whole manuscript text and mark all type collections in specimens examined section for all species.
Response 15: They are marked.
Point 16: Comment: you sre stating that A. mesenterica is a species complex. On the basis of which study you are claiming this? Morphology, phylogenetic studies or you need to cite the literature? Please clarify.
Response 16: The literature [25] is added.
Point 17: Comment: It is not a paratype. Definition of paratype: A specimen not formally designated as a type but cited along with the type collection in the original description of a taxon. A. pusio was described in 19th century and later collection of the same species cannot be a paratype.
Response 17: Thank for your explanation. The ", paratype" is deleted.
Point 18: Comment: You listed six specimens in specimens examined and here you wrote that you studied two? Please check.
Response 18: Yes, it is six.
Point 19: Question: Do you mean to write number of A. submesenterica samples that you phylogenetically studied?
Response 19: Yes, it is. The sentence is revised as " These eight species form eight independent lineages with high support in the phylogeny based on the concatenated ITS+nLSU dataset except A. submesenterica (Figure 1), but six samples of A. submesenterica form a single lineage with high support in the phylogeny based on the concatenated ITS+nLSU+rpb1+rpb2 dataset (Figure 2)".
Reviewer 3 Report
Global diversity and updated phylogeny of Auricularia (Auriculariales, Basidiomycota)
Fang Wu, Ablat Tohtirjap, Long-Fei Fan, Li-Wei Zhou, Renato L. M. Alvarenga, Tatiana B. Gibertoni and Yu-Cheng Dai
Review
This manuscript is an excellent quality taxonomically and also phylogenetically. All new species are well supported. Also English is very good as well as photos. This manuscript is fully suitable for JOF.
Here are some (minor) suggestions and questions:
The more important notes:
- Only one colour in the fresh condition is given for several species. This also applies to e. g. Auricularia auricula-judae and A. mesenterica, which certainly have more colours than are mentioned in these descriptions (based on literature and also based on my own field experience). A. nigricans without any mention about blackish colour of any part of basidioma is somewhat confusing to me. Please, add another colours in all species where it is appropriate.
- In some cases, the collectors of individual collections are not listed including some holotypes; also some abbreviations of (maybe) personal herbariums are not explained, in others they are. Please add these data.
- For some new species, a description when fresh is mentioned, but none of the collections were probably collected by any of the authors of the new species, but only thanks to the herbariums from which the material was borrowed is mentioned. If the descriptions of fresh collections are based on the descriptions of collectors, please add these collectors to the acknowledgment. If the descriptions are based on revived basidiomata, change this information in the description.
- Some names traditionally considered synonyms of Auricularia auricula-judae are not mentioned in this work either in the list of synonyms or as separate taxa. It would be good to comment on at least some of them, or to list them as synonyms although they are old or little known.
- Some microscopic features of Auricularia auricula-judae have bigger variability according to the literature and my personal experiences (collections from the Central Europe); than based on your measurements. It would be good to mention these differences in the discussion part, at least the width of the spores and a numbers of septa (rarely are 2 or 4). The similar situation is in A. mesenterica.
- Photographs of new species of A. camposii spores are missing. Please, add some photo or drawing of spores.
- Page 26: The authors wrote: Another species, A. leucochroma, was also described from Asia [16] and was treated as a synonym of A. nigricans (http://www.indexfungorum.org/Names/NamesRecord.asp?RecordID=326832, accessed on 3 September 2021). However, the original literature about this synonymisation (Looney et al. 2013) have to be studied; and confirmed/ rejected your opinion based on this text if species are really synonyms or not. Also the colours used in the protologue is recommended to add to the description.
- You wrote: In addition, Cameroon is very close to Equatorial Guinea, so we consider the specimen (K 43873) represents the real A. delicata or A. delicata sensu stricto. It is not clear from this text what difference you mean when you use A. delicata or A. delicata sensu stricto. Moreover, if no type was preserved/ found from Equatorial Guinea, I suggest to establish K 43873 as neotype – or epitype if you afraid some type may be find later.
- Both trees are nice and well-supported, only check the quality of A. pilosa sequences
Details:
- In several cases you wrote that type is too old to extract DNA when these types are from 1980-1989. Because for some genera is possible to sequenced collections from 1920-1950, this formulation is not ideal. I suggest to rewrite it and point rather to poor material condition or insufficient type material.
- For many species, the the phrase upon dying is used. This is not a phrase used in mycology, but in medicine for death of vital functions. Please reformulate (Drying? Fading?).
- The word determined is used in the abstract. This word is used for predestination rather than for identification. Please use the word
- Page 18, part Specimens examined: France...Orlienas is it Orliénas or Orléans, please, choose which is correct and add a diacritical mark
- Page 18, part Specimens examined: Malysheva (LE254030) – E.F. or V. Malysheva? Both twins are mycologists.
- Page 3, line 3 Most of studied specimens... add The (The most of studied specimens

Author Response
Dear reviewer 3
Thank you for your carefuf review and coments.
We have revised the manuscript according to your comments.
Thank you.
Best regards.
Point 1: Only one colour in the fresh condition is given for several species. This also applies to e. g. Auricularia auricula-judae and A. mesenterica, which certainly have more colours than are mentioned in these descriptions (based on literature and also based on my own field experience). A. nigricans without any mention about blackish colour of any part of basidioma is somewhat confusing to me. Please, add another colours in all species where it is appropriate.
Response 1: Thank you for your careful review. Auricularia auricula-judae also has white basidioma, and A. mesenterica has buff to white colour. They are added in the manuscript. A. nigricans becomes dark-grey upon drying. The additional colours of other species are added in manuscript.
Point 2: In some cases, the collectors of individual collections are not listed including some holotypes; also some abbreviations of (maybe) personal herbariums are not explained, in others they are. Please add these data.
Response 2: The collectors of the almost specimens are listed except a few loaned specimens that their collectors were abbreviated in the origial labels.
Point 3: For some new species, a description when fresh is mentioned, but none of the collections were probably collected by any of the authors of the new species, but only thanks to the herbariums from which the material was borrowed is mentioned. If the descriptions of fresh collections are based on the descriptions of collectors, please add these collectors to the acknowledgment. If the descriptions are based on revived basidiomata, change this information in the description.
Response 3: Some new species descriptions are based on the dry herbarium specimen, and their fresh characteristics are described as specime soaked. The sentence “morphological description when fresh of some species are described after their basidiomata were soaked” is added in Materials and Methods.
Point 4: Some names traditionally considered synonyms of Auricularia auricula-judae are not mentioned in this work either in the list of synonyms or as separate taxa. It would be good to comment on at least some of them, or to list them as synonyms although they are old or little known.
Response 4: The related synonyms are added.
Point 5: Some microscopic features of Auricularia auricula-judae have bigger variability according to the literature and my personal experiences (collections from the Central Europe); than based on your measurements. It would be good to mention these differences in the discussion part, at least the width of the spores and a numbers of septa (rarely are 2 or 4). The similar situation is in A. mesenterica.
Response 5: Yes, the basidiospores of Auricularia auricula-judae actually have big variability. Basidiopores of additional three specimens were measured, and data of those basidospores are integrated into previous data. The sentence “samples of these seven species in the complex are nested in seven lineages with high support in our phylogenies, and intraspecific genetic variations obviously present among samples of these taxa.”is revised as “Samples of these seven species in the complex are nested in seven lineages with high support in our phylogenies, and intraspecific genetic variations are obviously presented among samples of these taxa which is probably the reason why A. auricula-judae has big variability in the size of basidiospores”. However, we did not find the same situation in A. mesenterica.
Point 6: Photographs of new species of A. camposii spores are missing. Please, add some photo or drawing of spores.
Response 6: They are added.
Point 7: Page 26: The authors wrote: Another species, A. leucochroma, was also described from Asia [16] and was treated as a synonym of A. nigricans (http://www.indexfungorum.org/Names/NamesRecord.asp?RecordID=326832, accessed on 3 September 2021). However, the original literature about this synonymisation (Looney et al. 2013) have to be studied; and confirmed/ rejected your opinion based on this text if species are really synonyms or not. Also the colours used in the protologue is recommended to add to the description.
Response 7: The sentence is revised as ‘Another species, A. leucochroma, was also described from Asia [16], and treated as a synonym of A. nigricans [20]. We rejected this opinion, because we examined the type material of A. leucochroma, and treated it as a synonym of A. cornea.
Point 8: You wrote: In addition, Cameroon is very close to Equatorial Guinea, so we consider the specimen (K 43873) represents the real A. delicata or A. delicata sensu stricto. It is not clear from this text what difference you mean when you use A. delicata or A. delicata sensu stricto. Moreover, if no type was preserved/ found from Equatorial Guinea, I suggest to establish K 43873 as neotype – or epitype if you afraid some type may be find later.
Response 8: The sentence is revised as In addition, Cameroon is very close to Equatorial Guinea, so we consider the specimen (K 43873) represents the real A. delicata and K 43873 is treated as epitype of A. delicata.
Point 9: Both trees are nice and well-supported, only check the quality of A. pilosa sequences
Response 9: Thank you, we have checked the quality of A. pilosa sequences.
Point 10: In several cases you wrote that type is too old to extract DNA when these types are from 1980-1989. Because for some genera is possible to sequenced collections from 1920-1950, this formulation is not ideal. I suggest to rewrite it and point rather to poor material condition or insufficient type material.
Response 10: The sentence type is too old to extract DNA... is revised as “We failed to extract DNA, because the type material is in poor condition”.
Point 11: For many species, the phrase upon dying is used. This is not a phrase used in mycology, but in medicine for death of vital functions. Please reformulate (Drying? Fading?).
Response 11: Sorry! upon dying is revised as upon drying.
Point 12: The word determined is used in the abstract. This word is used for predestination rather than for identification. Please use the word identified.
Response 12: Yes, it is revised as identified.
Point 13: Page 18, part Specimens examined: France...Orlienas is it Orliénas or Orléans, please, choose which is correct and add a diacritical mark
Response 13: Sorry, it is revised as Orliénas
Point 14: Page 18, part Specimens examined: Malysheva (LE254030) – E.F. or V. Malysheva? Both twins are mycologists.
Response 14: It is V. Malysheva, and revised as V. Malysheva.
Point 15: Page 3, line 3 Most of studied specimens... add The (The most of studied specimens)
Response 15: It is done.
Round 2
Reviewer 1 Report
Beyond loads of data, work and analysis this manuscript is still not suitable for publication. It is a pity.
As evidenced previously, the English used by authors needs strong improvements and the manuscript should be better organized and be consistent in all sections. Moreover, the authors should give the opportunity to the reviewers to work at their best giving readable files and pages correctly numbered. As a matter of fact, line numbers have been partially added (pages 4-5 haven’t and courts restart on page 12) but the corrections are so tight to make the manuscript unreadable in large sections. Even if it is requested to track corrections the best way to apply it is to accept cancelled lines leaving readable the other changes. This habit is useful for both authors and reviewers.
The figures showing trees are still not consistent in font size and style, legend dots position (sometimes after the strain others at the end of the box), species complex name, and legend position. Captions should be improved.
Below are some additional but not exhaustive notes.
L16 phylogenetic change to multi-gene phylogenetic analysis
L17 North America, South America- North and South America
L18-19 Phylogenetic analyses were based on the concatenated ITS+nLSU dataset and the concatenated ITS+nLSU+rpb1+rpb2 data set sequences of ITS, nLSU, rpb1 and rpb2 using methods of Maximum likelihood and Bayesian analyses – this detail is not requested in the abstract. Please remove
L29 illustration? What do you mean?
L29 ecological habits please change to ecological traits
L67- 68 However, the macro-morphological characters are variable with different temperatures, humidities, light intensities and growth stages, therefore resulting in inaccurate identification of similar species- As told previously, the phenomenon described is called plasticity. Please amend the sentence
L70-88 the sentences should be reworked because are quite confused and written in poor English.
L97-99 “In addition, the previous phylogenetic analysis based on nLSU sequences showed that Auricularia is polyphyletic [27–29], but the genus Auricularia is showed as monophyletic group in recent multi-gene phylogenetic analysis”. The sentence sounds confusing because it’s hard to imagine a change like that using the same specimen. If its explanation is that at lines 101-103: (1) “Although afore-mentioned” should not open a new section; (2) the references cannot be different; (3) if the previous assumptions were wrong because partial, it is better to talk about on reliable results only.
L106-113 please rewrite in understandable English considering that (1) is not necessary to list the molecular markers you used because there is the materials & methods section; (2) relationships (L111) is used to describe connections between people; (3) ten species described as new to science is wordy.
Page 4 The most of studied specimens are deposited at the herbarium of Beijing Forestry University (BJFC) and some materials were loaned from other herbaria – check for grammar issues as the rest of the section considering that should be explanatory and not a telegram. “Morphological description when fresh of some species are described after their basidiomata were soaked.” Is understandable. Moreover “most” and “some” are indefinite quantities that don’t fit with scientific reports style the authors should give numbers! In the abstract 277 specimens are reported…so the authors should explain how they were used and how many were considered for each step.
DNA Extraction, PCR and Sequencing how many samples were processed?
Page 4 phylogenetic analysis: “All newly generated sequences were submitted to GenBank and were aligned with additional related sequences downloaded” is wordy, please rework it
F 234519, type - Did you mean holotype?
Table 1. Newly generated sequences for this study are indicated in bold. Should be part of the caption
Page 12L4-20 even if reporting the information requested, needs to be improved
Same page L22 reliability and not credibility
L25-33 Please rewrite. The models SYM+I+G and GTR+I+G should be “The SYM+I+G and GTR+I+G models”; the plural of locus is loci; models are applied to alignments and not to loci as written
L50-51 rework
Results 149 are not 277 reported in the abstract
L53-54 Most and except several species are contradictory…please improve the whole section.
L64-65 The analysis showed that the morphological complexes do not fully correspond to the phylogenetic clades. Of what “morphological complexes” are you talking at?
Author Response
Point 1: Beyond loads of data, work and analysis this manuscript is still not suitable for publication. It is a pity.
As evidenced previously, the English used by authors needs strong improvements and the manuscript should be better organized and be consistent in all sections. Moreover, the authors should give the opportunity to the reviewers to work at their best giving readable files and pages correctly numbered. As a matter of fact, line numbers have been partially added (pages 4-5 haven’t and courts restart on page 12) but the corrections are so tight to make the manuscript unreadable in large sections. Even if it is requested to track corrections the best way to apply it is to accept cancelled lines leaving readable the other changes. This habit is useful for both authors and reviewers.
The figures showing trees are still not consistent in font size and style, legend dots position (sometimes after the strain others at the end of the box), species complex name, and legend position. Captions should be improved.
Response 1: The English is improved by a native English speaker. Line numbers have been added. The cancelled lines are accepted. The figures showing trees and captions are improved. Captions in different species are in bold.
Below are some additional but not exhaustive notes.
Point 2: L16 phylogenetic change to multi-gene phylogenetic analysis
Response 2: Yes, it is done.
Point 3: L17 North America, South America- North and South America
Response 3: Yes, it is done.
Point 4: L18-19 Phylogenetic analyses were based on the concatenated ITS+nLSU dataset and the concatenated ITS+nLSU+rpb1+rpb2 data set sequences of ITS, nLSU, rpb1 and rpb2 using methods of Maximum likelihood and Bayesian analyses – this detail is not requested in the abstract. Please remove
Response 4: The sentence is revised as "Phylogenetic analyses were based on ITS, nLSU, rpb1 and rpb2 sequences using methods of Maximum Likelihood and Bayesian Inference analyses.".
Point 5: L29 illustration? What do you mean?
Response 5: "illustrations" is revised as "photos".
Point 6: L29 ecological habits please change to ecological traits
Response 6: Yes, it is done.
Point 7: L67- 68 However, the macro-morphological characters are variable with different temperatures, humidities, light intensities and growth stages, therefore resulting in inaccurate identification of similar species- As told previously, the phenomenon described is called plasticity. Please amend the sentence
Response 7: The sentence is revised as " However, the macro-morphological characters of Auricularia species present plasticity, resulting in inaccurate identification of similar species."
Point 8: L70-88 the sentences should be reworked because are quite confused and written in poor English.
Response 8: They are reworked.
Point 9: L97-99 “In addition, the previous phylogenetic analysis based on nLSU sequences showed that Auricularia is polyphyletic [27–29], but the genus Auricularia is showed as monophyletic group in recent multi-gene phylogenetic analysis”. The sentence sounds confusing because it’s hard to imagine a change like that using the same specimen. If its explanation is that at lines 101-103: (1) “Although afore-mentioned” should not open a new section; (2) the references cannot be different; (3) if the previous assumptions were wrong because partial, it is better to talk about on reliable results only.
Response 9: The nLSU sequences usually have lower identity than ITS sequences, so some species of Auricularia including A. fuscosuccinea, A. auricula-judae, A. delicata and A. mesenterica et al. were not clustered into one clade in the phylogenetic analysis based on nLSU sequences by Weiß and Oberwinkler (2001), Zhou et al. (2013) and Sotome et al. (2014), but these species were clustered into one clade with high support in the multi-gene phylogenetic analysis based on ITS and nLSU sequences by Malysheva et al. (2017) and Yuan et al. (2018). Therefore, the sentence is revised as "In addition, the phylogenetic analysis based on nLSU sequences showed that Auricularia was polyphyletic [27–29], but the genus Auricularia was showed as monophyletic group in recent multi-gene phylogenetic analysis based on ITS and nLSU sequences [30,31]."
Point 10: L106-113 please rewrite in understandable English considering that (1) is not necessary to list the molecular markers you used because there is the materials & methods section; (2) relationships (L111) is used to describe connections between people; (3) ten species described as new to science is wordy.
Response 10: They are rewritten. The molecular marker and ten species described as new to science are deleted. "relationships" is changed to "relations" throughout the text.
Point 11: Page 4 The most of studied specimens are deposited at the herbarium of Beijing Forestry University (BJFC) and some materials were loaned from other herbaria – check for grammar issues as the rest of the section considering that should be explanatory and not a telegram. “Morphological description when fresh of some species are described after their basidiomata were soaked.” Is understandable. Moreover “most” and “some” are indefinite quantities that don’t fit with scientific reports style the authors should give numbers! In the abstract 277 specimens are reported…so the authors should explain how they were used and how many were considered for each step.
Response 11: The first sentence is revised as "180 specimens were deposited at the herbarium of the Beijing Forestry University (BJFC) and 97 specimens were borrowed from other herbaria ". “Morphological description when fresh of some species are described after their basidiomata were soaked" is revised as " Macro-morphological description of thirteen species based on 31 herbarium specimens are described after their whole basidiomata were rehydrated." The "When fresh" in those thirteen species are revised as "When rehydrated "
Point 12: DNA Extraction, PCR and Sequencing how many samples were processed?
Response 12: CTAB rapid plant genome extraction kit-DN14 (Aidlab Biotechnologies Co., Ltd, Beijing) was used to obtain DNA from 277 dried specimens, according to the manufacturer’s instructions with some modifications.
Point 13: Page 4 phylogenetic analysis: “All newly generated sequences were submitted to GenBank and were aligned with additional related sequences downloaded” is wordy, please rework it
Response 13: It is revised as "rapid plant genome extraction kit-DN14 (Aidlab Biotechnologies Co., Ltd, Beijing) was used to obtain DNA from 277 dried specimens, according to the manufacturer’s instructions with some modifications."
Point 14: F 234519, type - Did you mean holotype?
Response 14: Yes, it is revised as holotype.
Point 15: Table 1. Newly generated sequences for this study are indicated in bold. Should be part of the caption
Response 15: The sentence "Newly generated sequences for this study are indicated in bold." is removed to left.
Point 16: Page 12L4-20 even if reporting the information requested, needs to be improved
Response 16: They are improved.
Point 17: Same page L22 reliability and not credibility
Response 17: It is done.
Point 18: L25-33 Please rewrite. The models SYM+I+G and GTR+I+G should be “The SYM+I+G and GTR+I+G models”; the plural of locus is loci; models are applied to alignments and not to loci as written
Response 18: It is done.
Point 19: L50-51 rework
Response 19: They are reworked.
Point 20: Results 149 are not 277 reported in the abstract
Response 20: Although 277 samples were examined, sequences of only 149 specimens of Auricularia were used in the phylogenetic analysis because some materials are not successfully sequenced.
Point 21: L53-54 Most and except several species are contradictory…please improve the whole section.
Response 21: They are improved.
Point 22: L64-65 The analysis showed that the morphological complexes do not fully correspond to the phylogenetic clades. Of what “morphological complexes” are you talking at?
Response 22: In present study, the species complex is mainly based on their morphology.
Reviewer 2 Report
Dear all,
The paper is now substantially improved according to my comments and I recommend it for publication in JoF.
Best, Reviewer
Author Response
Dear reviewer
Thank you for your review again.
Best regards
Fang Wu